# How to Probe: Simple Yet Effective Techniques for Improving Post-hoc Explanations

**Siddhartha Gairola[1,2], Moritz Böhle[†,3], Francesco Locatello[2], Bernt Schiele[1]**

[1]Max Planck Institute for Informatics, Saarland Informatics Campus, Germany,

[2]Institute of Science and Technology Austria, [3]Kyutai, France.

{sgairola, schiele}@mpi-inf.mpg.de, moritz@kyutai.org, francesco.locatello@ista.ac.at

[†] Work done while at MPI for Informatics

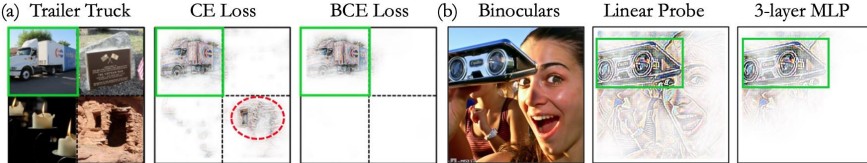

Figure 1: **(a)** Using Binary Cross-Entropy (BCE, col. 3) instead of Cross-Entropy (CE, col. 2) to train linear probes on **identical backbones** significantly increases the class-specificity of explanations. **(b)** Non-linear probes better extract class-specific features from pretrained representations: moving from a linear to a 3-layer BCE probe noticeably improves class localization. These critical findings are observed across a wide range of pre-training and attribution methods. Here, we show B-cos (Böhle et al., 2022) explanations for a model pre-trained using DINO (Caron et al., 2021).

## Abstract

Post-hoc importance attribution methods are a popular tool for "explaining" Deep Neural Networks (DNNs) and are inherently based on the assumption that the explanations can be applied independently of how the models were trained. Contrarily, in this work we bring forward empirical evidence that challenges this very notion. Surprisingly, we discover a strong dependency on and demonstrate that the training details of a pre-trained model's classification layer ($<10\%$ of model parameters) play a crucial role, much more than the pre-training scheme itself. This is of high practical relevance: (1) as techniques for pre-training models are becoming increasingly diverse, understanding the interplay between these techniques and attribution methods is critical; (2) it sheds light on an important yet overlooked assumption of post-hoc attribution methods which can drastically impact model explanations and how they are interpreted eventually. With this finding we also present simple yet effective adjustments to the classification layers, that can significantly enhance the quality of model explanations. We validate our findings across several visual pre-training frameworks (fully-supervised, self-supervised, contrastive vision-language training), model architectures and analyze how they impact explanations for a wide range of attribution methods on a diverse set of evaluation metrics. *Code available at:* *https://github.com/sidgairo18/how-to-probe*

## 1 Introduction

Most prominently in image classification models, importance attribution methods have emerged as a popular approach to mitigate the 'black box' problem of modern Deep Neural Networks (DNNs) (Samek et al., 2021). These methods assign importance values to input features, such as image pixels, to help humans understand why a DNN arrived at a particular classification decision.

While most attribution methods do not take the model training into account (are applied "post-hoc"), given the recent emergence of increasingly diverse pre-training paradigms for representation learning (Chen et al., 2020a;b; He et al., 2019; Chen et al., 2020c; Grill et al., 2020; Caron et al., 2021; Chen & He, 2020; Caron et al., 2020; Bardes et al., 2022; Xie et al., 2023; Li et al., 2023a; Radford et al., 2021) and the popularity of importance attribution methods (Bach et al., 2015; Selvaraju et al., 2017; Böhle et al., 2022; Shrikumar et al., 2017) to examine model decisions, a better understanding of the interplay between model explanations and training details is of great practical importance.

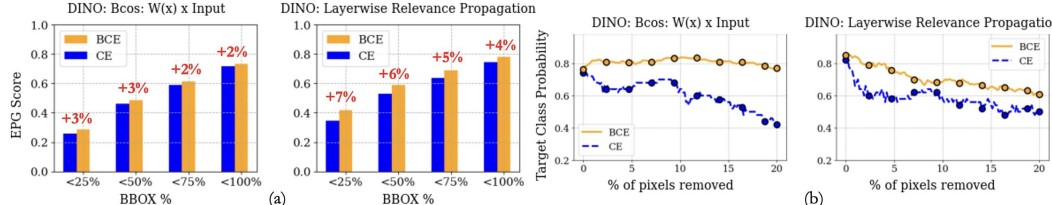

Figure 2: **Impact of Loss (BCE vs. CE).** (a) EPG Scores, and (b) Pixel Deletion scores for Bcos and LRP attributions for a **linear probe** when trained on **frozen** pre-trained features (DINO in this case). We find that BCE probes lead to more localized and stable attributions, thus highlighting the significant impact of the **loss function** on well-established attribution methods. Interestingly, despite the fact that the models differ *only by a single classification layer*, the attributions show stark differences in commonly used metrics for evaluating their quality.

In this work, we present an important finding that raises questions about the underlying assumptions of post-hoc attribution methods and their utility on downstream tasks. In particular, we find that the quality of attributions for pre-trained models can be highly dependent on how the classification head (i.e. the 'probe') is trained, even if the model backbone remains frozen. For example, in figure Figure 2a we show that the localization scores of B-cos (Böhle et al., 2022) and LRP (Bach et al., 2015) attributions significantly differ depending on which loss (BCE / CE) is used, with consistent improvements under the BCE loss; this can also be observed qualitatively, see Figure 1a. Similar improvements can also be seen under the commonly used pixel deletion paradigm for evaluating attribution methods (Figure 2b), despite the fact that the two models *only differ by a single linear layer* and the linear probes themselves have limited modeling capacity (see discussion Appendix B).

Importantly, we find these effects to be robust across several pre-training paradigms, model architectures and attribution methods on downstream classification tasks. Specifically, we conduct a thorough evaluation of the resulting explanations on a wide range of common interpretability metrics, that include object localization (Zhang et al., 2018; Samek et al., 2017; Wang et al., 2020), pixel deletion (Samek et al., 2017; Hedström et al., 2024), compactness (Chalasani et al., 2018) and complexity (Tseng et al., 2020) measures. By this we demonstrate empirically that training the probes via the binary cross-entropy (BCE) loss, as opposed to the conventionally used cross-entropy (CE) loss, leads to consistent and significant gains across several interpretability metrics, which thus might have important implications for many DNN-based applications.

To further study the impact of the probes, we investigate the interplay between the probe's complexity and the properties of the resulting explanations. Interestingly, we find that more complex (i.e. multi-layer) probes can better distill class-specific features from the pre-trained representations and thus significantly increase the localization performance (cf. Figure 1b), in particular when using interpretable B-cos Multi-Layer Perceptrons (MLPs) (Böhle et al., 2024).

**In short**, we make the following contributions: **(1)** We identify and analyze a critical yet overlooked problem of importance attribution methods, namely that how models are trained can significantly impact the resulting attributions. **(2)** We show both quantitatively and qualitatively that even when models differ only in their linear probe, explanations can dramatically differ based on the objective used to train the probe on downstream tasks; in particular, we find that using BCE instead of CE leads to significantly improved explanations when evaluated on a diverse set of interpretability metrics. **(3)** We demonstrate that our findings are independent of how the visual encoder (both convolutional and vision transformer architectures) is trained by conducting a detailed study across supervised, self-supervised (MoCov2 (He et al., 2019), DINO (Caron et al., 2021), BYOL (Grill et al., 2020)), and vision-language-based learning (CLIP, (Radford et al., 2021)). For this, we use diverse explanation methods (LRP (Bach et al., 2015), IntGrad (Sundararajan et al., 2017), B-cos (Böhle et al., 2022), Input×Gradients (Shrikumar et al., 2017), GradCAM (Selvaraju et al., 2017), ScoreCAM (Wang et al., 2020), LIME (Ribeiro et al., 2016)), CausalX (Xie et al., 2022), CGW1 (Chefer et al., 2020), Rollout (Abnar & Zuidema, 2020) and assess the quality of the resulting explanations on multiple datasets (ImageNet (Russakovsky et al., 2015), VOC (Everingham et al., 2009), and COCO (Lin et al., 2014)). **(4)** Furthermore, we find that non-linear B-cos MLP probes further boost downstream performance and 'class-specific' localization ability of attribution methods across pre-trained backbones. **(5)** We also show for the first time that the inherently interpretable B-cos (Böhle et al., 2022) models are compatible with SSL approaches, preserving both their performance and interpretability.

## 2    RELATED WORK

**Importance Attributions.** To understand deep neural networks (DNNs), several *post-hoc* attribution methods (Selvaraju et al., 2017; Chattopadhyay et al., 2018; Bach et al., 2015; Jiang et al., 2021; Wang et al., 2020; Petsiuk et al., 2018; Shrikumar et al., 2017; Sundararajan et al., 2017; Springenberg et al., 2015; Simonyan et al., 2014; Teney et al., 2020; Fong & Vedaldi, 2017; Ribeiro et al., 2016; Dabkowski & Gal, 2017; Xie et al., 2022; Chefer et al., 2020; Abnar & Zuidema, 2020), as well as *inherently interpretable* models (Chen et al., 2019; Brendel & Bethge, 2019; Böhle et al., 2021; 2022) have been developed. The attribution (or explanation) map summarizes the DNN computation and assigns importance values to the pixels that the model has used to make a decision. Such attribution methods can be broadly classified into the following categories: (1) *backpropagation-based* (Bach et al., 2015; Sundararajan et al., 2017; Shrikumar et al., 2017; Springenberg et al., 2015) that rely on gradients computed either with respect to the input or intermediate layers, (2) *perturbation-based* (Ribeiro et al., 2016; Petsiuk et al., 2018) that assign importance by noting the change in output on perturbing the input while treating the network as a 'black-box', (3) *activation-based* (Selvaraju et al., 2017; Jiang et al., 2021; Wang et al., 2020; Desai & Ramaswamy, 2020) that leverage the weights of activation maps across different layers to assign importance values to the input, and (4) *inherently interpretable* models (Chen et al., 2019; Brendel & Bethge, 2019; Böhle et al., 2021; 2022) that have been architecturally designed to be more interpretable.

In this work, we present surprising empirical evidence that shows that both 'post-hoc' and inherently interpretable explanation methods highly depend on the weights of the last classification layer, such as the linear probe for self-supervised methods.

**Evaluating Importance Attributions.** Recent work has studied important properties of such explanation methods, like faithfulness (Adebayo et al., 2018; Hooker et al., 2019; Srinivas & Fleuret, 2018; Rao et al., 2022), robustness to adversarial attacks (Ghorbani et al., 2019; Slack et al., 2020; Dombrowski et al., 2019) and fairness (Dai et al., 2022). In contrast to this, we focus on another important dimension—*the sensitivity of explanations to the model training*, which has thus far not been systematically studied. And although a dependence on the training has been reported in a few instances (Caron et al., 2021; Tsipras et al., 2019; Böhle et al., 2024), these are mostly limited to single explanation methods and pre-training paradigms. Moreover, while explanation methods are often developed and evaluated in the context of supervised models, the pre-training paradigms even for classification models have become increasingly diverse. Given the fact that many explanations are applied 'post hoc', it is important to understand whether and to what degree they yield consistent results independent of the pre-training paradigm.

To address this, we systematically study a wide range of explanation methods across a variety of pre-trained backbones and find that the results are consistent across explanation methods, suggesting that our conclusions are not an artifact of a particular explanation method or backbone combination.

**Non-linear Probes.** He et al. (2022) argue that linear probes cannot disentangle non-linear representations, and show improved accuracy by fine-tuning multiple-layers of pre-trained models. Li et al. (2023b) propose to dynamically choose the complexity of a 'readout' (i.e. probing) module to increase performance. Similarly, we also find non-linear probes to consistently improve classification accuracy. In contrast to these works, we propose using interpretable MLPs that lead to *both* improved accuracy and quality of explanations.

## 3    INTERPRETABLE PROBING OF PRE-TRAINED REPRESENTATIONS

To test the generality of our findings and isolate the impact of pre-training, we evaluate model explanations across a broad range of pre-training paradigms, with a specific focus on commonly used self-supervised representation learning methods. Linear probing of pre-trained models is a widely adopted approach to evaluate the learned representations on downstream tasks, making it very relevant to understand how to obtain effective explanations for combined models (backbone + probe).

### 3.1    SETUP

To holistically evaluate explanations methods, we utilize a diverse suite of interpretability metrics described below. Note, however, that in contrast to fully supervised models, where output neurons are optimized to represent specific classes, this is not the case for self-supervised models.

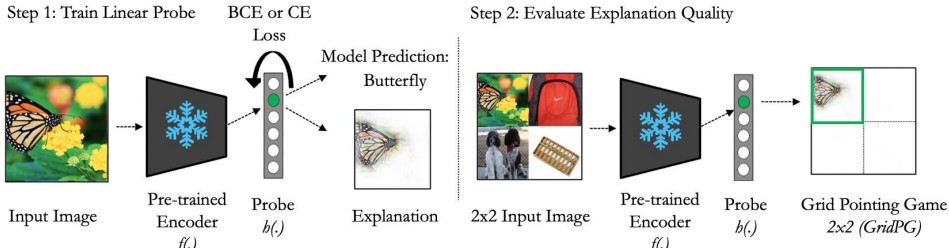

Figure 3: **Setup: Step 1.** Linear or MLP probes $h$ are trained on frozen pre-trained models $f$. **Step 2.** Explanation methods are applied to the classification predictions of the trained probes, and evaluated across a wide array of interpretability metrics to assess explanation quality (e.g. localization).

To nonetheless compare the explanations obtained from various pre-trained model backbones, we propose the following experimental setting: **(1)** we pre-train the models based on various pre-training paradigms and freeze the model parameters; **(2)** we train linear and non-linear classifiers (probes) on those frozen features for downstream image classification; **(3)** we apply attribution methods to the classification predictions of the trained probes and evaluate the quality of the generated explanations (see Figure 3).

This allows us to compare all pre-trained backbones in a standardized setting and to leverage existing evaluation metrics that were developed in the context of explaining classification models (see below). Further, by freezing the backbone features, we are able to isolate the impact of the pre-training paradigm. Finally, note that with a linear probe, the classification output is the result of simply a linear combination of the backbone representations. For most attribution methods (e.g. LRP, B-cos, LIME, IxG, and IntGrad, see Section 4.2), the resulting importance attributions, in turn, are also just a linear superposition of the attributions that would be obtained for individual neurons in the backbones' representations, and therefore a direct reflection of the backbones' interpretability itself.

**Evaluation metrics.** To assess the class-specificity of model explanations, we follow prior work and measure the fraction of positive contributions $A_i^R$ that fall within a pre-specified region $R$ of a given image vs. the total amount of positive contributions $\Sigma A_i$. The score $s_i$ for each image $i$ is thus given by $s_i = A_i^R / \Sigma A_i$. We discuss the motivation for our metric selection in Appendix A.

For single-label classification (ImageNet (Russakovsky et al., 2015)), we employ the *grid pointing game (GridPG)* (Böhle et al., 2021; 2022; Zhang et al., 2018; Samek et al., 2017). Here, the trained models are evaluated on a synthetic grid of images of distinct classes and for each of the class logits the region $R$ is given by its respective position in the synthetic image grid, see also Figure 4.

For multi-label classification (VOC (Everingham et al., 2009), COCO (Lin et al., 2014)), we rely on the bounding box annotations provided in the datasets and use the *energy pointing game (EPG)* (Wang et al., 2020). I.e., the region $R$ corresponds to the bounding boxes of the class for which the explanations are computed. The ImageNet validation set also includes bounding box annotations, which we use to additionally report the EPG score on this dataset. We report both the GridPG and EPG scores in percentages, with higher scores indicating better localization.

We further analyze the explanations using the *pixel deletion method* (Samek et al., 2017; Hedström et al., 2024), and also evaluate their *compactness* (Gini index p.p. as in (Chalasani et al., 2018)) and *complexity* (entropy as in (Tseng et al., 2020)), thus ensuring a comprehensive evaluation setting.

## 3.2 THE IMPACT OF THE PROBES' TRAINING OBJECTIVE

As discussed in the previous section, we train linear probes on the frozen, pre-trained representations, to apply common explanation methods and metrics.

Following prior work (Caron et al., 2021; Grill et al., 2020; Chen et al., 2020c;a;b; He et al., 2022), we optimize these probes via the cross-entropy (CE) loss $\mathcal{L}_{\text{CE},i}$ for image $i$, which is given by

$$\mathcal{L}_{\text{CE},i} = -\sum_c \log \frac{\exp\left(\hat{y}_{c,i}\right)}{\sum_k \exp\left(\hat{y}_{k,i}\right)} \times t_{c,i} \ . \tag{1}$$

Here, $\hat{y}_{c,i}$ denotes the probe's output logit for class $c$ and input $i$, and $t_{c,i}$ the respective one-hot encoded label. Interestingly, in our experiments we noticed that the explanations of the predicted class

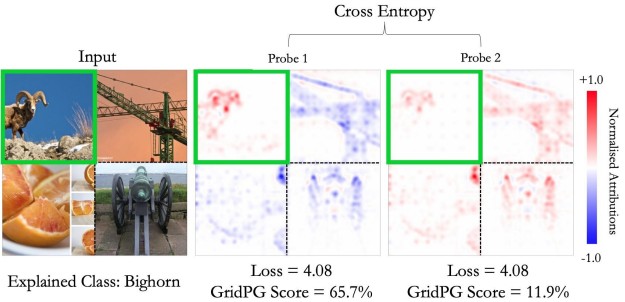

Figure 4: **Due to the shift-invariance of softmax,** one cannot expect positive and negative attributions to be well calibrated, which can lead to unintuitive model explanations, see also Equation (3). Specifically, one can easily define **equivalent** linear probes (Probe 1,2) that achieve the same CE loss, but visually dissimilar explanations and GridPG scores (65.7% vs. 11.9%). Col. 2+3 show LRP (Bach et al., 2015) attributions for two equivalent probes explaining the same class (*bighorn*).

for CE-trained linear probes were highly distributed and failed to localize the class objects effectively (see Figure 1a). We hypothesize that this could be due to the shift-invariance of the CE loss.

**Softmax Shift-Invariance Issue.** To understand this, note that the CE loss is invariant to adding a shift $\delta$ to all output logits, as long as this shift is the same for all classes (Srinivas & Fleuret, 2021); in fact, this shift can even be specific to image $i$ (see Figure 4):

$$\frac{\exp\left(\hat{y}_{c,i} + \delta_i\right)}{\sum_k \exp\left(\hat{y}_{k,i} + \delta_i\right)} = \frac{\exp\left(\hat{y}_{c,i}\right)\exp\left(\delta_i\right)}{\sum_k \exp\left(\hat{y}_{k,i}\right)\exp\left(\delta_i\right)} = \frac{\exp\left(\hat{y}_{c,i}\right)}{\sum_k \exp\left(\hat{y}_{k,i}\right)} \ . \tag{2}$$

Importantly, note that such an image-specific shift can be obtained by shifting the probes' weight vectors $\mathbf{w}_k$ for all classes $k$ by a fixed vector $\mathbf{w}'_k = \mathbf{w}_k + \Delta\mathbf{w}$:

$$\frac{\exp\left(\mathbf{w}'^T_c \mathbf{a}_i\right)}{\sum_k \exp\left(\mathbf{w}'^T_k \mathbf{a}_i\right)} = \frac{\exp\left(\mathbf{w}^T_c \mathbf{a}_i + \Delta\mathbf{w}^T \mathbf{a}_i\right)}{\sum_k \exp(\underbrace{\mathbf{w}^T_k \mathbf{a}_i}_{\hat{y}_{k,i}} + \underbrace{\Delta\mathbf{w}^T \mathbf{a}_i}_{\delta_i})} = \frac{\exp\left(\hat{y}_{c,i} + \delta_i\right)}{\sum_k \exp\left(\hat{y}_{k,i} + \delta_i\right)} \tag{3}$$

Here, $\mathbf{a}_i$ denotes the backbones' frozen input representation. As can be seen from Equations (2) and (3), there are an infinite number of linear probes which achieve the same loss and are thus indistinguishable as far as the optimization is concerned.

As most attribution methods, in some form or another, rely on the models' weights to compute the importance attributions, this reliance can have a crucial impact on the resulting explanations, as we show in Figure 4. Specifically, we show the attributions derived via layer-wise relevance propagation (Bach et al., 2015) for two **functionally equivalent probes**. While the probes give the same predictions and achieve the same loss for every input $\mathbf{x}$ by design, they yield vastly different importance attribution maps (cf. Sundararajan et al. (2017)). This, in turn, results in very different GridPG scores (65.7% vs. 11.9%).

Since both probes are equivalent under CE-based optimization, it cannot be expected that for CE-trained probes the attributions are calibrated such that 'positive' attributions will be class-specific. Thus, we also evaluate BCE-based probes, which do not exhibit the shift invariance as CE models.

**BCE Probing.** The BCE objective has recently been shown to perform well for image classification (Wightman et al., 2021; Böhle et al., 2024); in detail, the BCE loss is given by

$$\mathcal{L}_{\text{BCE},i} = -\sum_c t_{c,i} \log\left(\sigma\left(\hat{y}_{c,i}\right)\right) + (1 - t_{c,i}) \log\left(1 - \sigma\left(\hat{y}_{c,i}\right)\right) \ , \tag{4}$$

with $\sigma$ denoting the sigmoid function. Importantly, in contrast to the CE loss, the BCE loss is **not shift-invariant**. Specifically, note that for BCE, the linear probe is penalized for adding a constant positive shift to non-target classes and thus biased towards focusing on class-specific features. We therefore expect it to result in better calibrated explanations, which is indeed what we observe: specifically, as we show in Section 5.1, BCE probes exhibit a significantly higher degree of class-specificity and lend themselves better for localizing class objects. They also lead to more stable predictions under the pixel deletion evaluation, as well as more compact and less complex explanations.

### 3.3 Non-linear Probing with Interpretable MLPs

In addition to the impact of the probes' loss function on the explanations (Section 3.2), here we discuss the interplay between classifier's complexity and the resulting explanations.

In particular, we note that features computed by self-supervised (Chen et al., 2020c; Grill et al., 2020; Caron et al., 2021) or vision-language backbones (Radford et al., 2021) are not necessarily optimized to be linearly separable with respect to the classes of any arbitrary downstream task (He et al., 2022). We posit that this lack of linear separability might further diminish the localization ability of explanation methods, as different classes may share certain features in the frozen feature representations of pre-trained backbones that are not trained with full supervision. To address this, prior work (Li et al., 2023b; Hewitt & Liang, 2019) investigates using non-linear probes, which have been shown to result in improved downstream performance. However, this might come at a cost to model interpretability, as it has been shown (Rao et al., 2022) that explanation methods like GradCAM (Selvaraju et al., 2017), perform significantly worse at earlier layers than at the last layer of a DNN.

**MLP Probes.** To mitigate this, we propose to use an interpretable Multi-Layer Perceptron (MLP) probing technique to improve model accuracy *and* explanation quality. Specifically, we also train more complex probes on the frozen features, namely two-layer and three-layer conventional and B-cos (Böhle et al., 2022) MLPs and evaluate how this impacts the explanations' class-specificity.

A *conventional* fully connected MLP $\mathbf{f}(\mathbf{x}; \theta)$ with $L$ layers is given by:

$$\mathbf{f}(\mathbf{x}; \theta) = \mathbf{l}_L \circ \mathbf{l}_{L-1} \circ \ldots \circ \mathbf{l}_2 \circ \mathbf{l}_1(\mathbf{x}) \tag{5}$$

where $\mathbf{l}_j$ denotes a linear layer $j$ with parameters $\mathbf{W}_j$, and $\theta$ is the set of all parameters within the MLP. For a given input $\mathbf{a}_l$ to layer $l$, the output is computed as: $\mathbf{l}_l(\mathbf{a}_l; \mathbf{W}_l) = \phi(\mathbf{W}_l^T \mathbf{a}_l)$, with $\phi$ a non-linear activation function (e.g. ReLU).

Instead of computing their outputs as a ReLU-activated linear transformation, the B-cos layers $\mathbf{l}_l^*$ employ the B-cos transformation, which is given by:

$$\mathbf{B\text{-cos layer}} \quad \mathbf{l}_l^*(\mathbf{a}_l; \mathbf{W}_l) = |c(\mathbf{a}_l; \mathbf{W}_l)|^{\mathbf{B}-1} \odot \mathbf{W}_l \, \mathbf{a}_l \,, \tag{6}$$

Here, $\odot$ is row-wise multiplication, i.e. all rows of $\mathbf{W}_l$ are scaled by the scalar entries of the vector to its left and $c$ computes the cosine similarity between the input vector $\mathbf{a}_l$ and each row of $\mathbf{W}_l$.

Böhle et al. (2022) showed that the additional cosine factor in Equation (6) induces increased weight-input alignment during optimization, which significantly increase the localization performance of a linear summary of the B-cos models. Interestingly, we show in Section 5.2, using B-cos MLP probes on *conventional* backbones can also improve the localization of post-hoc explanations.

## 4 Experimental Scope

In the following section, we outline our experimental scope, and discuss the pre-training (Section 4.1) and explanation methods (Section 4.2) that we evaluate.

### 4.1 Pre-Training Frameworks

We aim to have a broad enough representative set that highlights how our evaluation generalizes across differently trained feature extractors, particularly (1) fully-supervised, (2) self-supervised, and (3) contrastive vision-language learning.

**(1) Fully-Supervised Learning** First, we evaluate the explanation methods on fully supervised backbones. On the one hand, backbones pre-trained in a supervised manner are still often used for transfer learning (Cheng et al., 2022; Xie et al., 2021; Chen et al., 2017). On the other hand, an evaluation of fully supervised models also provides a useful reference value, as most explanation methods have been developed in this context.

In addition to evaluating the end-to-end trained classifiers, we also evaluate linear probes on the frozen representations of these models, in order to increase the comparability to the self-supervised approaches we present in the following.

**(2) Self-Supervised Learning.** We consider three popular self-supervised pre-training frameworks: MoCov2 (Chen et al., 2020c), BYOL (Grill et al., 2020), and DINO (Caron et al., 2021).

**(3) Vision-Language Learning.** For this we use the multi-modal CLIP (Radford et al., 2021), that is pre-trained on a large-scale dataset comprising of image-text pairs.

To summarize, we evaluate across a broad spectrum of pre-training mechanisms: a *contrastive*, two *self-distillation-based*, and a *multi-modal* pre-training paradigm, which cover some of the most popular approaches to self-supervised learning. We contrast our evaluations with fully-supervised trained models (for more details refer to Appendix F.3).

## 4.2 EXPLANATION METHODS

To evaluate model interpretability, we apply explanation methods to the classification predictions of the probes trained on the frozen features (cf. Figure 3 and Section 3.1). In the following, we provide a short overview on the explanation methods.

**Input×Gradient** (Shrikumar et al., 2017) is a backpropagation-based attribution method that involves taking the element-wise product of the input and the gradient. For a given model $\mathbf{f}(\mathbf{x}; \theta)$ and input $\mathbf{x}$, it is denoted by $\mathbf{x} \odot \frac{\partial \mathbf{f}(\mathbf{x}; \theta)}{\partial \mathbf{x}}$.

**Integrated Gradients** (Sundararajan et al., 2017) follows an axiomatic method and is formulated as the integral of gradients over a straight line path from a baseline input $\mathbf{x}'$ to the given input $\mathbf{x}$. IntGrad for an input $\mathbf{x}$ is equal to $(\mathbf{x} - \mathbf{x}') \times \int_0^1 \frac{\partial \mathbf{f}(\mathbf{x}' + \alpha(\mathbf{x} - \mathbf{x}'))}{\partial \mathbf{x}} \partial \alpha$.

**Layer-wise Relevance Propagation** (Bach et al., 2015) generates attribution maps by propagating relevance scores backwards through the network, thus decomposing the model prediction into contributions from individual input features.

**GradCAM** (Selvaraju et al., 2017) is an activation based explanation method, that generates attributions corresponding to the gradient of the class logit with respect to the feature map of the last convolutional layer of a DNN.

**ScoreCAM** (Wang et al., 2020) computes a weight for each activation map based on its impact on the final prediction (instead of relying on gradients), leading to more interpretable saliency maps.

**LIME** (Ribeiro et al., 2016) samples perturbed versions of the input of interest and observes the changes in predictions. A linear model is fit to these perturbed instances to provide local explanations for a model's decision.

**B-cos** (Böhle et al., 2022) are attributions generated by the inherently interpretable B-cos networks. Essentially, the attribution map is computed by an element-wise product of the dynamic weights with the input that faithfully encapsulates the contribution of each pixel to a given class $c$: $(\mathbf{W}_c^T(\mathbf{x}) \odot \mathbf{x})$.

**Rollout** (Abnar & Zuidema, 2020) generates attribution maps by aggregating attention scores across layers in a vision transformer (ViT). This is done by recursively multiplying the attention matrices.

**CGW1** (Chefer et al., 2020) propagates explanations through self-attention layers, effectively capturing token interactions to provide an intuitive understanding of ViT-based models.

**CausalX** (Xie et al., 2022) identifies important input features by analyzing their causal effects on model predictions through interventions, allowing a more robust interpretation of ViT decisions.

**In short**, we evaluate a wide range of attribution methods, including *gradient-based, activation-based, perturbation-based* post-hoc explanations, as well as the *inherent model explanations* of the recently proposed B-cos models, for both convolutional- and vision transformer-based architectures.

For exact implementation details on datasets, models, pre-training please see Appendix F.

## 5 RESULTS

We now discuss our experimental findings. Specifically, we analyze the impact of the **probes' optimization objective** (Section 5.1) and the **probe complexity** (Section 5.2) on the model explanations.

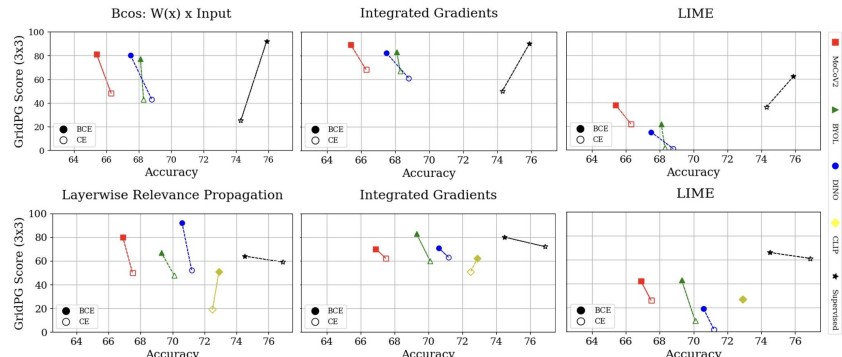

Figure 5: **BCE vs. CE — Accuracy and GridPG scores on ImageNet for ResNet-50 models.** GridPG scores improve significantly with BCE loss over CE loss, and this is consistent across pre-training paradigms for both B-cos models (**top row**) and conventional models (**bottom row**).

Table 1: **BCE vs. CE —Accuracy and GridPG scores on ImageNet for ViT models.** A consistent improvement in *localization score* for BCE over CE probes is seen.

| Backbone | Pre-training | Acc.%↑ | | Localization%↑ | | |
|---|---|---|---|---|---|---|
| | | CE | BCE | $\Delta_{\text{bce-ce}}^{\text{CausalX}}$ | $\Delta_{\text{bce-ce}}^{\text{CGW1}}$ | $\Delta_{\text{bce-ce}}^{\text{Rollout}}$ |
| ViT-B/16 | Sup. | 73.2 | 72.8 | +0.7 | +15.7 | +0.4 |
| ViT-B/16 | DINO | 77.2 | 77.6 | +5.3 | +11.7 | +4.8 |
| ViT-B/16 | MoCov3 | 76.3 | 75.1 | +3.8 | +4.4 | +3.3 |

## 5.1 IMPACT OF BCE VS. CE

Here, we evaluate how the optimization objective of linear probes impacts their accuracy and explanations, using metrics in Section 3.1. Results are for ResNet-50 (He et al., 2016), unless specified.

**Localization.** In Figure 5, we plot the linear probe accuracy versus the GridPG scores for MoCov2 (**red**), BYOL (**green**), DINO (**blue**), CLIP (**yellow**) and supervised (**black**) models for B-cos/LRP (col. 1), IntGrad (col. 2) and LIME (col. 3) on ImageNet. We do so for linear probes trained via BCE (filled markers) and CE (hollow markers) for conventional (row 2) and B-cos (row 1) models.

We find significant gains in the explanations' localization for all approaches, models, and explanations when using the BCE loss instead of the commonly used CE loss. E.g., for conventional models explained via LRP (Figure 5, col. 1, row 2), the GridPG score for CLIP improves by 32p.p. (**19%**→**51%**), MoCov2 improves by 30p.p. (**50%**→**80%**), for BYOL by 18p.p. (**48%**→**66%**), and for DINO even by 40p.p. (**52%**→**92%**).

Similarly, the GridPG score for B-cos explanations also significantly increases (Figure 5, col. 1, row 1): for MoCov2, it improves by 33p.p. (**48%**→**81%**), for BYOL by 28p.p. (**43%**→**71%**), and for DINO by 37p.p. (**43%**→**80%**).

Interestingly, I×G and GradCAM (see Appendix E) only show consistent improvements for B-cos models. For I×G, this is in line with prior work, as model gradients for conventional models are known to suffer from 'shattered gradients' (cf. (Balduzzi et al., 2017)).

Notably, the observed improvements in localization lead to significant qualitative improvements regarding the class-specificity of the explanations, see Figure 6 (left). The B-cos attributions in Figure 6 (left) for probes trained with the BCE loss (top row) are much more localized as compared to the probes trained with CE loss (bottom row). We see this behavior is consistent across all pre-training approaches; for results on additional explanation methods, please see the Appendix D.1. To also compare with supervised models qualitatively, in Figure 6 (right) we show B-cos explanations for DINO (cols. 2+3) and supervised models (cols. 4+5). We find both explanations to visually look similar, and to follow a similar trend when trained using either CE (cols. 2+4) or BCE (cols. 3+5). For BCE, the model attributions are better localized for both types of training.

In Figure 2a similar improvements are seen for the EPG metric (on ImageNet) when using BCE probes for B-cos and LRP attributions. We specifically see a greater improvement on smaller bounding boxes thus highlighting the ability for improved localization as compared to CE probes.

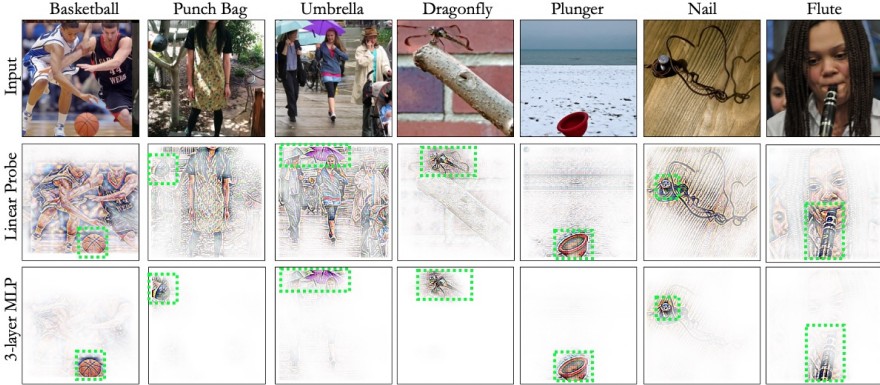

Figure 6: **(a) BCE vs. CE.** The B-cos attributions for a linear probe trained with CE loss (**bottom row**) leak into nearby cells in the 2x2 GridPG evaluation setting. The attributions for linear probes trained with a BCE loss (**top row**) are consistently much more localized. **(b) SSL vs Supervised**. The B-cos attributions for DINO (**cols. 2+3**) are visually very similar to supervised models (**cols. 4+5**), despite being optimized very differently, thus highlighting the importance of the training objective of the linear probe (see Appendix D.1 for more results).

Figure 7: **Qualitative results of MLP probes on ImageNet**. We find that explanations for B-cos MLPs trained on the DINO features exhibit better localization than a linear probe. For other pre-training and explanation methods, see Appendix D.2.

Vision Transformer specific explanation methods demonstrate similar improvements (see Table 1).

**Pixel Deletion.** Under the pixel deletion setup, in Figure 2b we observe the BCE probes to lead to more stable predictions when least important pixels are successively removed. This is consistent across majority of the pre-training methods as well as attribution methods.

**Complexity and Compactness.** Table E1 demonstrates a consistent improvement in compactness (Gini index p.p.) and reduction in complexity (entropy) for BCE vs. CE, except for I×G and Grad-CAM in the case of conventional backbones.

We thus note, that in contrast to the loss function, the choice of pre-training method has a limited impact only on explanation quality, with no particular method consistently outperforming others.

## 5.2 EFFECT OF MORE COMPLEX PROBES

In this section, we present the experimental results on the effect of training more complex classifiers on top of the frozen features (ResNet-50 unless specified). In particular, we train 2- and 3-layer MLPs and evaluate the impact on accuracy and the quality of explanations.

In Figure 8, we show the results on ImageNet and COCO for **B-cos MLPs**. It can be seen that for all pre-trained models there is not only a steady increase in accuracy but also an improvement in the localization ability (GridPG and EPG scores) of model explanations.

E.g., for standard models explained via IntGrad we observe the following improvements when going from a single linear probe to a 3-layer MLP in (accuracy, GridPG): MoCov2 (66.9, 69.0) → (**71.3, 83.0**), BYOL (69.3, 83.0) → (**72.1, 85.0**), and DINO (70.6, 70.0) → (**73.1, 89.0**). A similar trend is observed for B-cos models, and across explanation methods and datasets (see Figure 8 for results on COCO), with GradCAM being an exception (for details and discussion on this, see the Appendix E).

In Figure 9, the EPG scores (for ImageNet with bounding boxes) improve across all pre-training paradigms for **B-cos MLP** probes, which is especially prominent in smaller bounding boxes. Furthermore, Figure 7 depicts the observed localization improvements qualitatively. We find MLP

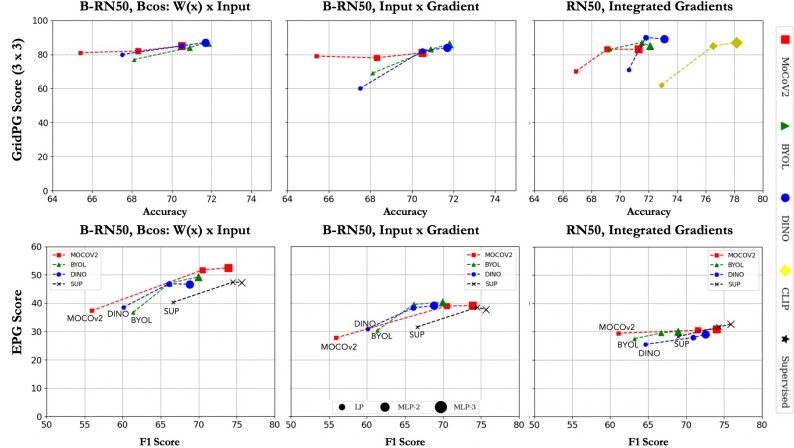

Figure 8: **Effect of more complex probes on Accuracy vs. GridPG on ImageNet (top) and F1 Score vs. EGP COCO (bottom).** A more complex B-cos probe (MLP) not only increases performance on the downstream task (**x-axis**), but interestingly also the GridPG score (**y-axis**, top) and the EPG score (**y-axis**, bottom) of the explanations (**individual plots**). This is true for B-cos models (**left 2 cols.**) and for conventional models (**right 1 cols.**) probed via B-cos MLPs.

probes to better localize the object class of interest, relying less on background. This indicates that they better capture class-specific features, which, in turn, improves their classification performance.

Interestingly, this trend in improvement in both accuracy and localization is only seen consistently for B-cos MLPs, independently of them being applied to conventional or B-cos models. While an increase in localization as measured by the EPG score is observed for conventional MLPs on COCO, on ImageNet we see a consistent *decrease* in the GridPG score (see Appendix E.2). In order to get well-localizing attribution maps on downstream tasks, we thus recommend to probe pre-trained models via B-cos probes.

**In short,** we find that training relatively lightweight B-cos MLPs ($\sim 10\%$ of entire model parameters) on frozen features of SSL-trained models with BCE loss is a versatile approach to obtain both highly interpretable and also highly performant classifiers on downstream tasks.

**Note:** Detailed results for vision transformers and other explanation methods are in Appendix C,E.

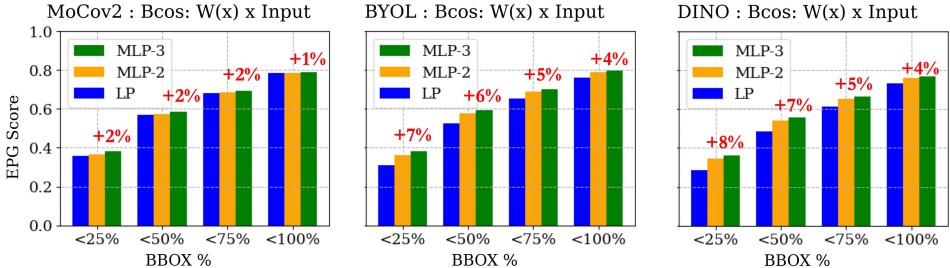

Figure 9: **Stronger probes lead to more localized attributions.** EPG scores on ImageNet.

## 6  CONCLUSION

We discover an important and surprising finding, that the quality of explanations derived from a wide range of attribution methods for pre-trained models is more dependent on how the classification layer is trained for a given downstream task, than on the choice of pre-training paradigm itself. This places important practical considerations on end-users when using 'post-hoc' attribution methods that are typically assumed to be applied independent of model training. Further, we show that employing lightweight multi-layer B-cos probes contributes to enhanced localization performance of the explanations, providing a simple and effective improvement. We support our findings through extensive experiments across several pre-training frameworks (fully-supervised, self-supervised, vision-language pre-training), and analysis on the quality of explanations for popular attribution methods on a diverse evaluation setting. Our findings are robust to the pre-training paradigms, therefore they have broad implications for DNN-based applications using XAI methods.

**Acknowledgments.** We sincerely thank Sukrut Rao and Yue Fan for their valuable feedback on the paper and insightful discussions throughout the project. Additionally, we appreciate Sukrut's help with some LaTeX sorcery. This work was partially supported by ELSA Mobility Program[1] as part of the ELLIS[2] exchange program to the Institute of Science and Technology Austria (ISTA), where a portion of this research was conducted.

**Reproducibility Statement.** We provide the complete code for pre-training, probing and evaluation of the trained models as well as for generating the quantitative and qualitative results of the explanation methods used. The code is well-documented with helper scripts to run the different parts of the pipeline and help with reproducibility. Additionally, we also make the entire pipeline available to the broader community by open-sourcing our software and provide the pre-trained model checkpoints which further helps in reproducing the results in the manuscript. We were also careful to only use open-source software, and publicly available datasets, which thus supports reproducible research. This is an effort to provide transparency, and encourage further research in the field. *Code to reproduce all experiments:* `https://github.com/sidgairo18/how-to-probe`

**Broader Impact and Ethics Statement.** The findings and discussion in our work opens up a new conversation about designing attribution / XAI methods that do take the training details of underlying models into account when getting explanations for their decisions. The fact that this also impacts inherently interpretable models, which are designed to intrinsically learn interpretable features during training, speaks more to the importance of the finding (i.e. we need to be very careful about how to handle such models and methods). The user, really needs to consider such details when leveraging model explanations.

Typically past research has focused on evaluating the interpretability of attribution methods given a fixed model, post-hoc. Yet, in the real world, we often have use-cases where one has a large pre-trained backbone over which probes are trained depending on the downstream task. In this work, we find that, surprisingly, how these probes are trained can have a significant impact on how interpretable the attributions of the model is using a fixed post hoc attribution method. Given that training such probes is usually much cheaper as compared to the backbone, our findings can be used to guide the training to yield models that are more interpretable post-hoc. Notably, we find that with this simple approach, the improvements hold across model architectures, pre-training paradigms, and even across different attribution methods.

To conclude, we see no apparent ethical concerns raised by the scientific discovery presented in our work. However, we do acknowledge the privilige and availability of resources that enable deep learning research, e.g. running large-scale training on GPUs that do result in an increased carbon footprint and efforts must be made to be more careful when using such resources.

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

# Appendix

## Table of Contents

In this appendix to our work on simple yet effective techniques for improving post-hoc explanations, we provide:

## A DISCUSSION ON THE EVALUATION METRICS AND CO-12 PROPERTIES

In this section, we provide a discussion on the importance of the interpretability metrics we have selected in our evaluations, and also place them in context with the recently proposed *Co-12 Properties* by Nauta et al. (2023).

Nauta et al. (2023) posit that explainability is a multi-dimensional concept and propose various properties that describe the different aspects of explanation quality. Specifically, they introduce these properties as the *Co-12 properties* that are crucial to be evaluated for a comprehensive assessment of explanation methods. These 12 properties are namely the following:

1. *Correctness:* denotes how faithful the explanation is with respect to the underlying 'black-box' model.

2. *Completeness:* measures the extent to which the model is described by the explanation.

3. *Consistency:* evaluates the degree of determinisim and invariance of the explanation method.

4. *Continuity:* measures continuity and generalizability of the explanation function.

5. *Contrastivity:* measures the discriminativity of the explanation with respect to different targets.

6. *Covariate Complexity:* assesses the complexity (human interpretable) of features in the explanation.

7. *Compactness:* reports the overall size of the explanation.

8. *Composition:* describes the presentation format and organization of the explanation.

9. *Confidence:* a measure of confidence of the explanation or model output.

10. *Context:* measures how relevant is the generated explanation to users.

11. *Coherence:* evaluates the plausibility of the explanation.

12. *Controllability:* measures the control or influence users have on the explanation.

In our work we present an important finding for explainable artificial intelligence (XAI), and conduct a systematic study across pre-training schemes, model architectures and heatmap based attribution methods to evaluate to what extent the training influences the explanations derived from these attribution methods. To quantitatively evaluate the quality of the explanations we assess their ability to localize class-specific features using the *grid pointing game (GPG)* Böhle et al. (2021; 2022); Zhang et al. (2018); Samek et al. (2017) and the *energy pointing game (EPG)* Wang et al. (2020).

The grid pointing game (GPG) is an established metric that reflects various of the Co-12 properties (cf. Table 3 in (Nauta et al., 2023)). First, it constitutes a "controlled synthetic data check" which allows to (approximately) deduce ground truth explanations, thus reflecting the **correctness** of the explanations. Further, as multiple targets are present in the grid images, GPG reflects the target sensitivity of the explanations and thus their **contrastivity**. By highlighting relevant regions for the decision, explanations that score highly on GPG can be useful to end users (**context**).

The energy pointing game (EPG) can be seen as a general case of the pointing game (Hooker et al., 2019), and thus subsumes all the above properties of the GPG.

We also analyze the explanations under the pixel removal method, that has been shown to be a reliable measure of faithfulness (**correctness**) of an explanation (Samek et al., 2017; Hedström et al., 2024).

Finally, to have a comprehensive evaluation we also report explanation evaluations using the Gini index (as in (Chalasani et al., 2018)) to measure the **compactness** and the entropy (as in (Tseng et al., 2020)) to measure the **complexity**.

Importantly, note that other Co-12 properties might not relate to heatmap-based explanations (**composition**, **confidence**, **controllability**) or remain unchanged by design as they are intrinsic to the explanation method itself, such as **consistency**, **completeness**, **coherence** and **continuity**.

# B   LINEAR PROBING ON FROZEN BACKBONE FEATURES

**Interpretability of frozen backbone**: Linear probing on the frozen pre-trained backbone features is an important step when using pre-trained models for downstream tasks and can also inform us about the interpretability of the backbone itself as we explain further. Since the linear probes themselves have limited modeling capacity, these explanations must therefore necessarily reflect the 'knowledge' of the backbone model. The different probes (CE / BCE), compute their outputs by nothing but a weighted mean of the frozen backbone features.

It can be demonstrated mathematically by the following: let $\mathbf{z} \in \mathbb{R}^D$ represent the output feature vector from the frozen backbone after the global average pooling operation, where the vector has $D$ dimensions. Let $\mathbf{W} \in \mathbb{R}^{C \times D}$ be the weight matrix of the fully connected classification layer, where $C$ is the number of output classes (e.g., 1000 for ImageNet), and let $\mathbf{b} \in \mathbb{R}^C$ be the bias term of the classification layer. The output prediction vector $\mathbf{y} \in \mathbb{R}^C$ is then given by:

$$\mathbf{y} = \mathbf{W}\mathbf{z} + \mathbf{b}$$

Here, $\mathbf{W}\mathbf{z}$ represents a linear combination of the backbone features $\mathbf{z}$, with the weight matrix $\mathbf{W}$. The bias term $\mathbf{b}$ is added to each class output.

Since the feature extractor is frozen (i.e., its weights are not updated during training), the classification layer performs a simple linear combination of the extracted features, and only the weights $\mathbf{W}$ and bias $\mathbf{b}$ are learned during training. Thus, essentially when analyzing the explanations generated by the model, it is largely dominated by the backbone features. This is what makes our finding surprising, where **BCE trained probes lend themselves to generate better quality explanations.**

# C ADDITIONAL ARCHITECTURES: VISION TRANSFORMERS

In this section, we provide quantitative results on Vision Transformers (ViTs, Kolesnikov et al. (2021)) and the explanation (attribution) methods developed specifically for ViTs: (1) CGW1 (Chefer et al., 2020), (2) ViT-CX (CausalX (Xie et al., 2022)), (3) Rollout (Abnar & Zuidema, 2020), and (4) B-cos (Böhle et al., 2024).

In particular, we first evaluate the impact of the training objective on probing in terms of accuracy and explanation quality using the metrics discussed in Section 3.1 on ImageNet. Next we analyze the effect of probe complexity on the explanation localization.

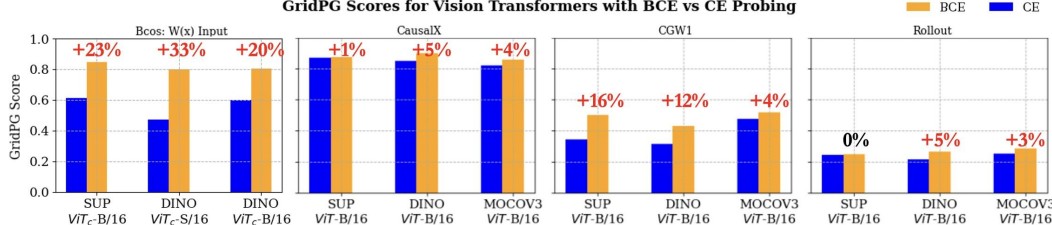

Figure C1: **ViT backbones BCE vs. CE Probing—GridPG scores on ImageNet.** GridPG for (1) B-cos (Böhle et al., 2024), (2) CausalX (Xie et al., 2022), (3) CGW1 (Chefer et al., 2020), and (4) Rollout (Abnar & Zuidema, 2020). BCE trained probes consistently outperform CE trained probes across all pre-training paradigms and explanation method.

## C.1 IMPACT OF BCE VS. CE

In the following, we evaluate the impact of the training objective of the linear probes on ViTs similar to the evaluations for CNNs as described in Section 5.1 on both supervised (Kolesnikov et al., 2021) and self-supervised (He et al. (2019), Caron et al. (2021)) pre-trained backbones.

**GridPG Localization.** In Figure C1, we plot the GridPG scores for ViT backbones when probed with the BCE (orange) vs. CE (blue) training objective for both conventional and B-cos models. We find significant gains in the explanations' localization for all approaches, models, and explanations when using the BCE loss instead of the CE loss. E.g., for conventional backbones when explained via CGW1 (Figure C1, col. 3), the GridPG score for DINO improves by 12p.p (**31%**→**43%**), MoCov3 improves by 4p.p. (**48%**→**52%**), and for supervised by 16p.p. (**34%**→**50%**). For B-cos explanations, the increase in GridPG score is more drastic (Figure C1, col. 1): for DINO, it improves by upto 33p.p (**47%**→**80%**), and for supervised by 23p.p (**61%**→**84%**).

In Table C1, we also report the accuracies and when probing with different training objectives and observe that BCE probes achieve similar performance as CE probes while achieving significantly greater localization scores.

Table C1: **ViT backbones BCE vs. CE Probing—Accuracy and GridPG scores on ImageNet.** A consistent improvement in *localization score* for BCE over CE probes is seen for both conventional ViTs (Kolesnikov et al., 2021) and inherently interpretable B-cos B-ViTs (Böhle et al., 2024) for supervised and self-supervised pre-training of the backbones with comparable accuracies.

| Backbone | Pre-training | Acc.%↑ | | Localization%↑ | | | |
| | | CE | BCE | $\Delta_{\text{bce-ce}}^{\text{Bcos}}$ | $\Delta_{\text{bce-ce}}^{\text{CausalX}}$ | $\Delta_{\text{bce-ce}}^{\text{CGW1}}$ | $\Delta_{\text{bce-ce}}^{\text{Rollout}}$ |
|---|---|---|---|---|---|---|---|
| ViT-B/16 | Sup. | 73.2 | 72.8 | – | +0.7 | +15.7 | +0.4 |
| ViT-B/16 | DINO | 77.2 | 77.6 | – | +5.3 | +11.7 | +4.8 |
| ViT-B/16 | MoCov3 | 76.3 | 75.1 | – | +3.8 | +4.4 | +3.3 |
| B-ViT$_c$-B/16 | Sup. | 77.3 | 78.1 | +23.5 | – | – | – |
| B-ViT$_c$-S/16 | DINO | 73.2 | 73.4 | +32.7 | – | – | – |
| B-ViT$_c$-B/16 | DINO | 77.1 | 77.3 | +20.4 | – | – | – |

Table C2: **ViT backbones BCE vs. CE Probing—Accuracy and EPG scores on ImageNet.** We see improvement in *localization score* for BCE over CE probes for a majority of cases across Supervised and self-Supervised pre-training of the backbones and explanation methods with comparable accuracies. *Note: We see a greater improvement in localization scores for smaller bounding boxes (with size $0 - 50\%$ of image area).*

| | | | | | Localization % | | | |
|---|---|---|---|---|---|---|---|---|
| Backbone | XAI Method | Pre-training | Loss | Acc.% | BBox size $<25\%$ | BBox size $<50\%$ | BBox size $<75\%$ | BBox size $<100\%$ |
| ViT-B/16 | CGW1 | Sup | CE | 73.2 | 26.4 | 42.8 | 53.7 | 71.0 |
| ViT-B/16 | CGW1 | Sup | BCE | 72.8 | $28.2_{(+1.8)}$ | $44.5_{(+1.7)}$ | $55.2_{(+1.5)}$ | $72.1_{+(1.1)}$ |
| ViT-B/16 | CGW1 | DINO | CE | 77.2 | 40.2 | 51.4 | 61.5 | 77.5 |
| ViT-B/16 | CGW1 | DINO | BCE | 77.6 | $41.7_{(+1.5)}$ | $52.5_{(+1.1)}$ | $62.2_{(+0.7)}$ | $77.7_{(+0.2)}$ |
| ViT-B/16 | CGW1 | MoCov3 | CE | 76.3 | 34.2 | 50.5 | 59.5 | 76.1 |
| ViT-B/16 | CGW1 | MoCov3 | BCE | 75.1 | $36.5_{(+2.3)}$ | $52.2_{(+1.7)}$ | $60.1_{(+0.6)}$ | $76.5_{(+0.4)}$ |
| ViT-B/16 | Rollout | Sup | CE | 73.2 | 28.6 | 42.8 | 53.1 | 70.5 |
| ViT-B/16 | Rollout | Sup | BCE | 72.8 | $30.1_{(+1.5)}$ | $43.9_{(+1.1)}$ | $53.5_{(+0.4)}$ | $71.1_{(+0.6)}$ |
| ViT-B/16 | Rollout | DINO | CE | 77.2 | 41.1 | 53.3 | 61.1 | 76.8 |
| ViT-B/16 | Rollout | DINO | BCE | 77.6 | $42.0_{(+0.9)}$ | $52.6_{(-0.7)}$ | $60.4_{(-0.7)}$ | $76.5_{(+0.3)}$ |
| ViT-B/16 | Rollout | MoCov3 | CE | 76.3 | 36.5 | 50.6 | 57.9 | 75.1 |
| ViT-B/16 | Rollout | MoCov3 | BCE | 75.1 | $36.8_{(+0.3)}$ | $51.2_{(+0.6)}$ | $58.5_{(+0.5)}$ | $75.5_{(+0.4)}$ |
| ViT-B/16 | CausalX | Sup | CE | 73.2 | 14.6 | 28.6 | 42.2 | 61.2 |
| ViT-B/16 | CausalX | Sup | BCE | 72.8 | $14.6_{(0.0)}$ | $29.1_{(+0.5)}$ | $41.6_{(-0.6)}$ | $61.4_{(+0.2)}$ |
| ViT-B/16 | CausalX | DINO | CE | 77.2 | 14.5 | 28.4 | 42.3 | 62.2 |
| ViT-B/16 | CausalX | DINO | BCE | 77.6 | $15.2_{(+0.7)}$ | $29.3_{(+0.9)}$ | $43.1_{(+0.8)}$ | $63.1_{(+0.9)}$ |
| ViT-B/16 | CausalX | MoCov3 | CE | 76.3 | 14.2 | 28.9 | 43.3 | 63.2 |
| ViT-B/16 | CausalX | MoCov3 | BCE | 75.1 | $15.4_{(+1.2)}$ | $30.1_{(+1.1)}$ | $44.2_{(+0.9)}$ | $63.3_{(+0.1)}$ |

Table C3: **ViT backbones MLP Probing—Accuracy and EPG scores on ImageNet.** A consistent improvement in *accuracy* $\Delta_{acc}$ and *localization score* $\Delta_{loc}$ is seen for B-cos explanations when using more complex 2 or 3 layer **B-cos MLPs** over a linear probe for DINO pre-trained small and base ViT$_c$s. This demonstrates that more powerful classifier heads are able to distill 'class-specific' information better.

| Backbone | Classifier | Acc.% | $\Delta_{acc}$ | Loc.% | $\Delta_{loc}$ |
|---|---|---|---|---|---|
| B-ViT$_c$-S/16 | Linear Probe | 73.4 | – | 79.9 | – |
| B-ViT$_c$-S/16 | Bcos-MLP-2 | 74.2 | **+0.8** | 80.4 | **+0.5** |
| B-ViT$_c$-S/16 | Bcos-MLP-3 | 74.7 | **+1.3** | 82.4 | **+2.5** |
| B-ViT$_c$-B/16 | Linear Probe | 77.3 | – | 80.3 | – |
| B-ViT$_c$-B/16 | Bcos-MLP-2 | 78.2 | **+0.9** | 82.1 | **+1.8** |
| B-ViT$_c$-B/16 | Bcos-MLP-3 | 79.7 | **+2.4** | 83.4 | **+3.1** |

**EPG Localization.** In Table C2 and Figure C2, similar improvements are seen for the EPG metric (on ImageNet) when using using BCE probes for CGW1 (Chefer et al., 2020), Rollout (Abnar & Zuidema, 2020) and CausalX (Xie et al., 2022) attributions. We specifically see a greater improvement on smaller bounding boxes thus highlighting the ability for improved localization as compared to CE probes.

## C.2 IMPACT OF COMPLEX PROBES

In Table C3, we show the results for probing DINO B-ViT$_c$-S (small) and B-ViT$_c$-B (base) with **B-cos MLPs**. It is seen that for these models there is an increase in both accuracy $\Delta_{acc}$ as well as the localization ability $\Delta_{loc}$ (GridPG scores) of the B-cos explanations. This does seem to suggest that using more complex probes helps distill 'class-specific' information from the frozen backbone features thus leading to improved localization scores even for transformer based architectures.

*These results are consistent with the improvements seen for convolutional backbones (see Section 5) and demonstrate the robustness of our findings across different model architectures (vision transformers and convolutional networks).*

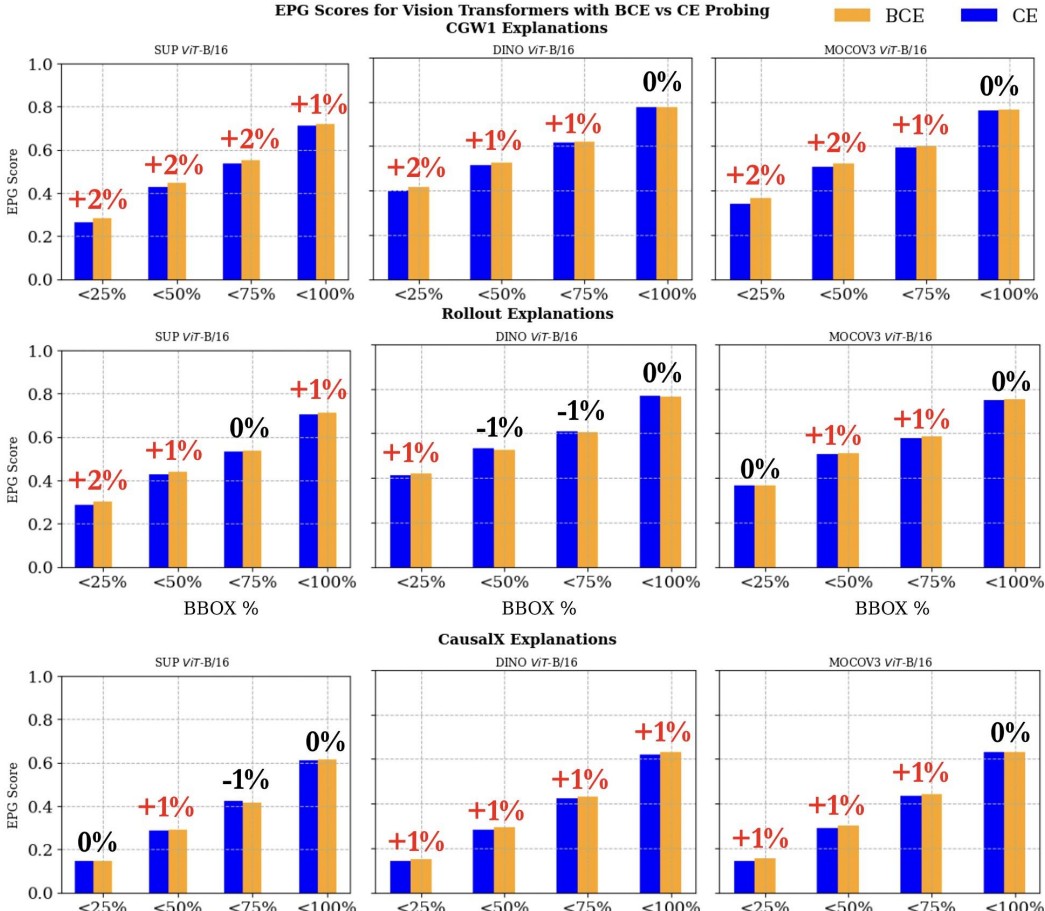

Figure C2: **ViT backbones BCE vs. CE Probing—EPG scores on ImageNet.** EPG for (1) CGW1 (Chefer et al., 2020), (2) Rollout (Abnar & Zuidema, 2020), and (3) CausalX (Xie et al., 2022). BCE trained probes outperform CE trained probes for a majority of cases; with larger and more consistent gains for smaller bounding boxes (occupying $0 - 50\%$ of the image area).

.

# D    ADDITIONAL QUALITATIVE RESULTS

In this section we show additional qualitative results for the explanations of both conventional as well as B-cos models under the different evaluation settings studied in the main paper. We first qualitatively show the impact of training the probes with *binary cross entropy* (BCE) vs. *cross entropy* (CE) loss in Sec. D.1. Then we show the effect of more complex probes on localization in Sec. D.2. Additionally, in Sec. D.3 we also provide qualitative comparisons between different explanation methods for all SSL models as well as fully supervised models. Finally, in Sec. D.4 we add more qualitative results showing a diverse set of samples for all self-supervised pre-trainings.[3]

## D.1    IMPACT OF BCE VS. CE

The probing strategy can have a significant influence on the localization ability of model explanations. As described in Sec. 5.2 of the main paper, probes trained with BCE loss localize more strongly as compared to probes trained with CE loss. Figures D1, D2, D3 (cf. Figure 6 in main paper), show the B-cos (Böhle et al., 2022) explanations in the $2 \times 2$ GridPG evaluation setting for DINO, BYOL and MoCov2 respectively when probed with BCE or CE loss. It can be observed that for all SSL-frameworks as well as supervised training, the explanations for linear probes trained with BCE loss (**cols. 3+5**) are much more localized as compared to probing with CE loss. Interestingly, however, while the focus region seems to be very similar between the SSL-trained and the fully supervised models, the B-cos explanations of BYOL and MoCov2 show lower saturation.

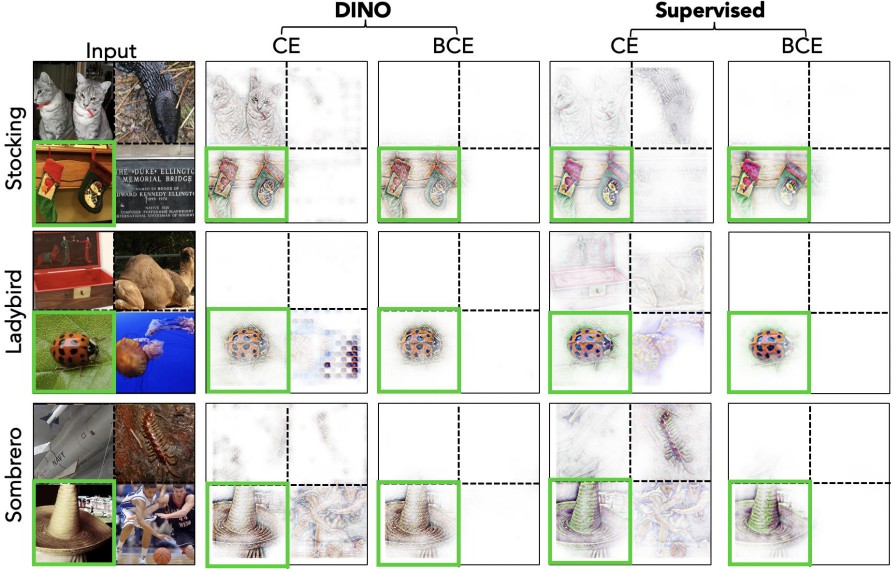

Figure D1: **DINO vs. Supervised Explanations**. The B-cos explanations for DINO (**cols. 2+3**) are visually very similar to supervised models (**cols. 4+5**), despite being optimized very differently. We also see that for B-cos models the improvements in attribution localization for probes trained with BCE loss are consistent for both DINO and supervised models.

---

[3]Results are for ResNet-50 backbones, unless specified otherwise.

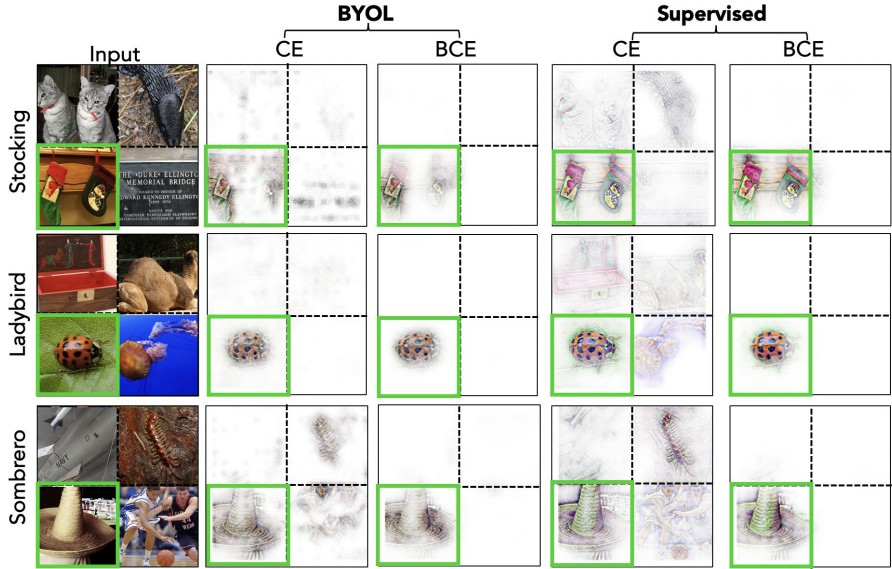

Figure D2: **BYOL vs. Supervised Explanations**. Similar to Figure D1 the B-cos explanations for models trained via BYOL (**cols. 2+3**) exhibit significant improvements in localization when trained via BCE. Interestingly, compared to the explanations of supervised models (**cols. 4+5**), we find BYOL explanations to exhibit less saturation.

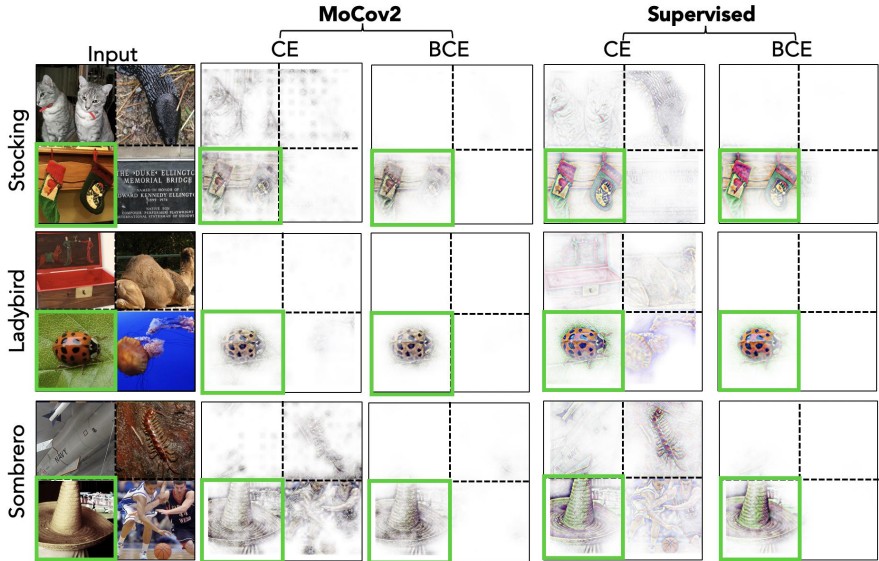

Figure D3: **MoCov2 vs. Supervised Explanations**. Similar to Figure D1, D2 the B-cos explanations for models trained via MoCov2 (**cols. 2+3**) exhibit significant improvements in localization when trained via BCE. Interestingly, compared to the explanations of supervised models (**cols. 4+5**), we find MoCov2 explanations to exhibit less saturation.

In Figure D4, we additionally visualize the LRP (Bach et al., 2015) explanations for conventional models. Similar to the B-cos models in the preceding figures, we find the explanations for BCE-trained models to exhibit significantly higher localization. Interestingly, despite the consistent improvements in localization, we again observe clear qualitative differences between the differently trained backbones: e.g., the explanations for the DINO model seem to cover the full object, whereas the explanations for the models trained via BYOL and MoCov2 are much sparser. See also Figure D5 for LRP visualizations for the CLIP (Radford et al., 2021) (vision-language) model.

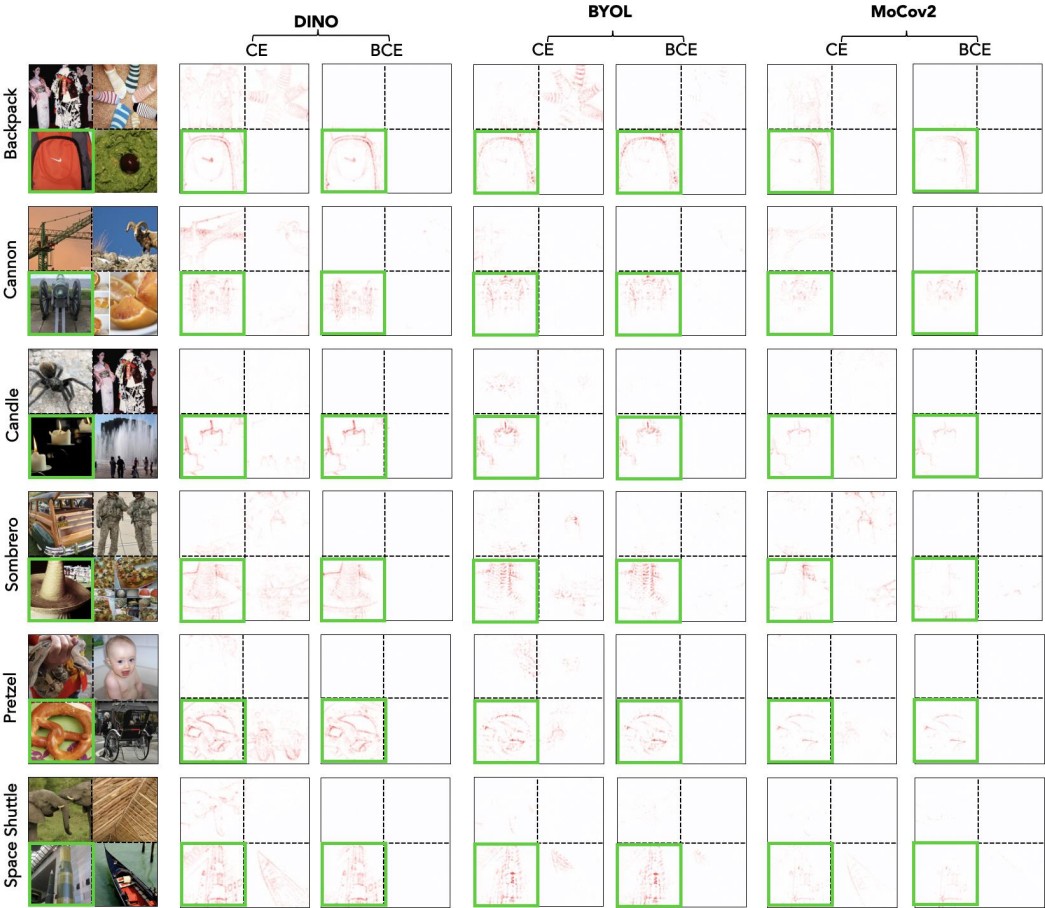

Figure D4: **Layer-wise Relevance Propagation Explanations**. Even for conventional models we see a consistent improvement in attribution localization for probes trained with BCE loss. This is seen across all SSL methods (DINO, BYOL and MoCov2). We also observe significant qualitative differences between differently trained backbones: e.g., the explanations for DINO model (**left**) cover the full object, as compared to explanations for models trained via BYOL (**center**) or MoCov2 (**right**), that are much sparser.

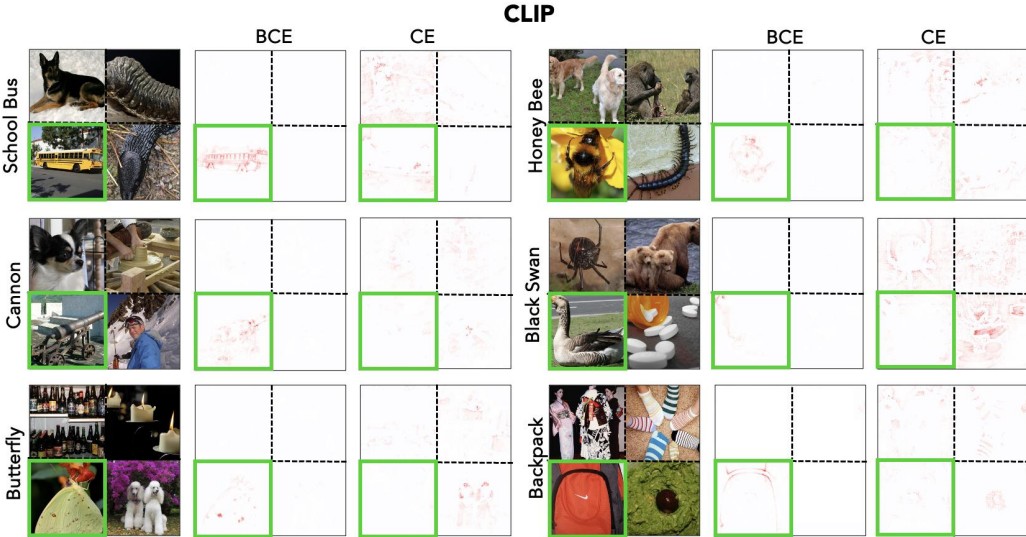

Figure D5: **Layer-wise Relevance Propagation Explanations for CLIP**. Similar to SSL and supervised pre-trained models, for CLIP (Radford et al., 2021) which is a vision-language pre-trained model we see BCE-trained probes (cols. 2+5) to localize better as compared to CE-trained (cols. 3+6) probes.

## D.2 EFFECT OF COMPLEX PROBES

Figure D6 shows visual results depicting the impact of training with MLP probes. Notice, e.g., in **cols. 3+4**, two- and three-layer MLPs better localize the object class of interest with minimal reliance on background features for DINO model. Since the features learned by SSL models may not always be linearly separable with respect to the classes present in different downstream tasks, stronger MLP classifiers are able to distill class-specific features better, leading to an improvement in localization scores and also improved performance on the downstream task. This behaviour is consistent across all SSL frameworks when we go from single probes to MLP probes. As seen previously, the B-cos explanations for BYOL and MoCov2 show lower saturation as compared to DINO.

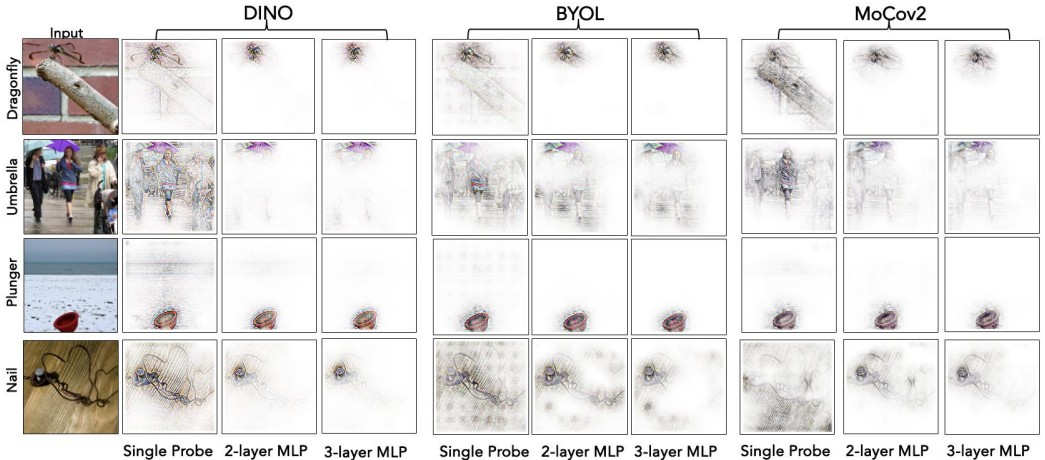

Figure D6: **Qualitative results of MLP probes on ImageNet**. We find explanations for B-cos MLPs **(cols. 3+4+6+7+9+10)** trained on SSL pre-trained features to exhibit better localization than a single linear probe **(cols. 2+5+8)**. This behavior is seen more prominently for DINO **(left)** as compared to BYOL **(center)** and MoCov2 **(right)**.

## D.3 ADDITIONAL EXPLANATION METHODS

**Comparison between explanation methods.** Figures D7, D8, D9 show additional comparisons between different attribution methods for each SSL model (both B-cos and conventional) and how it compares to their fully-supervised counterparts; in particular we show results for B-cos (Böhle et al., 2022), Layer-wise Relevance Propagation (LRP) (Bach et al., 2015), GradCAM (Selvaraju et al., 2017), Integrated Gradients (IntGrad) (Sundararajan et al., 2017), and Input×Gradient (I×G) (Shrikumar et al., 2017). Notice, every pair of consecutive rows illustrates this for a single SSL method and supervised training. We observe that visually the explanations for SSL models and supervised models are quite similar. Interestingly, this is more consistent for B-cos models as compared to conventional models. *Note: For all SSL models, these are explanations for the setting when the frozen backbone features are trained using only a single linear probe with BCE loss.*

Figure D7: Qualitative comparison of different explanation methods for **DINO** and **supervised** training for (a) B-cos and (b) conventional models. Notice that B-cos explanations are able to highlight the object of interest quite well. Also LRP explanations for conventional models seem to localize well to object features however are quite sparse. GradCAM explanations although do focus on the object but are more spread out for DINO model as compared to supervised model. IntGrad and I×G explanations are scattered across the entire image and also highlight background regions.

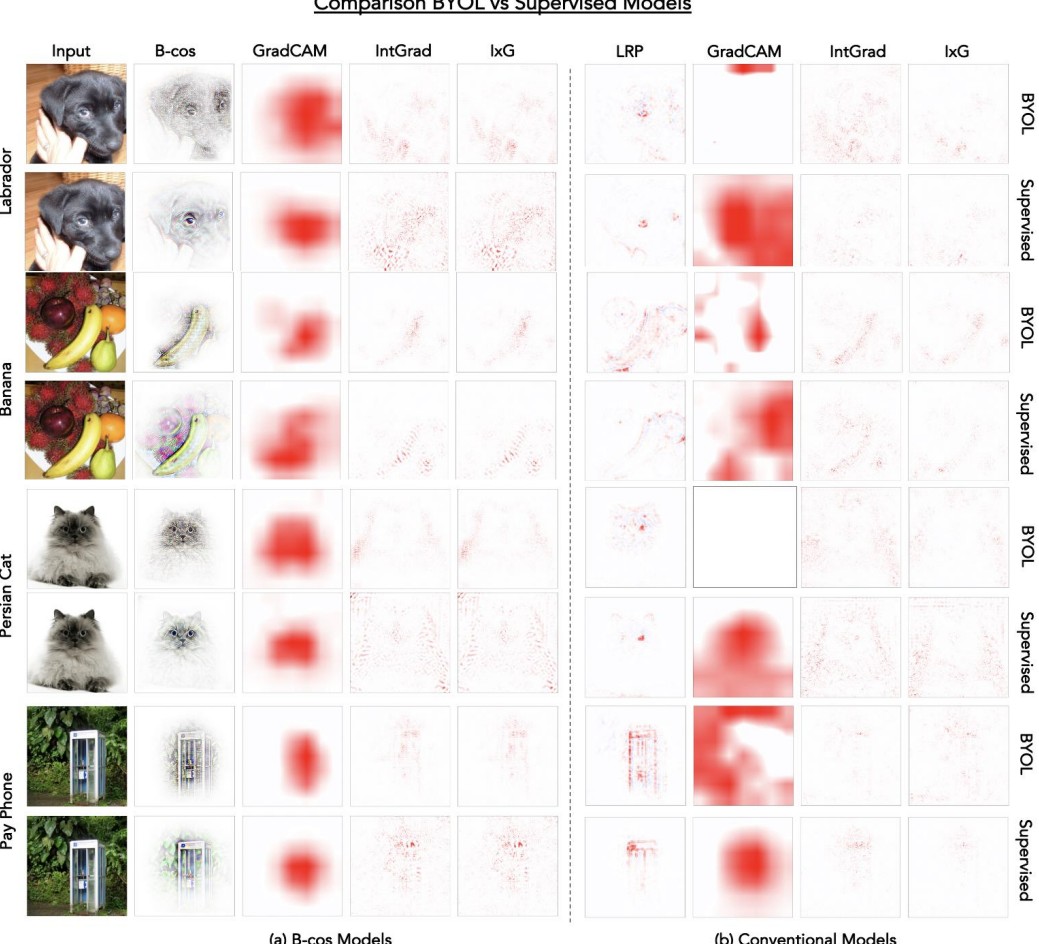

Figure D8: Qualitative comparison of different explanation methods for **BYOL** and **supervised** training for (a) B-cos and (b) conventional models. Similar to Figure D7, we observe that B-cos explanations are able to highlight the object of interest quite well. The LRP explanations for conventional models seem to localize well to object features however are quite sparse. GradCAM explanations although do focus on the object but are more spread out for BYOL model as compared to supervised model (especially for conventional models). IntGrad and I×G explanations are scattered across the entire image and also highlight background regions.

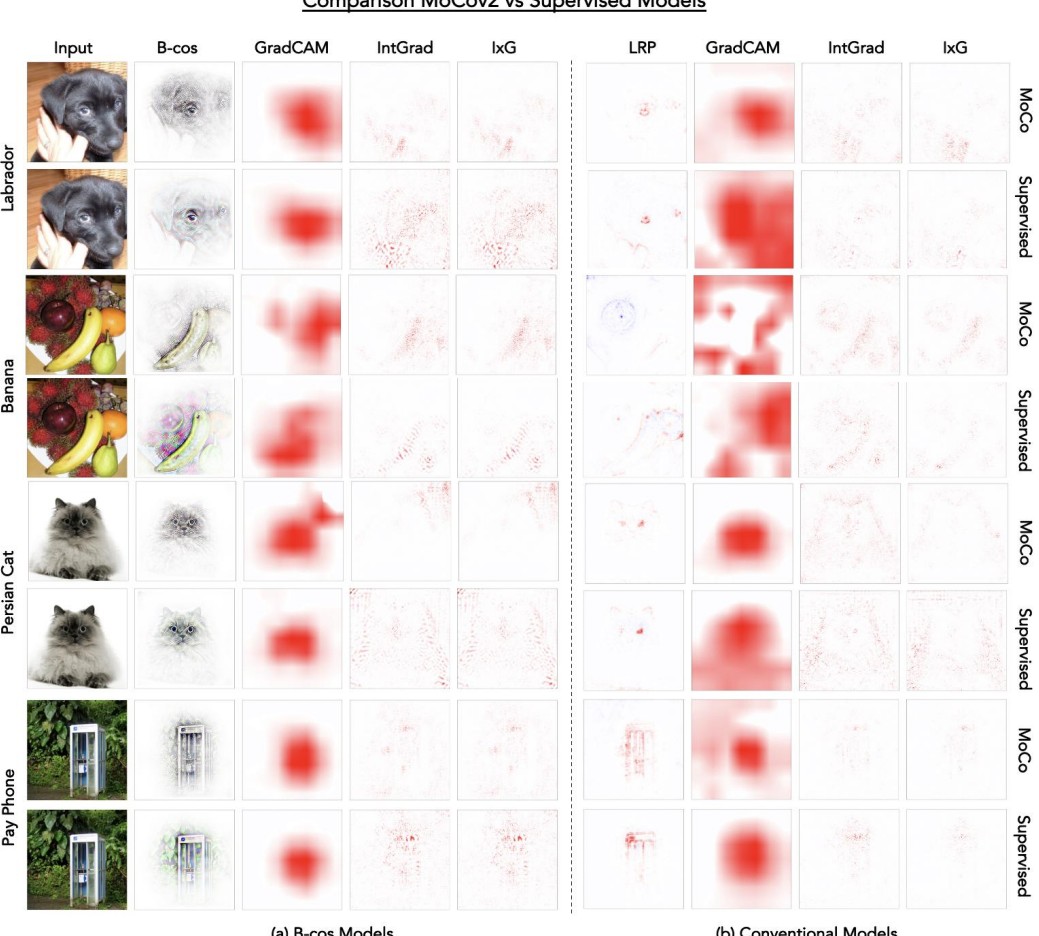

Figure D9: Qualitative comparison of different explanation methods for **MoCov2** and **supervised** training for (a) B-cos and (b) conventional models. Similar to Figures D7 (DINO), D8 (BYOL) we observe that B-cos explanations for MoCov2 are able to highlight the object of interest quite well. The LRP explanations for conventional models seem to localize well to object features however are quite sparse. GradCAM explanations although do focus on the object but are more spread out for MoCov2 model as compared to supervised model (especially for conventional models). IntGrad and I×G explanations are scattered across the entire image and also highlight background regions.

## D.4 Additional Qualitative Results

In this sub-section we add more qualitative results for B-cos, ScoreCAM, and GradCAM explanations to demonstrate the impact of the training objective and probe complexity on model explanations.

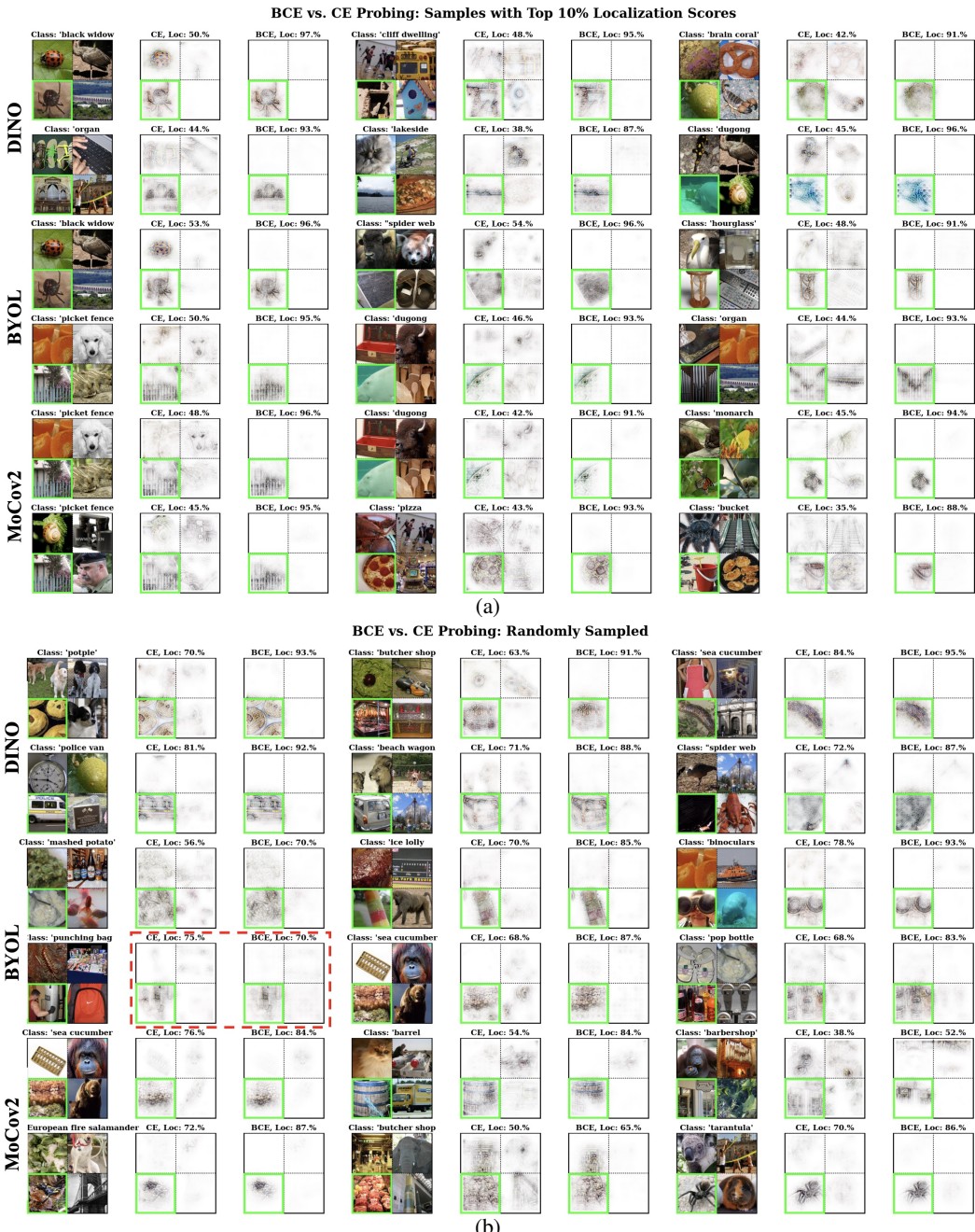

Figure D10: **Qualitative results for BCE vs. CE probing on ImageNet.** (a) Shows six examples each for DINO, BYOL, and MoCov2 sampled from the set of top $10\%$ of images to show greatest improvement of BCE probing over CE probing ($\Delta_{bce-ce}$ GridPG scores) when explained with B-cos explanations. (b) Additionaly, shows a set of six images *randomly sampled* for each SSL method; out of 18 samples we find only 1 sample where the BCE probe localizes worse than the CE probe (highlighted with a red box). *Please zoom-in to notice the finer differences in the visual explanations.*

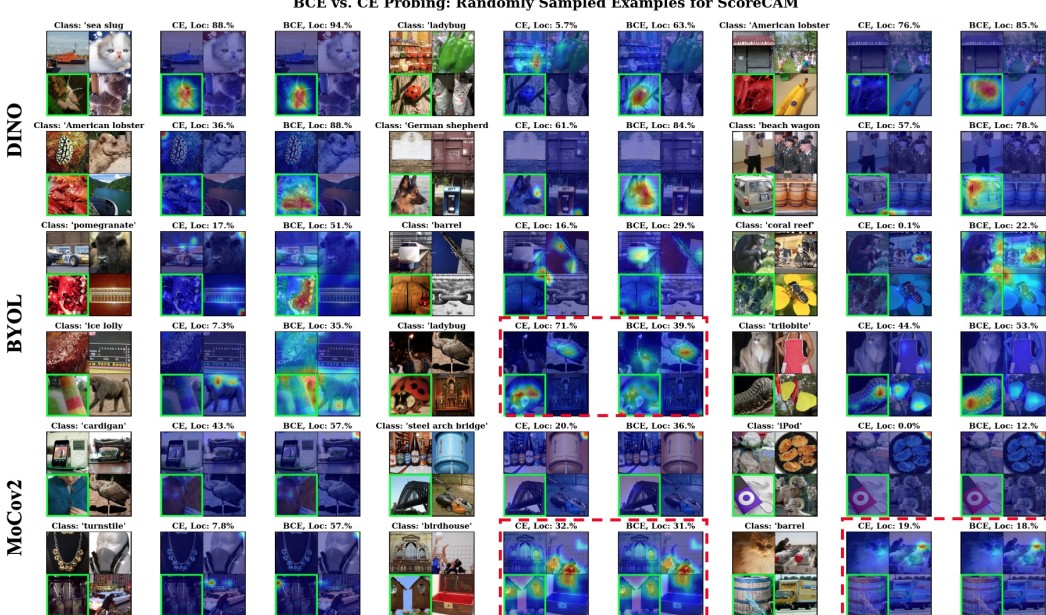

Figure D11: **Qualitative results for BCE vs. CE probing on ImageNet for ScoreCAM**. The figure shows a set of six examples sampled randomly for each SSL method, i.e. DINO, BYOL and MOCO when explained using ScoreCAM explanations. Overall BCE probes lead to more localized explanations over CE probes. We highlight examples where BCE probes perform worse than CE probes with a red box. *Please zoom-in to notice the finer differences in the visual explanations.*

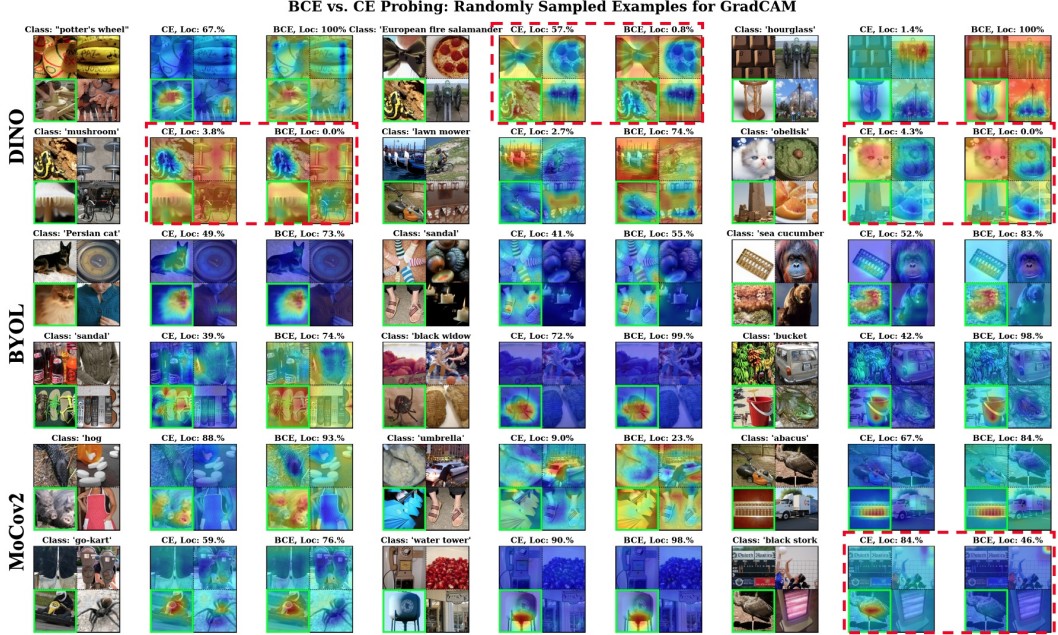

Figure D12: **Qualitative results for BCE vs. CE probing on ImageNet for GradCAM**. The figure shows a set of six examples sampled randomly for each SSL method, i.e. DINO, BYOL and MOCO when explained using GradCAM explanations. Overall BCE probes lead to more localized explanations over CE probes. We highlight examples where BCE probes perform worse than CE probes with a red box. *Please zoom-in to notice the finer differences in the visual explanations.*

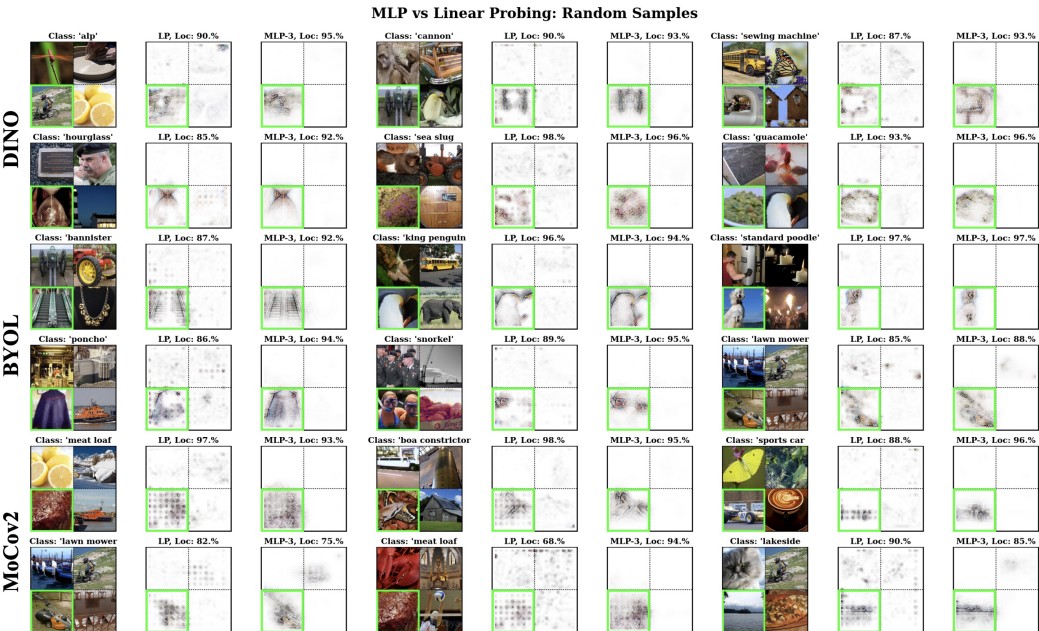

Figure D13: **More qualitative results for MLP Probing**. The figure shows a set of six examples sampled randomly for each SSL method, i.e. DINO, BYOL and MOCO when explained using B-cos explanations. MLP probes on average tend to get visually more localized samples as compared to a linear probe. *Please zoom-in to notice the finer differences in the visual explanations.*

# E ADDITIONAL QUANTITATIVE RESULTS

In this section, for completeness we present quantitative results for all explanation (or attribution) methods; specifically for B-cos (Böhle et al., 2022), Layer-wise Relevance Propagation (LRP) (Bach et al., 2015), GradCAM (Selvaraju et al., 2017), Integrated Gradients (IntGrad) (Sundararajan et al., 2017), Input×Gradient (I×G) (Shrikumar et al., 2017), LIME (Ribeiro et al., 2016) and Guided-Backpropagation (Springenberg et al., 2015). In Sec. E.1 we first evaluate the impact of the optimization objective on probing in terms of accuracy and explanation quality using the different metrics (see Section 3.1 in main paper) on ImageNet. Then in Sec. E.2 we analyze the effect of probe complexity on explanation localization. Finally in the second part of Sec. E.2, we also present the results on the multi-label classification setting on COCO and VOC datasets, where the explanation localization metric used is EPG score[4].

## E.1 IMPACT OF BCE VS. CE

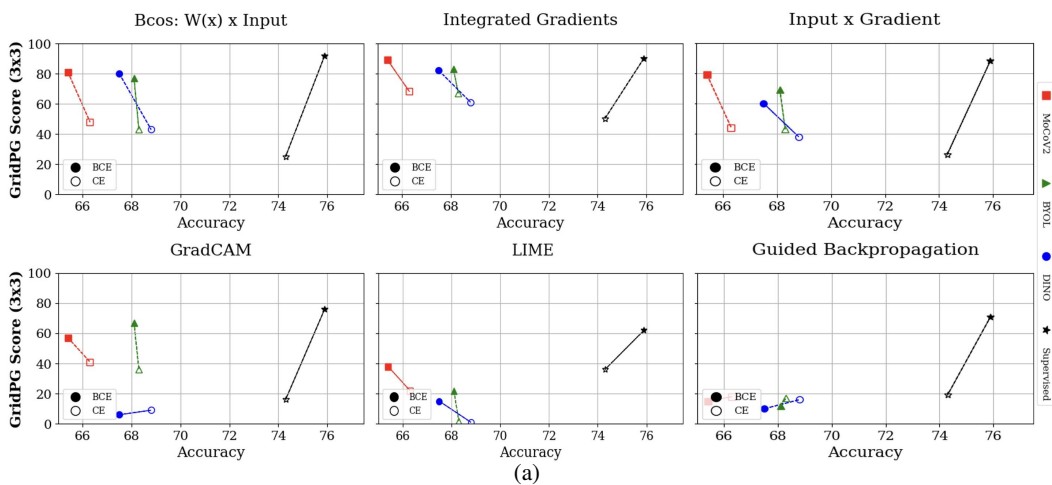

(a)

Figure E1: **BCE vs. CE — Accuracy and GridPG scores on ImageNet.** To study the impact of the optimization objective on probing we plot the accuracy vs localization (GridPG) score for **B-cos models** (B-RN50). For BCE trained probes (solid markers), a steady improvement in the localization score is seen for all SSL methods and across most explanation methods on ImageNet. For GradCAM (bottom left), DINO model does not show an improvement in localization score for BCE trained probe, and Guided Backpropagation explanations (bottom right) are inconsistent for SSL models.

In fig. E1 we plot the linear probe accuracies vs GridPG score for MoCov2 (**red**), BYOL (**green**), DINO (**blue**), CLIP (**yellow**) and supervised (**black**) models for (a) B-cos models (**B-RN50**) and (b) conventional models (**RN50**) on ImageNet. The linear probes trained with BCE loss are depicted with filled markers, and the hollow markers represent CE loss. We see consistent improvement in localization score for most explanation methods. **Note**, as discussed in the main paper I×G and GradCAM do not show consistent behaviour and only show consistent improvements for B-cos models (compare the **bottom rows** in fig. E1 (a) vs fig. E1 (b)). For I×G, this is in line with prior work, as model gradients for conventional models are known to suffer from 'shattered gradients' (cf. (Balduzzi et al., 2017)) and provide highly noisy attributions. Additionally, Guided Backpropagation also performs poorly overall to see any obvious trend (as it has been shown earlier to produce noisy explanations (Rao et al., 2022)). Finally, for GradCAM it has been shown that especially for self-supervised models, it can perform poorly due to focusing on background regions rather than object of interest (this can be induced due to the SSL specific training objectives) (Selvaraju et al., 2021).

Figures E2, E3 show the EPG localization scores on the bounding boxes provided for the ImageNet validation set. For B-cos backbones we observe a consistent improvement for BCE trained probes for 11 out of 12 cases. The only odd case is GradCAM explanations for the DINO model. For

---

[4]Results are for ResNet50 backbones, unless specified otherwise.

conventional backbones this is consistent for 9 out of 12 cases (GradCAM explanations being the exception again).

Next figures E4, E5 show the pixel deletion plots on the ImageNet validation set. For B-cos backbones we observe a more consistent improvement in terms of stability when compared to conventional models. For B-cos backbones for GradCAM method, CE probes produce more stability, and for conventional backbones we only see consistent results for LRP attribution methods (where BCE probes are more stable than CE probes). For other attribution methods and pre-trained backbones we see mixed results.

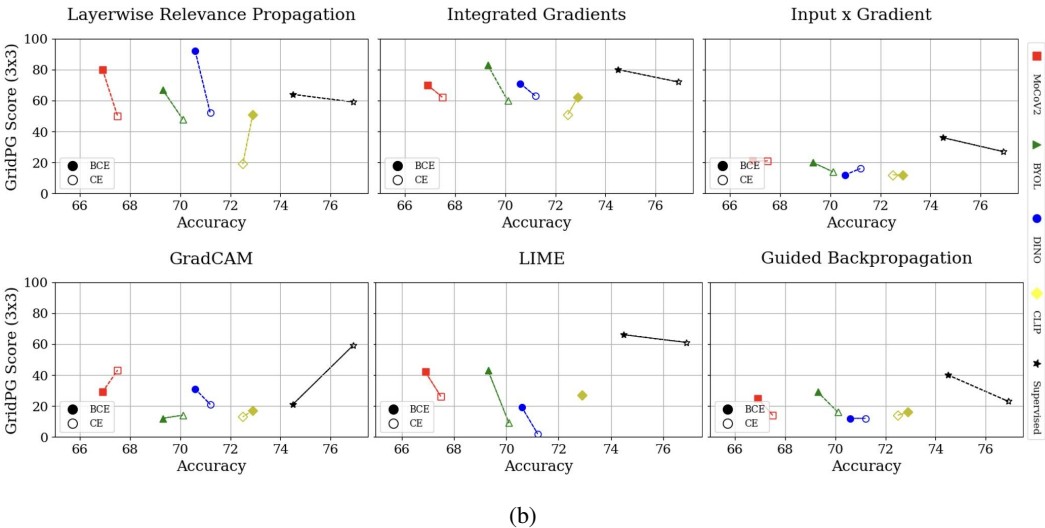

(b)

Figure E1: **BCE vs. CE — Accuracy and GridPG scores on ImageNet.** To study the impact of the optimization objective on probing we plot the accuracy vs localization (GridPG) score for **conventional models** (RN50). For BCE trained probes (solid markers), an improvement in the localization score is seen for all SSL methods for LRP, IntGrad and LIME explanations on ImageNet. However, for I×G, and GradCAM this is not seen. For Guided Backpropagation there is a slight improvement in localization score for BCE trained probes (except for DINO model).

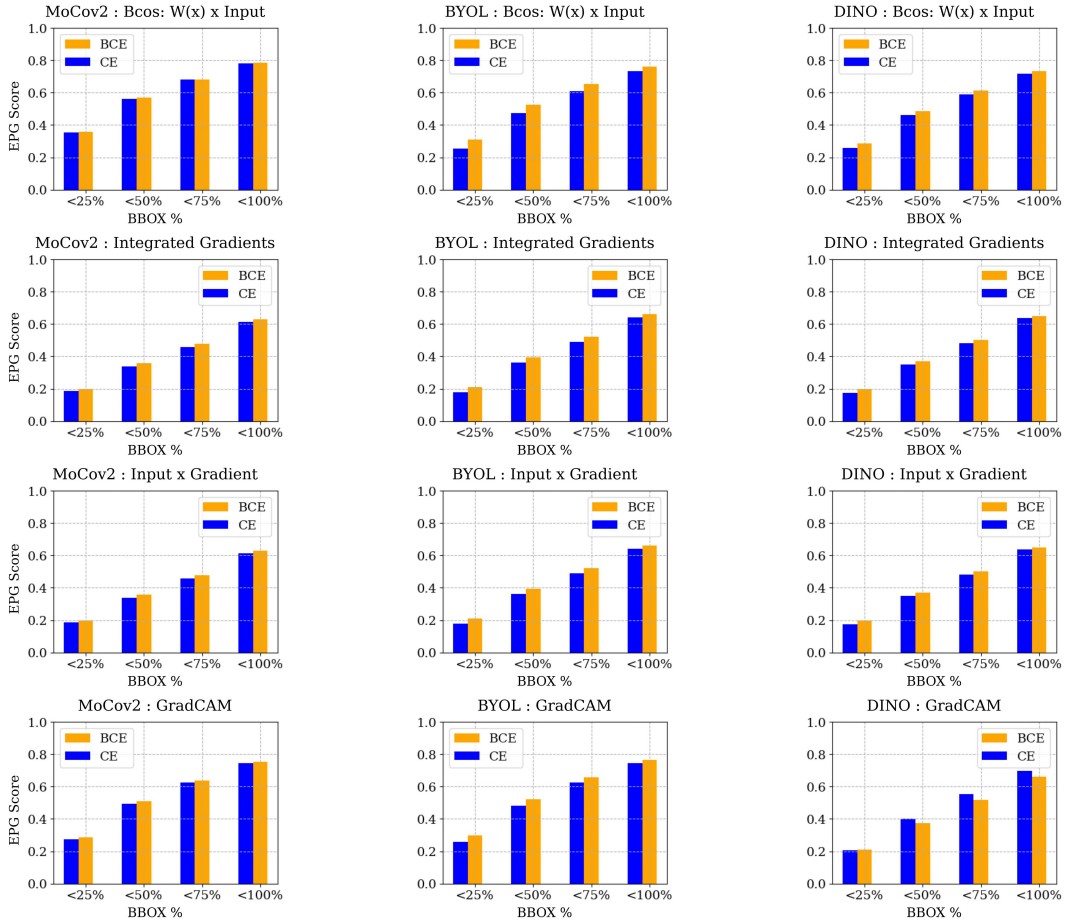

Figure E2: **BCE vs. CE — EPG scores for B-cos models on ImageNet.**

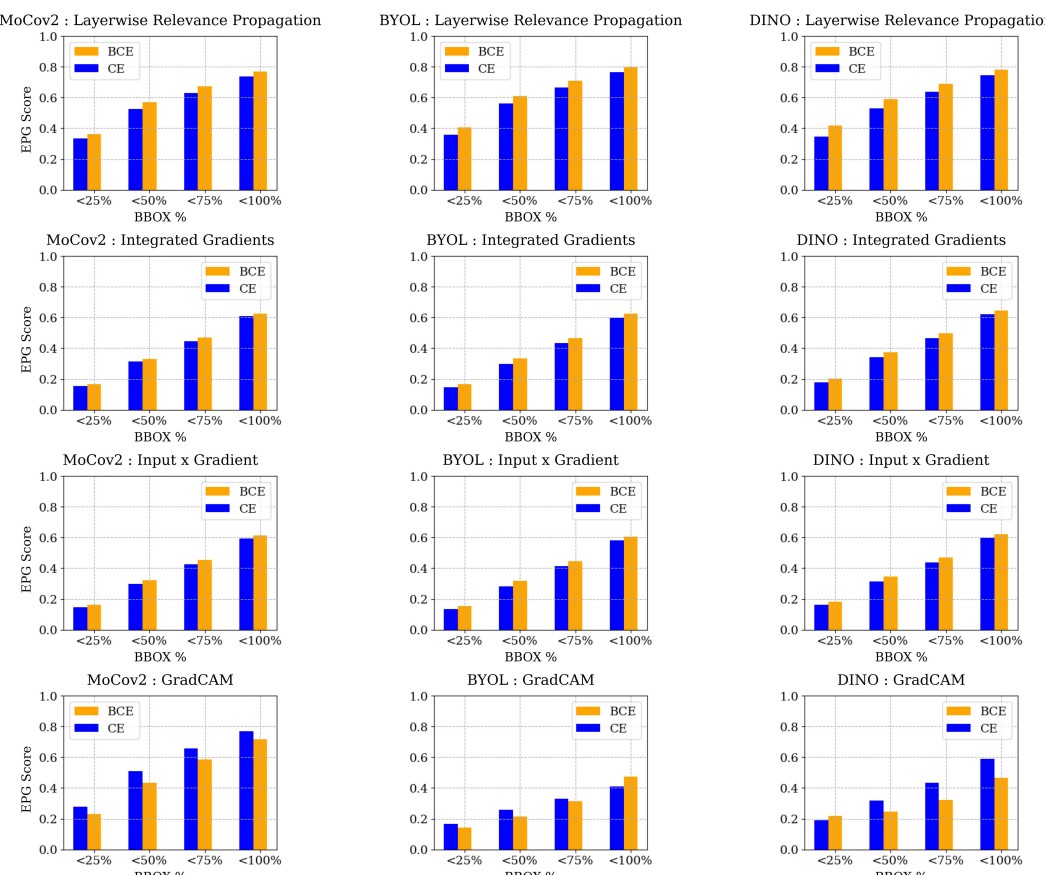

Figure E3: **BCE vs. CE — EPG scores for Conventional models on ImageNet.**

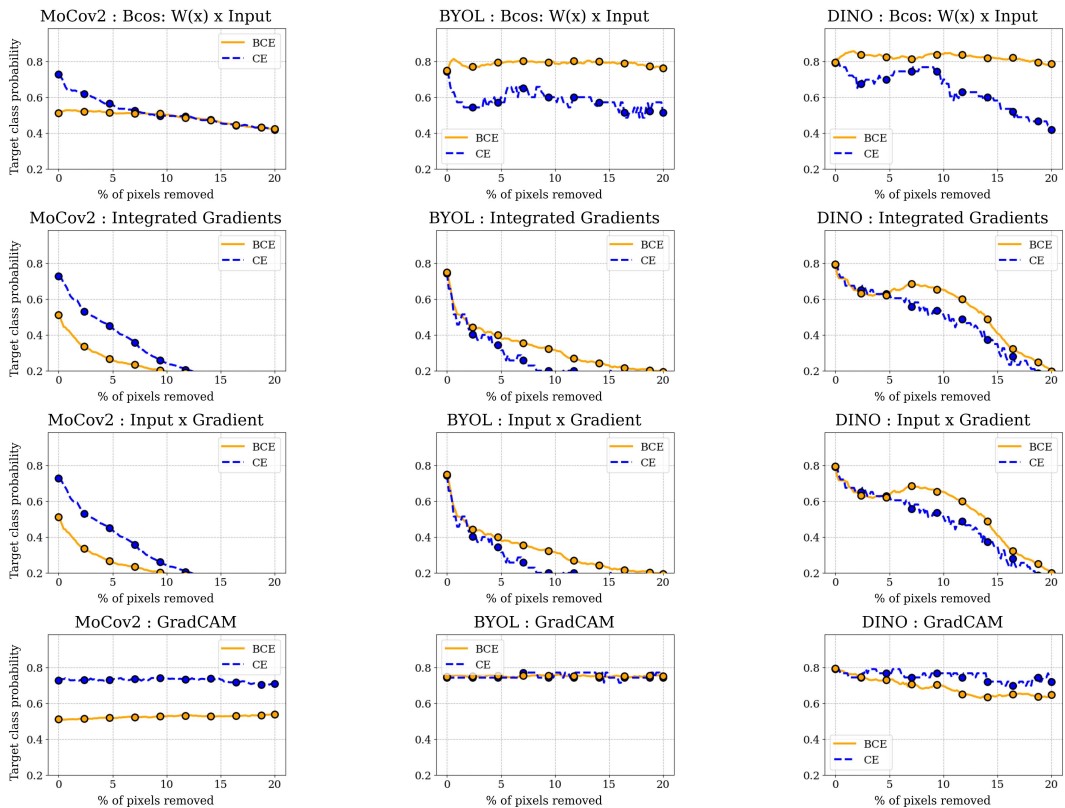

Figure E4: **BCE vs. CE — Pixel deletion scores for B-cos models on ImageNet.**

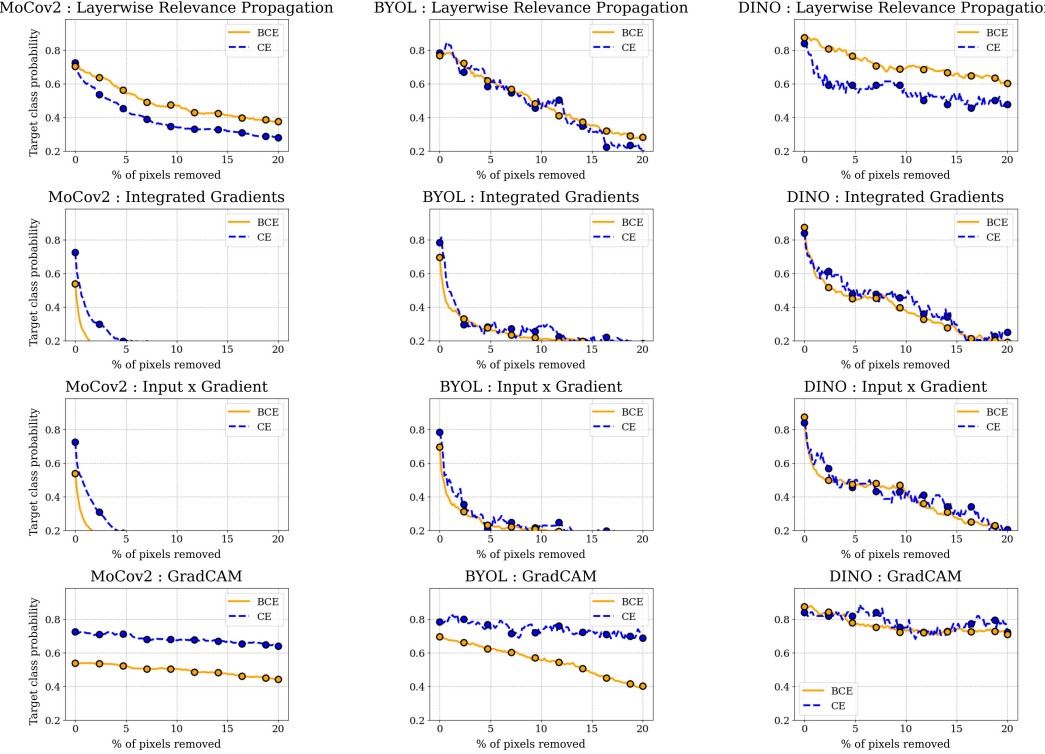

Figure E5: **BCE vs. CE — Pixel deletion scores for Conventional models on ImageNet.**

Table E1: **Compactness and Complexity of BCE vs CE Probing.** A consistent improvement in *compactness* (Gini index pp. as in (Chalasani et al., 2018)) and reduction in *complexity* (entropy as in (Tseng et al., 2020)) for BCE vs. CE, except for Input×Gradients and GradCAM with conventional backbones.

| Backbone | Pre-training Method | XAI Method | Accuracy | | ΔComplexity↓ | ΔCompactness↑ |
| | | | CE | BCE | | |
|---|---|---|---|---|---|---|
| B-RN50 | MoCoV2 | Bcos: W(x) x Input | 66.3 | 65.4 | -2.72 | +0.10 |
| B-RN50 | BYOL | Bcos: W(x) x Input | 68.3 | 68.1 | -1.87 | 0.14 |
| B-RN50 | DINO | Bcos: W(x) x Input | 68.8 | 67.5 | -2.35 | +0.16 |
| B-RN50 | MoCoV2 | Input x Gradient | 66.3 | 65.4 | -0.13 | 0.00 |
| B-RN50 | BYOL | Input x Gradient | 68.3 | 68.1 | -0.08 | 0.00 |
| B-RN50 | DINO | Input x Gradient | 68.8 | 67.5 | -0.08 | 0.00 |
| B-RN50 | MoCoV2 | IntGrad | 66.3 | 65.4 | -0.09 | 0.00 |
| B-RN50 | BYOL | IntGrad | 68.3 | 68.1 | -0.04 | 0.00 |
| B-RN50 | DINO | IntGrad | 68.8 | 67.5 | -0.05 | 0.00 |
| B-RN50 | MoCoV2 | GradCAM | 66.3 | 65.4 | -1.07 | +0.13 |
| B-RN50 | BYOL | GradCAM | 68.3 | 68.1 | -2.82 | +0.20 |
| B-RN50 | DINO | GradCAM | 68.8 | 67.5 | -2.51 | +0.19 |
| B-RN50 | MoCoV2 | GBP | 66.3 | 65.4 | -0.23 | +0.02 |
| B-RN50 | BYOL | GBP | 68.3 | 68.1 | -0.14 | +0.01 |
| B-RN50 | DINO | GBP | 68.8 | 67.5 | -0.13 | +0.01 |
| RN50 | MoCoV2 | Input x Gradient | 67.5 | 66.9 | +0.01 | 0.00 |
| RN50 | BYOL | Input x Gradient | 70.1 | 69.3 | -0.04 | +0.01 |
| RN50 | DINO | Input x Gradient | 71.2 | 70.6 | +0.01 | 0.00 |
| RN50 | MoCoV2 | IntGrad | 67.5 | 66.9 | -0.01 | 0.00 |
| RN50 | BYOL | IntGrad | 70.1 | 69.3 | -0.08 | 0.00 |
| RN50 | DINO | IntGrad | 71.2 | 70.6 | 0.00 | 0.00 |
| RN50 | MoCoV2 | LRP | 67.5 | 66.9 | -1.48 | +0.07 |
| RN50 | BYOL | LRP | 70.1 | 69.3 | -0.85 | +0.04 |
| RN50 | DINO | LRP | 71.2 | 70.6 | -2.01 | +0.12 |
| RN50 | MoCoV2 | GradCAM | 67.5 | 66.9 | +1.13 | -0.25 |
| RN50 | BYOL | GradCAM | 70.1 | 69.3 | +1.61 | -0.25 |
| RN50 | DINO | GradCAM | 71.2 | 70.6 | -2.94 | +0.23 |
| RN50 | MoCoV2 | GBP | 67.5 | 66.9 | -0.14 | +0.01 |
| RN50 | BYOL | GBP | 70.1 | 69.3 | -0.24 | +0.02 |
| RN50 | DINO | GBP | 71.2 | 70.6 | 0.00 | 0.00 |

Table E2: **Supervised Probing.** We present the effect of probing fully-supervised models (ref. Section 4.1 in main paper) for different attribution methods (B-cos Böhle et al. (2022), IntGrad Sundararajan et al. (2017), I×G Shrikumar et al. (2017), GradCAM Selvaraju et al. (2017), and LRP Bach et al. (2015)), and demonstrate how this impacts the localization (GridPG) score. We observe similar accuracy for both (CE and BCE) probes. However, as seen earlier (cf. fig. E1a,b), the probes trained with BCE loss lead to consistently high improvements in GridPG scores.

| | | Accuracy (%) | | | GridPG (%) | | |
|---|---|---|---|---|---|---|---|
| | **Att. Method** | CE | BCE | Δ | CE | BCE | Δ |
| **B-RN50** | – | 72.4 | 73.7 | +1.3 | – | – | – |
| | B-cos Böhle et al. (2022) | " | " | " | 15.0 | 87.0 | +72.0 |
| | IntGrad Sundararajan et al. (2017) | " | " | " | 24.0 | 86.0 | +62.0 |
| | I×G Shrikumar et al. (2017) | " | " | " | 15.0 | 65.0 | +50.0 |
| | GradCAM Selvaraju et al. (2017) | " | " | " | 14.0 | 72.0 | +58.0 |
| **RN50** | – | 75.5 | 75.6 | +0.1 | – | – | – |
| | LRP Bach et al. (2015) | " | " | " | 9.0 | 32.0 | +23.0 |
| | IntGrad Sundararajan et al. (2017) | " | " | " | 14.0 | 28.0 | +14.0 |
| | I×G Shrikumar et al. (2017) | " | " | " | 10.0 | 18.0 | +8.0 |
| | GradCAM Selvaraju et al. (2017) | " | " | " | 16.0 | 42.0 | +26.0 |

In table E1 we present the results for probing with BCE vs CE loss, when evaluated on the *compactness* and *complexity metrics*. A consistent improvement in **compactness** (Gini index pp. as in (Chalasani et al., 2018)) and reduction in **complexity** (entropy as in (Tseng et al., 2020)) for BCE vs. CE, except for Input×Gradients and GradCAM with conventional backbones.

**Probing Supervised Models.** In table E2, we demonstrate the effect of probing fully-supervised backbones with CE and BCE linear probes. This is to also increase comparability with the self-supervised and vision-language approaches that we present earlier (cf. fig. E1 (a) and (b)).

**Additional Perturbation-based Explanations** In table E3, we show the improved localization ability of BCE probes over CE probes for another perturbation-based explanation method called Score-CAM (Wang et al., 2020). See Figure D11 for qualitative results.

Table E3: **ScoreCAM BCE vs. CE Probing—GridPG scores on ImageNet.** A consistent improvement in *localization score* for BCE over CE probes is seen for both conventional and inherently interpretable B-cos backbones for supervised and self-supervised pre-training.

| | | Localization% | | |
|---|---|---|---|---|
| **Bakcbone** | **Pretraining** | CE | BCE | Δloc |
| RN50 | MoCov2 | 52.6 | 54.4 | +1.8 |
| RN50 | BYOL | 38.1 | 40.5 | +2.4 |
| RN50 | DINO | 44.2 | 55.9 | +11.7 |
| BRN50 | MoCov2 | 30.2 | 43.2 | +13.0 |
| BRN50 | BYOL | 45.3 | 50.2 | +4.9 |
| BRN50 | DINO | 51.0 | 67.7 | +16.7 |

E.2 EFFECT OF MORE COMPLEX PROBES

In this subsection, we present the quantitative results for probing with more complex MLP probes, specifically instead of using a single linear probe on top of frozen backbone features, we use two- and three-layer MLPs for training on downstream tasks and study its effect on performance (in terms of classification accuracy or f1-score) and explanation localization ability. For this setting, we train both B-cos and conventional MLPs, and show results for the same.

**Effect of MLP Probes on ImageNet** In fig. E6 we show the results on ImageNet for B-cos MLPs for both (a) B-cos and (b) conventional SSL models. It is observed that for all SSL models there is both an increase in accuracy as well as in the localization (GridPG) score across most attribution methods.

Next, in fig. E7 we plot the results to see the effect of conventional MLPs. In contrast to B-cos MLPs, here the stronger MLP based probing does lead to an increase in accuracy but does not always lead to an increase in GridPG localization score. This is observed independent of B-cos or conventional

model backbones. We attribute this to the alignment pressure introduced by B-cos layers (cf. (Böhle et al., 2022)) that helps distill object-class relevant features and rely less on background context. Thus, as discussed in the main paper in order to get attribution maps with good localization on downstream tasks, we suggest to probe pre-trained models with B-cos probes.

These are followed by presenting the bar plots for MLP probing experiments for EPG scores on ImageNet in figures E8, E9 for B-cos probes, and in figures E10, E11 for conventional probes.

Figures E12, E13, and E14 contain the pixel deletion plots on ImageNet for the MLP probing setup.

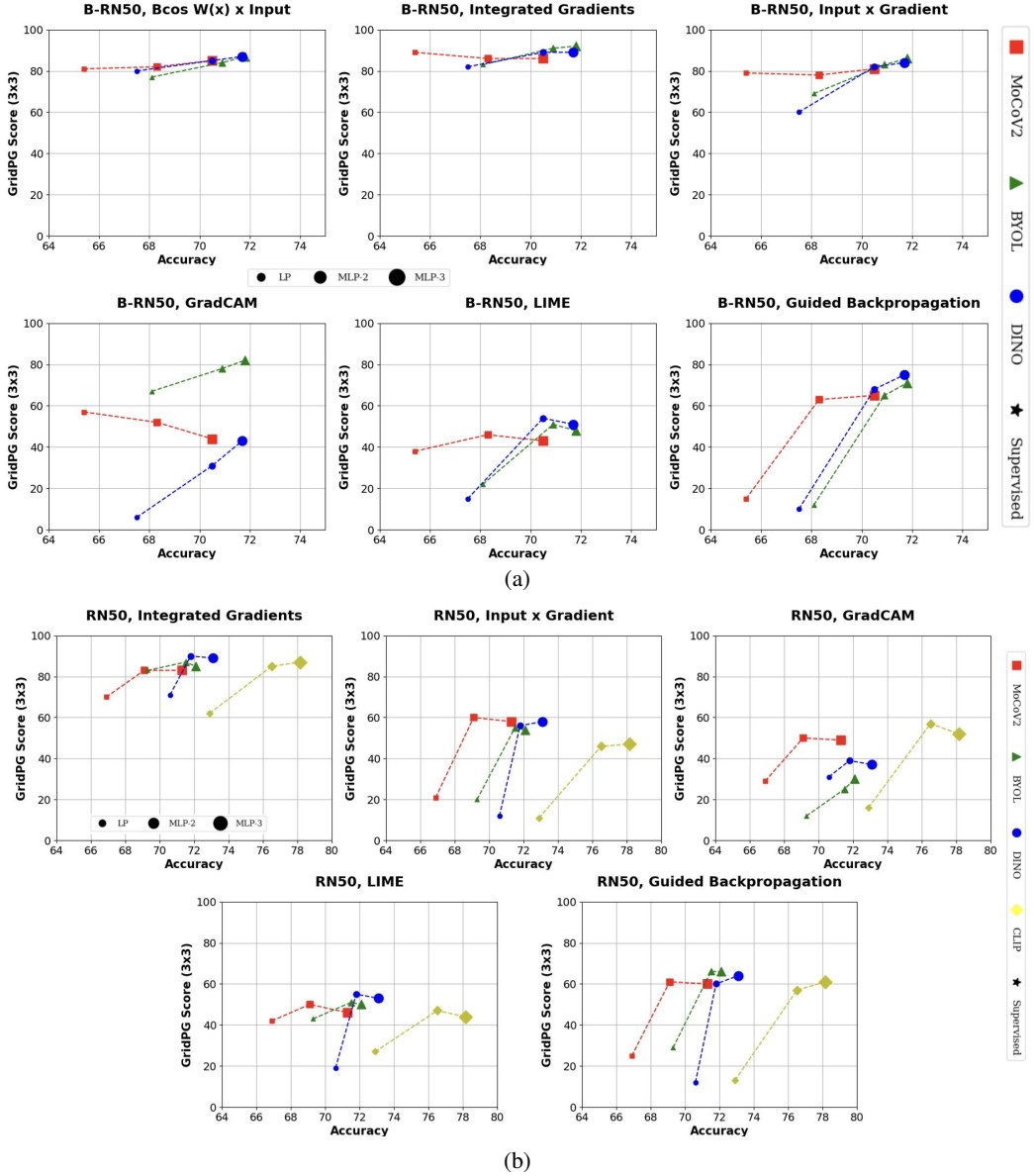

Figure E6: Effect of **complex B-cos probes** on accuracy and GridPG scores on **ImageNet** on (a) B-cos backbones (B-RN50) and (b) conventional backbones (RN50). Notice that there is a consistent improvement in accuracy (**x-axis**) as well as GridPG localization score (**y-axis**) as we move from a single probe to two- and three-layer MLP probes in most cases. Except only for B-cos models trained with MoCov2, for IntGrad (top-middle, subplot a) and GradCAM (bottom-left, subplot a) explanations we see a slight drop in localization scores. For the remaining cases (SSL frameworks and explanation methods), there is a steady increase in both accuracy and localization.

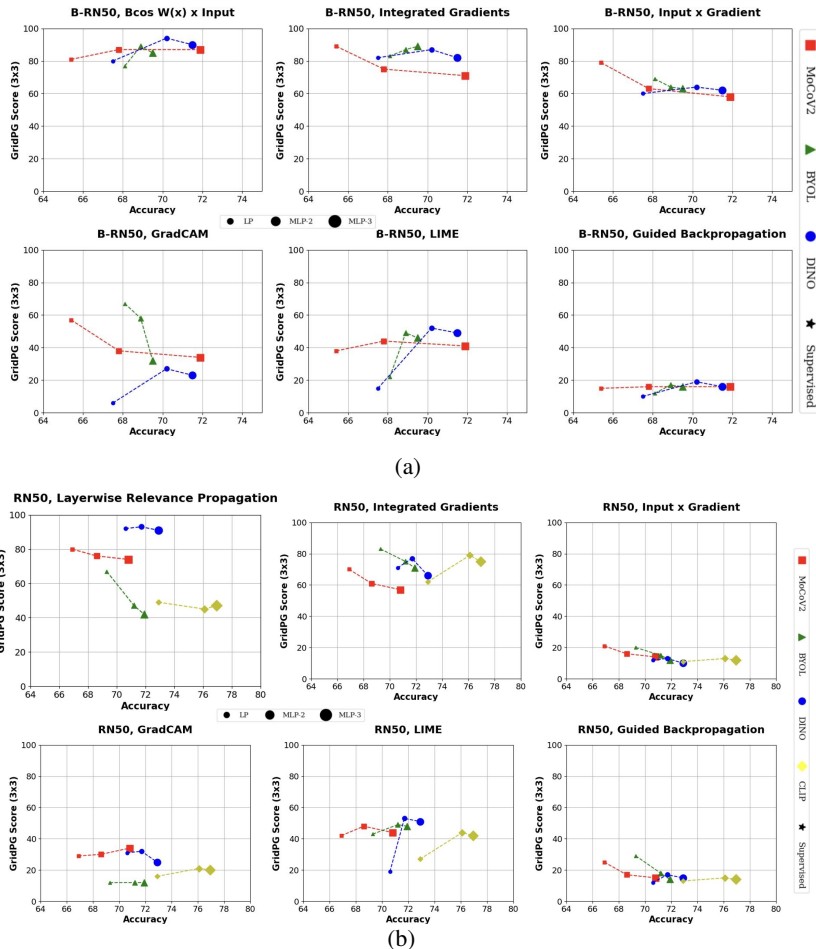

Figure E7: Effect of **complex conventional probes** on accuracy and GridPG scores on **ImageNet** on (a) B-cos backbones (B-RN50) and (b) conventional backbones (RN50). Interestingly, in contrast to B-cos MLP probes (cf. fig. E6), when conventional MLP probes are used, although there is an improvement in accuracy for all SSL models, there is no consistent improvement in the GridPG localization score. In fact, in several cases there is a drop in localization score as we move from a single probe to conventional MLP probes.

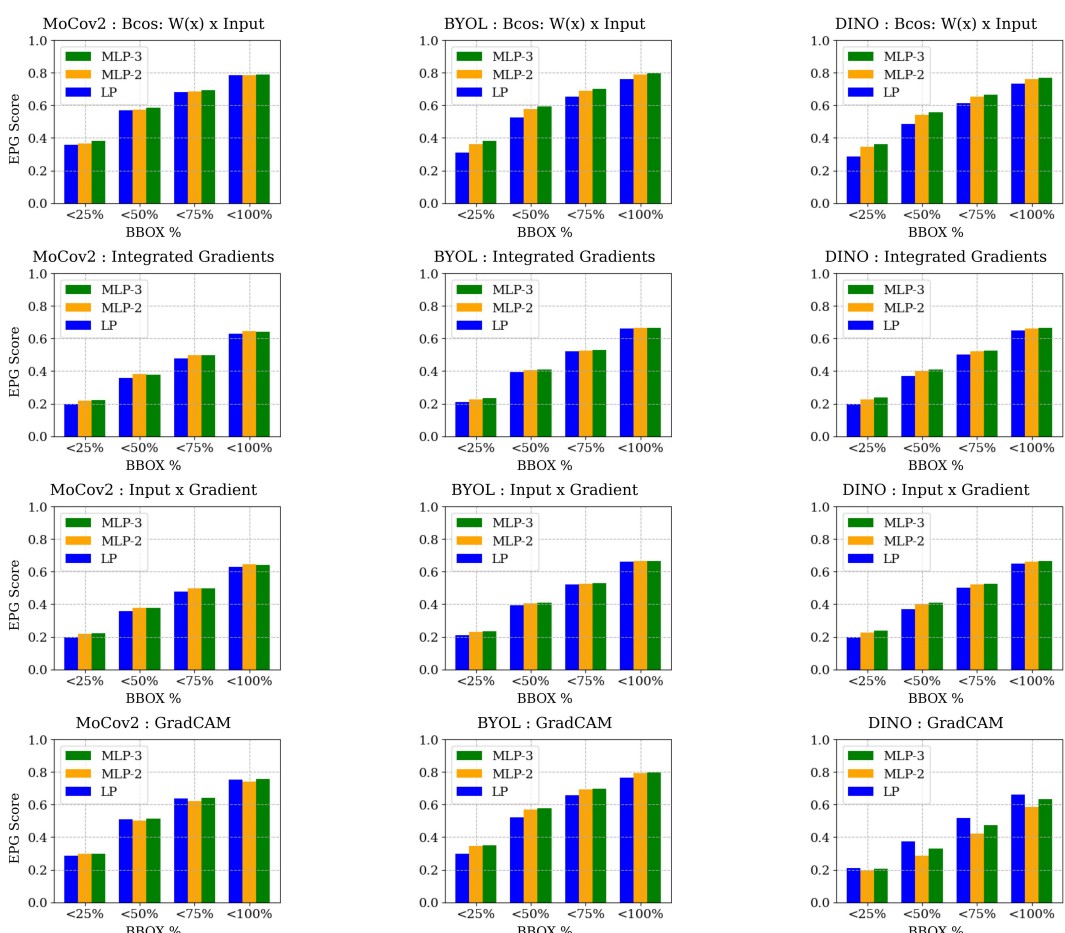

Figure E8: **Bcos-MLP — EPG scores for Bcos models on ImageNet.**

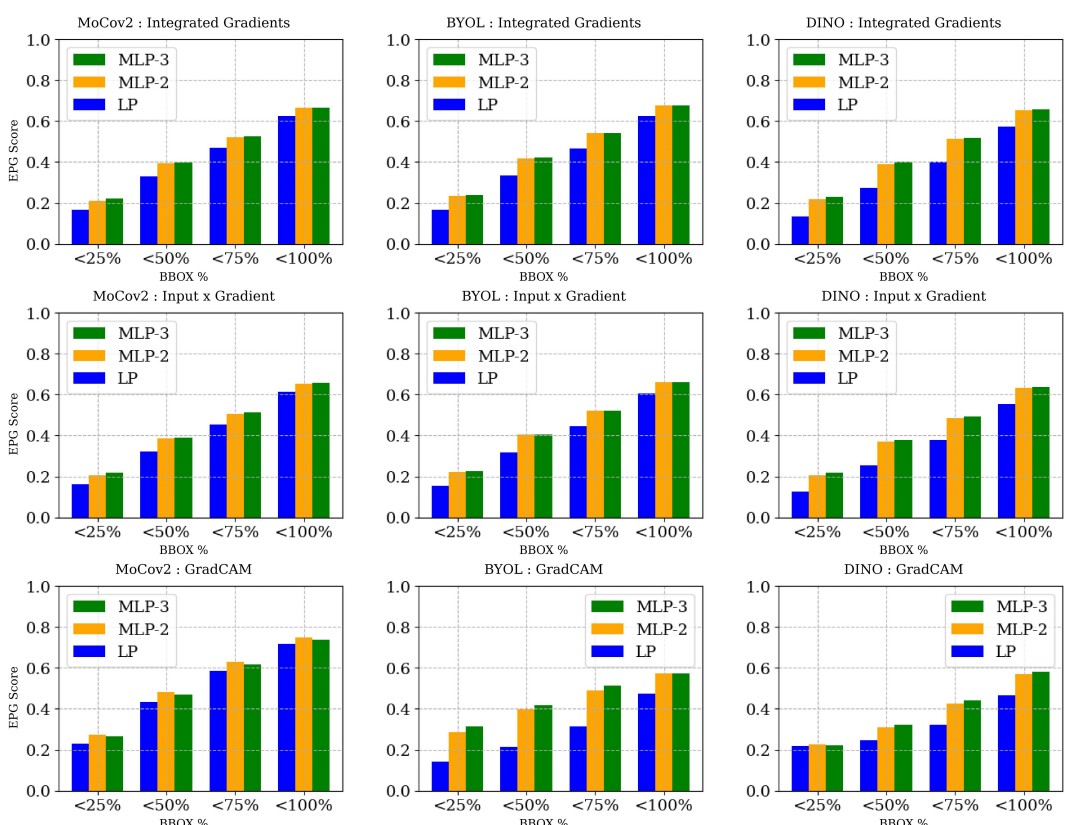

Figure E9: **Bcos-MLP —EPG scores for Conventional models on ImageNet.**

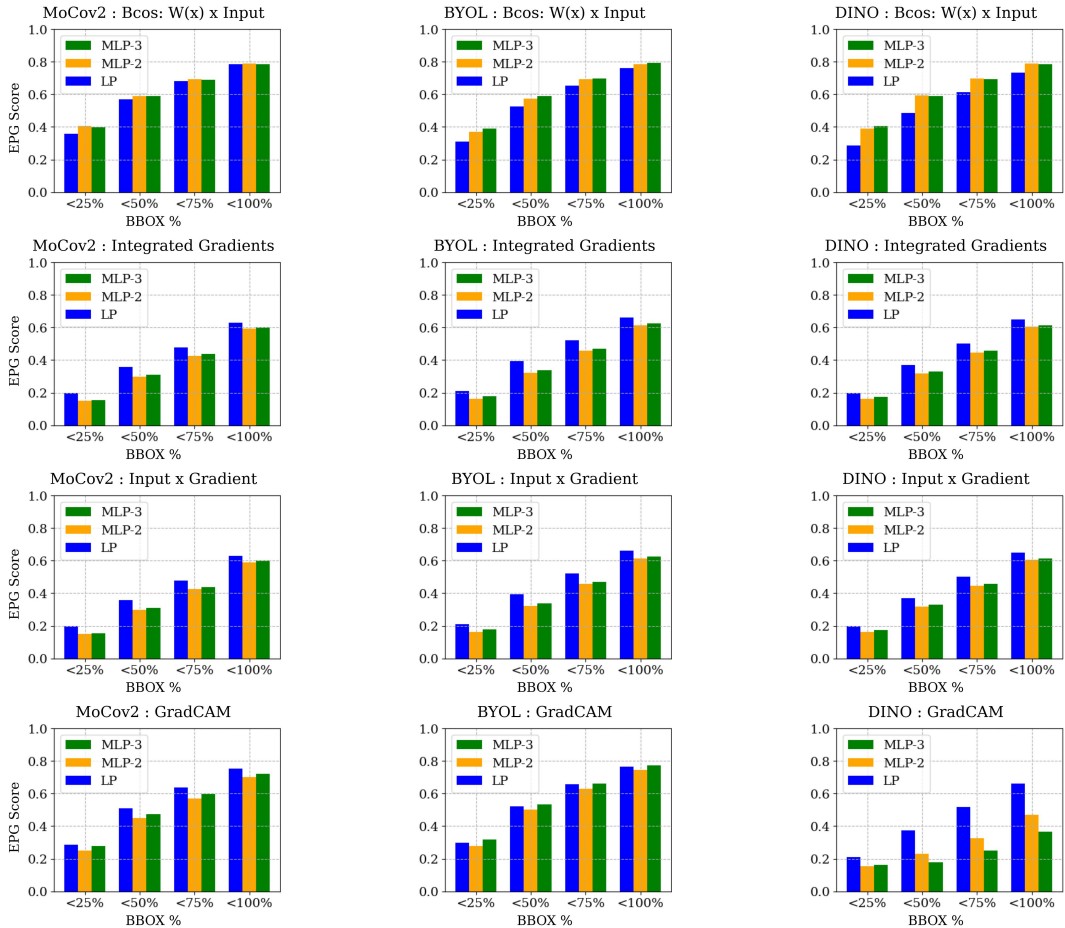

Figure E10: **Conventional-MLP — EPG for B-cos models on ImageNet.**

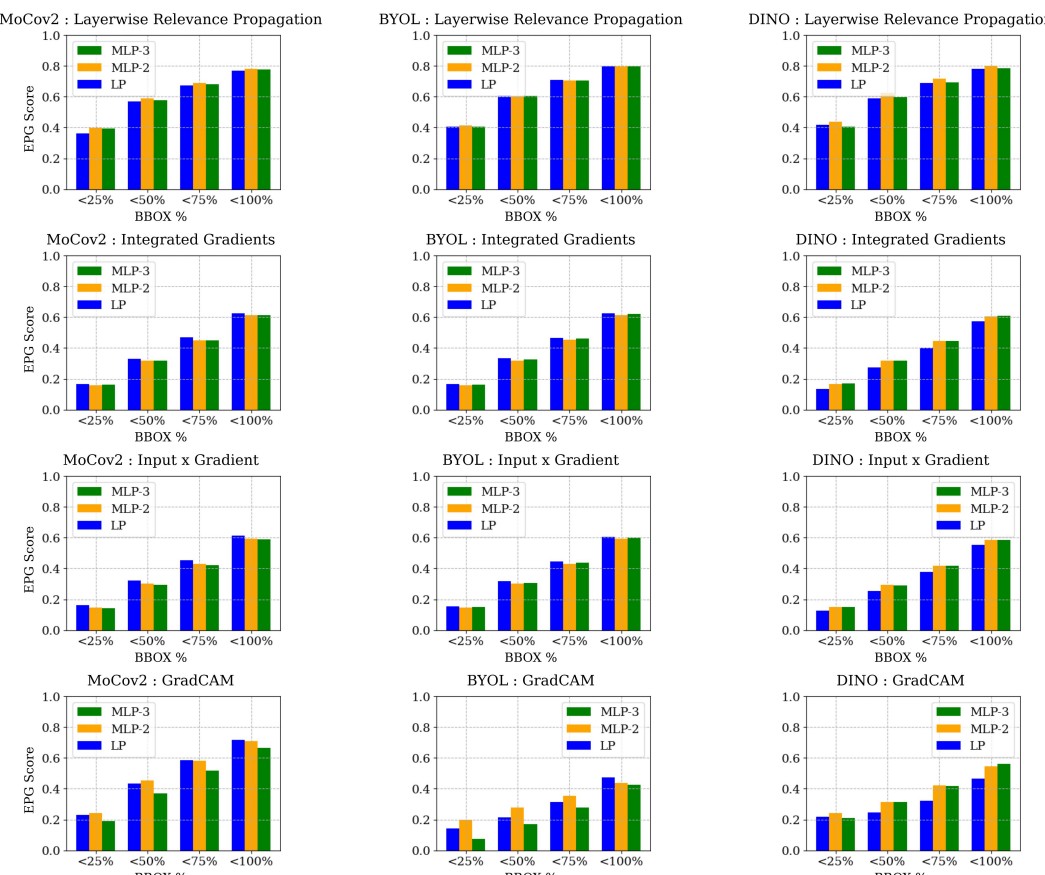

Figure E11: **Conventional-MLP — EPG for Conventional models on ImageNet.**

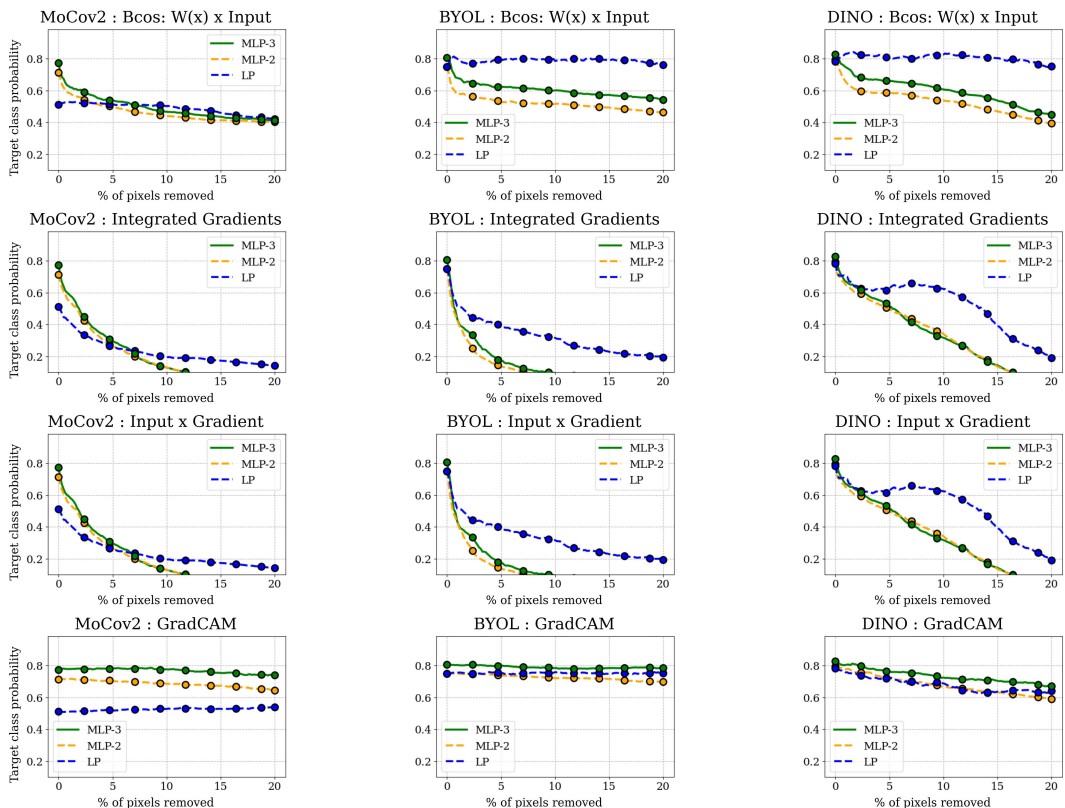

Figure E12: **Bcos-MLP — Pixel deletion scores for Bcos models on ImageNet.**

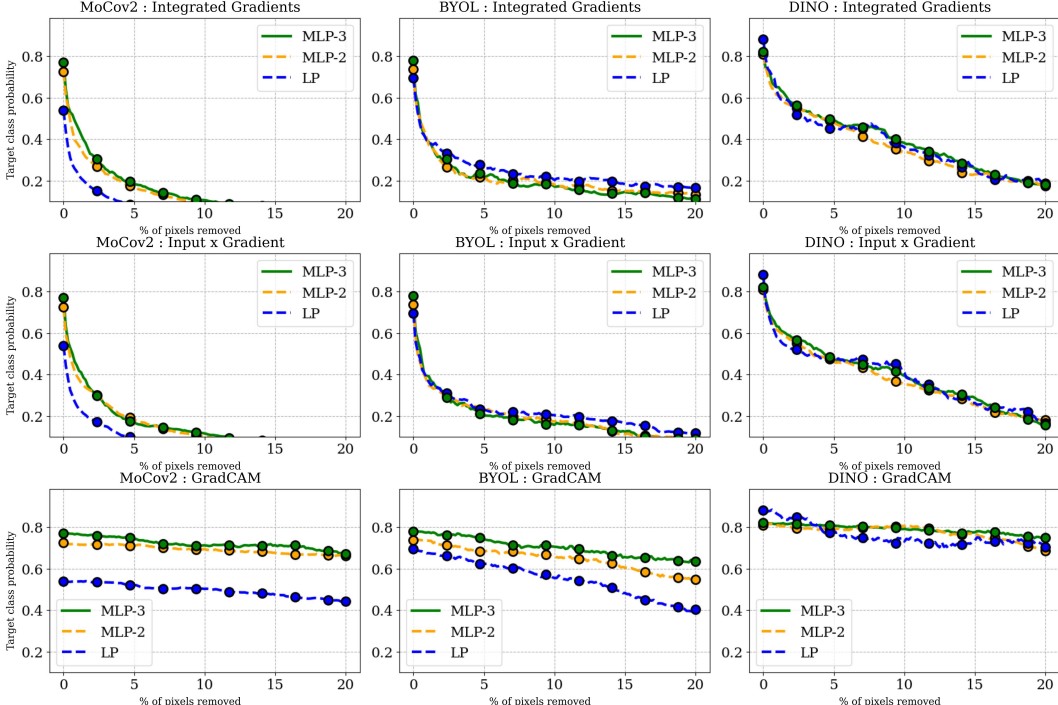

Figure E13: **Bcos-MLP — Pixel deletion scores for Conventional models on ImageNet.**

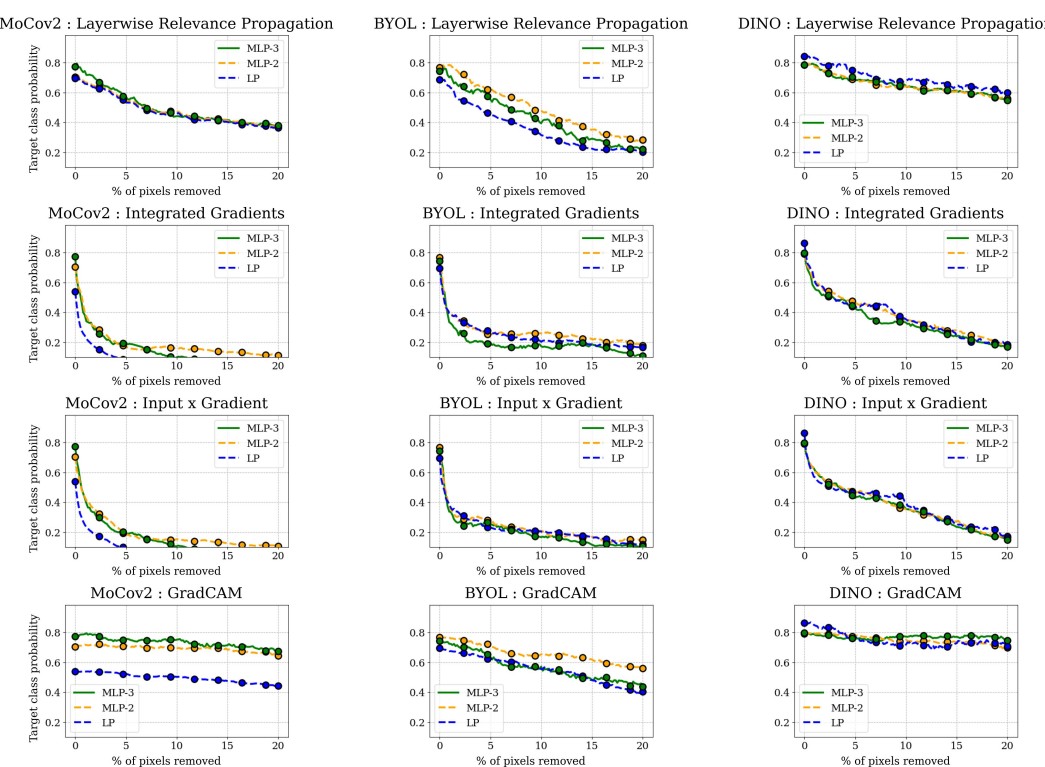

Figure E14: **Conventional-MLP — Pixel deletion scores for Conventional models on ImageNet.**

**Evaluation on Multi-label Classification** We also evaluate all SSL models (both B-cos and conventional backbones) and explanation methods under the multi-label classification setting. We train single probes and MLP probes using the BCE loss (as is typical) on COCO and VOC datasets. To measure model performance we report f1-score and explanation localization we report the EPG score. Figures E15, E16 show the f1-score vs EPG score plots for models probed with B-cos MLPs and conventional MLPs respectively on COCO. Here we notice, that stronger MLPs result in an improvement in both classification performance (depicted by an increase in f1-score), and also improved localization EPG scores.

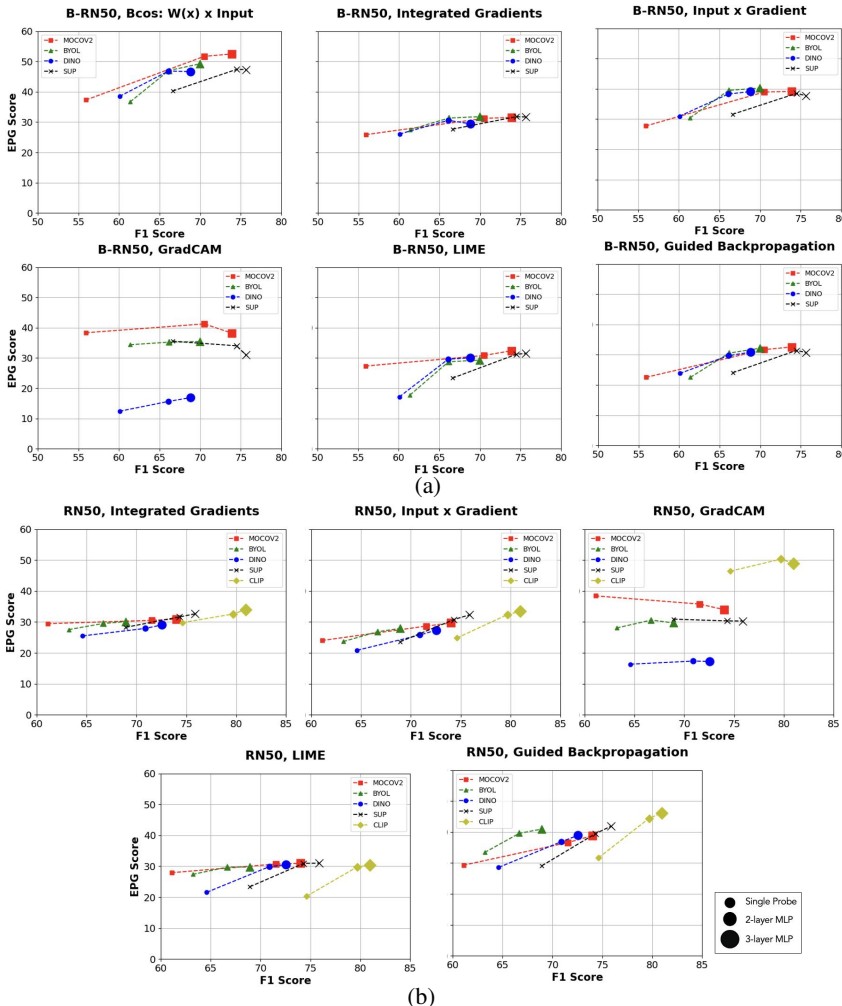

Figure E15: Effect of **complex B-cos probes** on accuracy and EPG scores on **COCO** on (a) B-cos backbones (B-RN50) and (b) conventional backbones (RN50). When we train with B-cos MLP probes as compared to a single probe, we observe an improvement in both f1-score (**x-axis**) and EPG localization score (**y-axis**) for all SSL models and explanation methods, except for GradCAM where improvement in localization is not seen consistently.

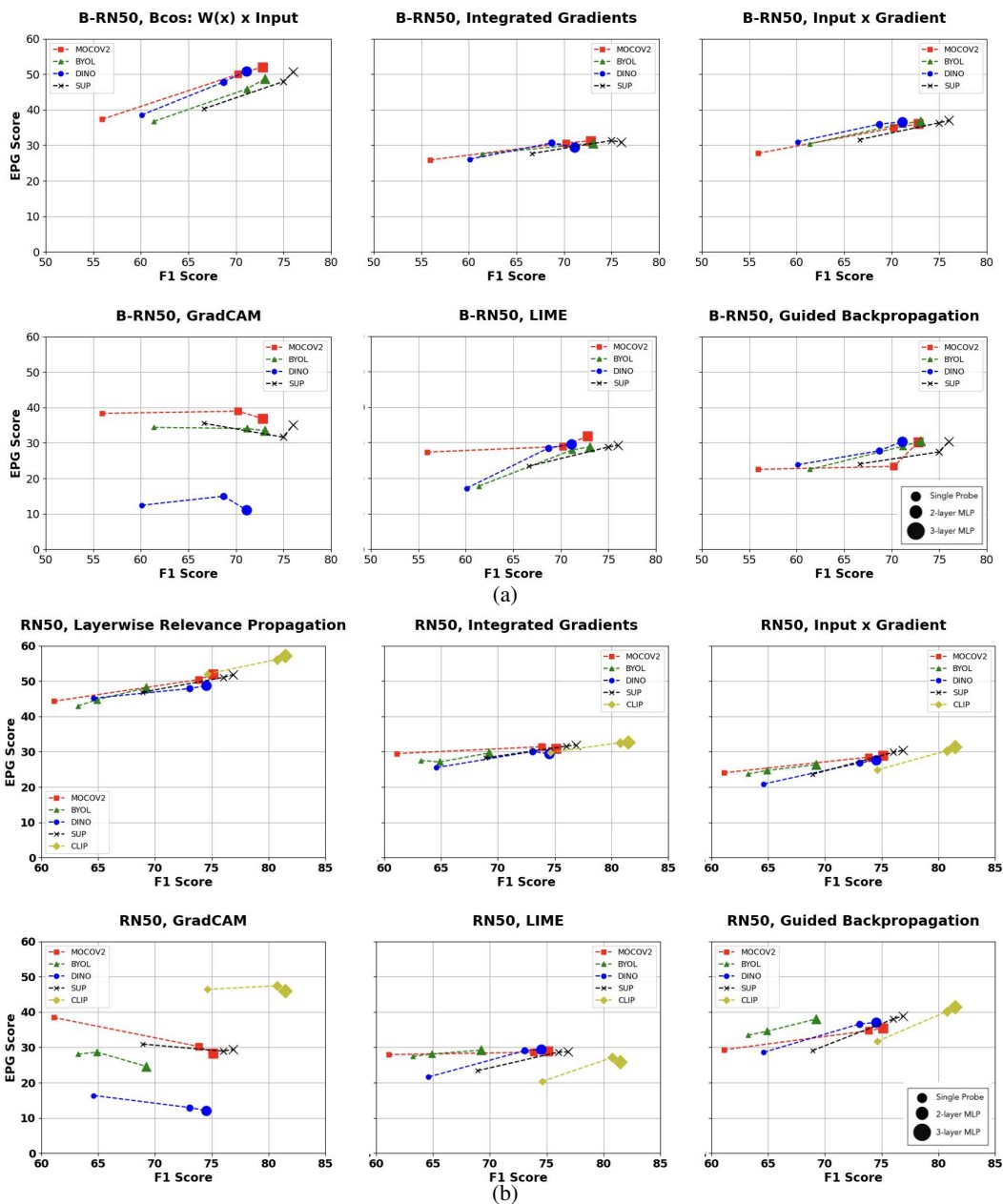

Figure E16: Effect of **complex conventional probes** on accuracy and EPG scores on **COCO** on (a) B-cos backbones (B-RN50) and (b) conventional backbones (RN50). Similar to fig. E15, when training with conventional MLP probes as compared to a single probe, we observe an improvement in both f1-score (**x-axis**) and EPG localization score (**y-axis**) for all SSL models and explanation methods, except for GradCAM (bottom-left, subplots a,b) where improvement in localization is not seen consistently.

Likewise, Figures E17, E18 show the corresponding plots for **VOC** for B-cos MLPs and conventional MLPs respectively. We see a similar trend as we saw on COCO, complex (two- and three-layer MLPs) probes lead to an increased f1-score and EPG score. Interestingly, we notice that GradCAM shows inconsistent behaviour, especially in the case of conventional MLPs (see Figures E16 and E18). This is a well-known limitation for GradCAM, which has been shown to perform poorly (in terms of localization) at earlier layers (Jiang et al., 2021). *Note: For B-cos MLPs we omit showing the results for LRP explanations as the relevance propagation rules for B-cos layers are not defined.*

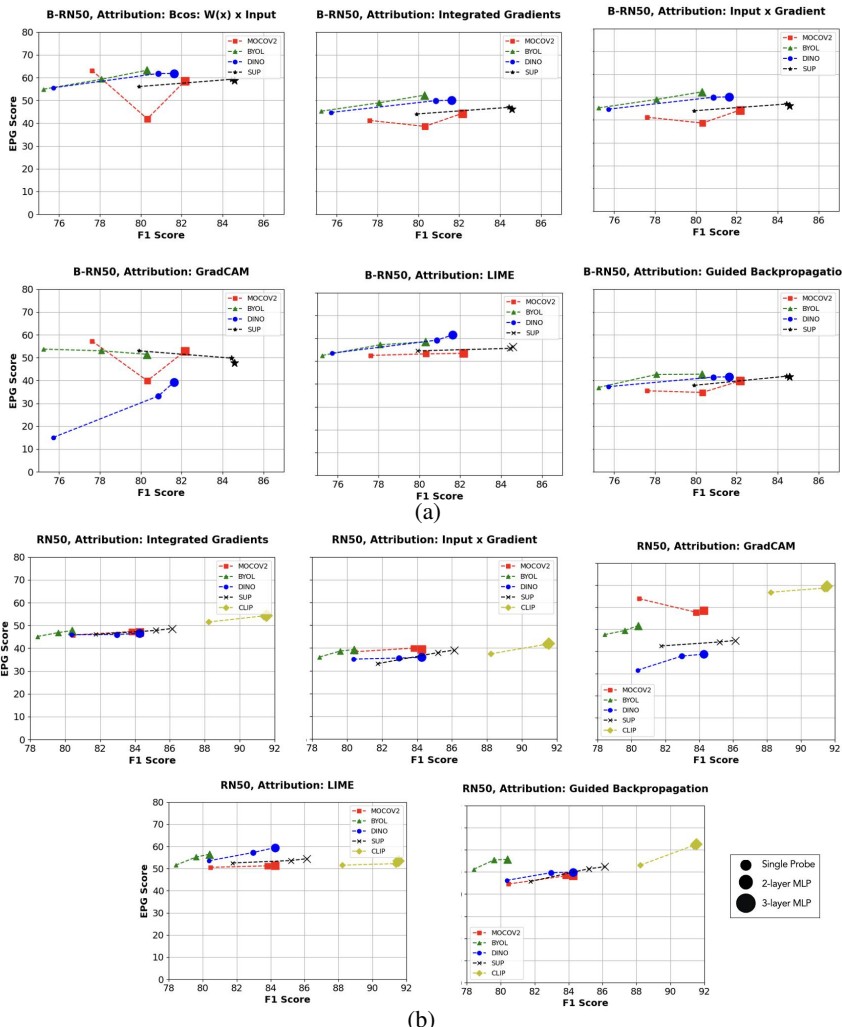

Figure E17: Effect of **complex B-cos probes** on accuracy and EPG scores on **VOC** on (a) B-cos backbones and (b) conventional backbones. Notice as seen for COCO (cf. fig. E15), stronger two- and three-layer MLPs lead to an improvement in both f1-score (**x-axis**) and EPG localization score (**y-axis**). Once again, for GradCAM this is not very consistent. Also note that for B-cos models trained with MoCov2, B-cos explanations (top-left, subplot a) for MLP probes perform worse in EPG-score as compared to a single probe.

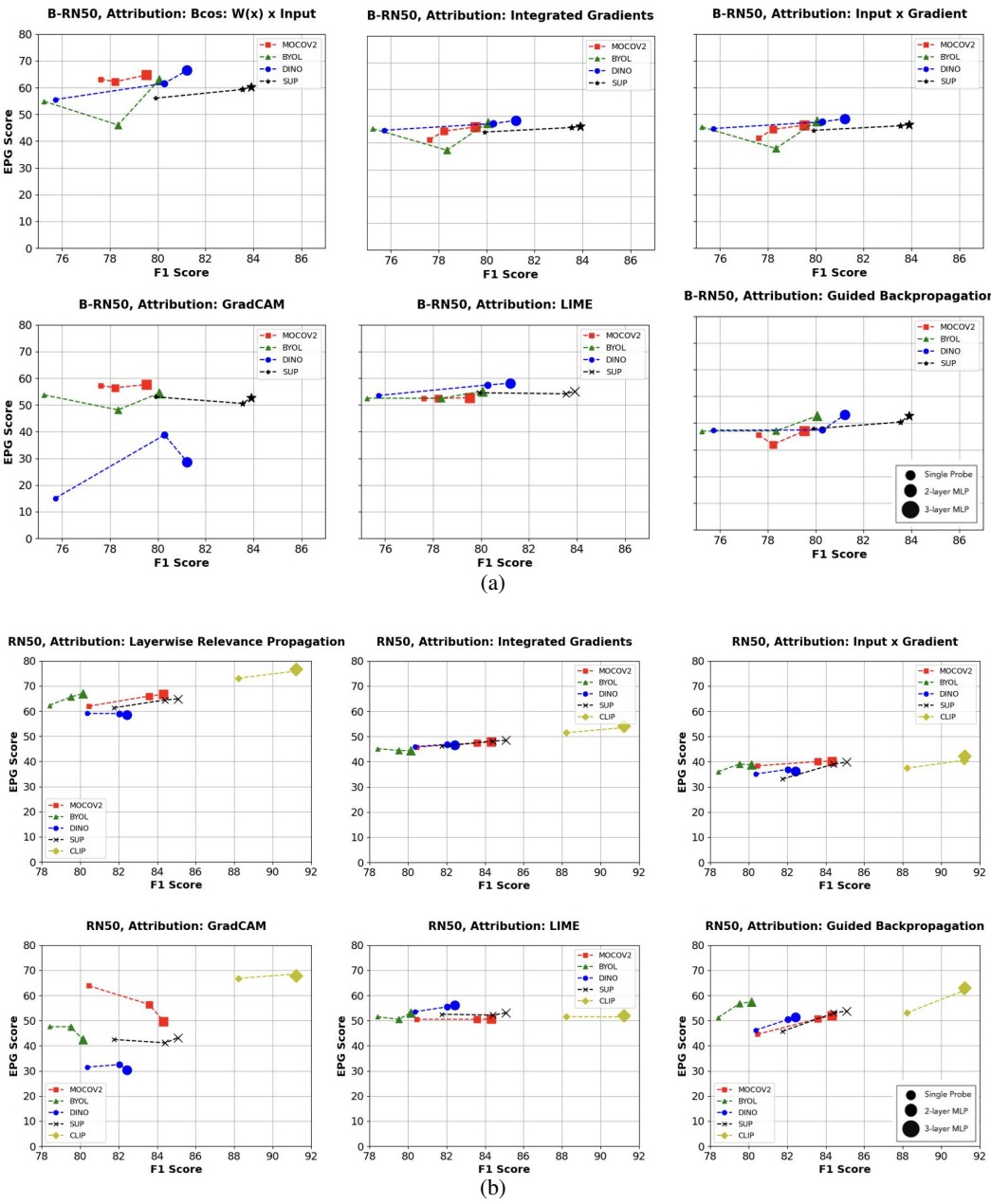

Figure E18: Effect of **complex conventional probes** on accuracy and EPG scores on **VOC** on (a) B-cos backbones and (b) conventional backbones. Even conventional two- and three-layer MLPs lead to an improvement in both f1-score (**x-axis**) and EPG localization score (**y-axis**). Yet again, for GradCAM this is not consistent.

# F    IMPLEMENTATION DETAILS

In this section we describe in detail our experimental setting, i.e. the datasets we train and evaluate on, the training and implementation details, and finally the evaluation metrics used.

## F.1    DATASETS

For our experiments we use three datasets namely ImageNet (Russakovsky et al., 2015), VOC 2007 (Everingham et al., 2009) and MS COCO 2014 (Lin et al., 2014).

**ImageNet** We use the ImageNet-1K dataset that is part of the ImageNet Large-Scale Visual Recognition Challenge (ILSVRC) (Russakovsky et al., 2015). This has 1000 classes, with roughly 1000 images belonging to each category. In total, there are 1,281,167 training images, and 50,000 validation images. We perform all self-supervised pre-training on the training set, and evaluation on the validation set. For training, the images are resized to an input resolution of 224×224. For measuring the performance of the model the top-1 accuracy (proportion of correctly classified samples) is reported, and to evaluate model explanations the *grid pointing game* (GridPG) localization score is reported (discussed in more detail below).

**VOC 2007** VOC 2007 (Everingham et al., 2009) is a popularly used multi-label image classification dataset. It comprises of 9,963 images in total and 20 object classes, and is split into the *train-val* set with 5,011 images and the *test* set with 4,952 images. We use the *train-val* set for training and *test* set for evaluation. For training, the images are resized to a fixed resolution of 224×224. As is typical (Rao et al., 2023) for evaluating multi-label classification datasets, we report the f1-score and use the *energy pointing game* (EPG) score to evaluate the model explanations.

**MS COCO 2014** Microsoft COCO (Lin et al., 2014) is another popular dataset generally used for image classification, segmentation, object detection and captioning tasks. We use COCO-2014 in our experiments, that has 82,081 training and 40,137 validation images and 80 object classes. For our training, we resize the images to a fixed size of 448×448. Similar to VOC for evaluation we report the f1-score and use the EPG score to evaluate model explanations. Also note that the variation of objects' shapes and sizes are more complicated (Zhu & Wu, 2021) on COCO than those in VOC, and it is also substantially larger with more object classes.

**In short,** we evaluate the models on (1) *single-label* and (2) *multi-label* classification. For (1), we train probes on top of the frozen pre-trained features on ImageNet (Russakovsky et al., 2015), and report top-1 accuracy on the validation set. For (2), the probes are trained to predict all classes that are present in an image; for this, we use VOC (Everingham et al., 2009) and COCO (Lin et al., 2014) and report the F1 scores on the test set.

## F.2    MODELS

We evaluate both conventional (ResNet-50 (He et al., 2016), ViT-B/16 (Kolesnikov et al., 2021)) and the recently proposed inherently interpretable B-cos models (B-cos ResNet-50, ViT$_c$-S/16, ViT$_c$-B/16) (Böhle et al., 2024). To adapt these to the various pre-training paradigms, we follow (Böhle et al., 2024) in converting the MLP heads of the respective SSL methods. Specifically, we replace the linear layers in the MLP *projection heads* (MoCov2, BYOL, DINO) and the *prediction head* (BYOL) with corresponding B-cos layers (B=2), remove all ReLU non-linearities, and replace all batch normalization layers with the corresponding uncentered version, see (Böhle et al., 2024).

## F.3    PRE-TRAINING FRAMEWORKS

As discussed previously, we aim to have a broad enough representative set that highlights how our evaluation generalizes across differently trained feature extractors, particularly (1) fully-supervised, (2) self-supervised, and (3) contrastive vision-language learning.

**(1) Fully-Supervised Learning** We first, evaluate the explanation methods on fully supervised backbones. Since, the backbones pre-trained in a supervised manner are still often used for transfer learning (Cheng et al., 2022; Xie et al., 2021; Chen et al., 2017). Additionally, an evaluation of fully supervised models also provides a useful reference value, as most explanation methods have been developed in this context.

In addition to evaluating the end-to-end trained classifiers, we also evaluate linear probes on the frozen representations of these models, in order to increase the comparability to the self-supervised approaches we present in the following.

**(2) Self-Supervised Learning.** We consider three popular self-supervised pretraining frameworks—MoCov2 (Chen et al., 2020c), BYOL (Grill et al., 2020), and DINO (Caron et al., 2021)—regarding their interpretability. In the following, we briefly describe each of them.

**MoCov2 (He et al., 2019)** employs a contrastive learning paradigm, which can be regarded as a form of instance classification. In particular, the backbone is trained to yield similar representations for different augmentations of the *same* image, whilst ensuring that representations of *different* images ('negatives') are dissimilar.

**BYOL (Grill et al., 2020)** constructs self-supervised learning as a form of Mean Teacher self-distillation (Tarvainen & Valpola, 2017) with no labels. Unlike MoCov2, it does not rely on negative pairs for training. The mean squared error between the normalized predictions (student) and target projections (teacher) is minimized during training.

**DINO (Caron et al., 2021)** uses a similar setup as BYOL, i.e., a self-distillation approach. Instead of employing an MSE loss, DINO optimizes for a low cross-entropy loss between the output representations of the teacher and the student models.

**(3) Vision-Language Learning.** CLIP (Radford et al., 2021) employs contrastive learning on a large-scale noisy dataset comprising of image-text pairs. It comprises of two encoders, an image encoder (ResNet (He et al., 2016) or ViT (Kolesnikov et al., 2021)) and a text encoder (Transformer (Vaswani et al., 2017)), and is optimized to align the embedding spaces of the two encoders.

Thus, to summarize, we evaluate across a broad spectrum of pre-training mechanisms: a *contrastive*, two *self-distillation-based*, and a *multi-modal* pre-training paradigm, which cover some of the most popular approaches to self-supervised learning. This is contrasted with fully-supervised trained models.

### F.4 TRAINING DETAILS

Here we mention the training details for the pre-training (self-supervised and fully-supervised), as well as probing experiments.

**Self-supervised Pre-training.** We pretrain all models on the ImageNet dataset (Russakovsky et al., 2015). For each self-supervised pre-training framework, we follow the standard recipes as mentioned in their respective works[5]. To keep the configuration consistent we use a batch size of 256 for all models, distributed over 4 GPUs and train for 200 epochs. The learning rate for each SSL framework is updated following the linear scaling rule (Goyal et al., 2017): $lr = 0.0005 \times \text{batchsize}/256$.

**Supervised Training** When training fully supervised models we train the model for 100 epochs. For probing the pre-trained SSL features, on ImageNet we train the probes for 100 epochs as is standard (Caron et al., 2021; Grill et al., 2020) and for 50 epochs when training on COCO and VOC datasets. Random resize crops[6] and horizontal flips augmentations are applied during training, and we report accuracy on a central crop. Note that we do not perform any hyperparameter tuning separately for every dataset, and use the same training procedure for all settings (except the number of epochs for which the probe is trained).

**Probing on Pre-trained SSL Features** When probing on the pre-trained SSL (and CLIP) features, as is done for B-cos (Böhle et al., 2022) models we first apply the classifier as a $1\times1$ convolution to the feature volume and then apply Global Average Pooling (GAP) to get the class-wise logits. To keep it consistent across all models, we apply this scheme to both B-cos as well as conventional models. Also it is important to note that when using conventional probes with B-cos models, we remove all biases.

**Note.** For fully supervised models and probing the pre-trained SSL features we use the following training configuration: the Adam (Kingma & Ba, 2015) optimizer, a batch size of 256, and a cosine learning rate schedule with warmup. Weight decay of 0.0001 is applied only for end-to-end training

---

[5]For B-cos models with DINO, Adam optimizer and a *learning rate* of *0.001* was used

[6]For VOC and COCO when applying random resized cropping, the crop size is limited within (0.7, 1.0) fraction of the original image size.

of standard models (i.e. non B-cos models). Random resize crops and horizontal flips augmentations are applied during training.

Finally, for the CLIP vision-language model we use the checkpoint for the conventional model provided by the authors' original implementation (Radford et al., 2021).

### F.5 EVALUATION DETAILS

**Model Performance** To evaluate the model performance on single-label datasets, i.e. ImageNet we report the top-1 accuracy: which is the proportion of correctly classified samples in the validation set. For multi-label datasets, i.e. COCO and VOC we report the f1-score as is typically done (Rao et al., 2023).

**Grid Pointing Game** As described in Section 3.1 of the main paper, we use the *grid pointing game* (GridPG) (Böhle et al., 2021) to evaluate the model explanations on single-label image classification datasets (ImageNet). Originally, GridPG was used to quantitatively evaluate explanation methods, instead in our work we use it to compare between diferently trained models. Figure 3 in the main paper shows an illustration of our evaluation setup for a 2×2 grid image. Similar to the original setting (Böhle et al., 2021; 2022), we construct 500, 3×3 image grids from the most confidently and correctly classified images for each model independently and report the mean GridPG score. In every grid all images belong to distinct classes, and for each of the corresponding class logits we measure the fraction of positive attribution an explanation method assigns to the correct location in the grid. To put it mathematically the GridPG localization score $L_i$ for the $i^{th}$ cell in the grid is defined as:

$$L_i = \frac{\sum_{p \in cell_i} A^+(p)}{\sum_{j=1}^{n^2} \sum_{p \in cell_j} A^+(p)}$$

where $A^+(p)$ denotes the amount of positive attributions given to the $p^{th}$ pixel in the $n \times n$ grid image. Also note that in the case that an image has no positive attributions (all attributions are negative), we ignore that sample from the evaluation set as $L_i$ is undefined in such cases since the denominator is 0.

**Energy Pointing Game** For both single-label and multi-label classification datasets we use the *energy pointing game* (EPG) to evaluate model explanations. Introduced in (Wang et al., 2020) the EPG measures the concentration of positive attributions inside the bounding box for each object class $k$ in the image. Finally we report the mean EPG score on the entire validation (or test) set. We follow the formulation as in (Rao et al., 2023) for a given image with height, width as H×W and $k^{th}$ class the $EPG_k$ score is given by:

$$EPG_k = \frac{\sum_{h=1}^{H} \sum_{w=1}^{W} M_{k,hw} A_{k,hw}^+}{\sum_{h=1}^{H} \sum_{w=1}^{W} A_{k,hw}^+}$$

where $A_k \in \mathbb{R}^{H \times W}$ denotes the attribution map for class $k$, and $A_k^+$ denotes the positive part of the attributions; $M_k \in \{0, 1\}^{H \times W}$ denotes the binary mask for class $k$ that is computed by taking the union of bounding boxes for all occurrences of class $k$ in the given image.

This evaluation setting might be used for any dataset that has available annotations in the form of bounding boxes or segmentation masks for the object classes of interest.

**Pixel Deletion** This evaluation procedure has been used in prior works (Samek et al., 2017; Hedström et al., 2024) as a reliable metric to evaluate the faithfulness of explanations derived from a given attribution method. In this setting, the least important pixels (or tokens) as given by an explanation are incrementally removed from the input. This should not affect the model's prediction and result in a slow decline of the model's prediction confidence.

**Compactness** In their work work Chalasani et al. (2018) used the Gini index p.p. as a measure of quality of explanations generated for neural networks during adversarial training. The higher the compactness, the better is the explanation quality.

**Complexity** The authors in (Tseng et al., 2020) used entropy as a measure of quality for explanations generated by attribution methods of deep learning models for genomics. Lower entropy of explanations is considered to indicate higher quality.

**Attribution Implementation** The following attribution (or explanation) methods are used in our work: B-cos (Böhle et al., 2022), Layer-wise Relevance Propagation (LRP) (Bach et al., 2015), GradCAM (Selvaraju et al., 2017), Integrated Gradients (IntGrad) (Sundararajan et al., 2017), Input×Gradient (I×G) (Shrikumar et al., 2017), LIME (Ribeiro et al., 2016) and GuidedBackpropagation (GBP) (Springenberg et al., 2015). For all methods except B-cos, LRP and LIME we use the implementations provided by the captum library (`github.com/pytorch/captum`). For IntGrad, similar to (Böhle et al., 2022) we set $n\_steps = 50$ for integrating over the gradients and a $batch\_size = 16$ to accommodate for limited compute. For computing LRP attributions we rely on the zennit library (`https://github.com/chr5tphr/zennit`) and use the $epsilon\_gamma\_box$ composite. Many relevance-propagation rules exist for LRP, in this work we focus on the $epsilon\_gamma\_box$ composite because it has been shown to work particularly well in (Montavon et al., 2019; Rao et al., 2024). For LIME attributions we use the official implementation available at `https://github.com/marcotcr/lime`. And for B-cos attributions we use the author provided implementation given at `https://github.com/B-cos/B-cos-v2/`.

We also evaluate on explanation methods developed specifically for Vision Transformers (Kolesnikov et al., 2021). In particular we use CGW1 (Chefer et al., 2020), Rollout (Abnar & Zuidema, 2020) and ViT-CX (CausalX, (Xie et al., 2022)). For CGW1 and Rollout we use the author provided implementation provided at `https://github.com/hila-chefer/Transformer-Explainability`. And for ViT-CX we use the official implementation available at `https://github.com/vaynexie/CausalX-ViT`.

