# OpenReview forum: "How to Probe: Simple Yet Effective Techniques for Improving Post-hoc Explanations"
_ICLR.cc/2025/Conference — ICLR 2025 Poster_

### Official Review · Reviewer_Awie · 2024-10-29

**Soundness:** 4
**Presentation:** 4
**Contribution:** 3
**Rating:** 8
**Confidence:** 4

**Summary:**

The authors find and support an interesting observation that the method of training the classifier layer of a model has a significant impact on the results of post-hoc attribution methods. Because many post-hoc attribution methods assume that model training does not have an impact, they find that this must be reconsidered, and in fact, simply modifying the method of training the last linear layer(s) can improve model accuracy and explainability.

**Strengths:**

This paper is very well written and planned. Not only are the approaches and findings very clear but the authors provide extensive support of their findings over numerous models, datasets, attribution methods, and metrics.

The choice to study multiple pre-training approaches adds significant strength to their arguments and findings.

The overall findings are simple, but impactful for future considerations of interpretable model design, post-hoc explainability, and improving model interpretation.

**Weaknesses:**

There are not significant weaknesses to address. There are minor spelling mistakes, but it does not hurt the delivery of the information.

**Questions:**

Would the authors suggest the development of future classification models take into consideration the information in this paper?

Do the authors think that a training loss could be created to further improve explainability as the minor differences in CE and BCE have a significant effect?

---

> ### Author Response · Authors · 2024-11-25
> **Author's Response to Reviewer Awie**
>
> Thank you for your review and feedback. We appreciate your assessment of our work as __“very well written”__ with __“clear findings”__ and __“impactful for future considerations of interpretable model design, post-hoc explainability, and improving model interpretation.”__
>
> We provide a pointwise response to individual concerns and queries below.
>
> _1. “Minor Spelling Mistakes”:_ __We have updated the manuscript with fixed spelling mistakes.__
>
> We thank the reviewer for their kind suggestion. We have taken a pass over the entire manuscript and fixed any spelling mistakes we could find. This has helped make our writing more clear and consistent.
>
> Specific fixes below:
> >L99: “explanations” -> “explanation methods”,
> L130: “Recent work have studied” -> “Recent work has studied”,
> L142: “method” -> “methods”,
> L172: “asses” -> “assess”,
> L305: “c is computes” -> “c computes”,
> L358: “repsect” -> “respect”,
> L428: “localizataion” -> “localization”,
> L541: “qualitiative” -> “qualitative”.
>
> _Response to Questions_
>
> _1. Would the authors suggest the development of future classification models take into consideration the information in this paper?_: __Our work demonstrates that the quality of outputs of explanation methods are closely tied to the downstream classification objective.__
>
> These findings are not common knowledge within the community and highlight a critical yet overlooked aspect of explainable AI (XI): __the interplay between model training and post-hoc explanations.__ While this is important for classification models themselves, it is even more critical for improving the alignment between attribution methods and classification objectives.
>
> We strongly believe this has further potential for future research and is important for the community to build on.
>
> _2. Do the authors think that a training loss could be created to further improve explainability as the minor differences in CE and BCE have a significant effect?_: __Yes, this is an excellent suggestion for future research, and our findings strongly support this direction!__
>
> Our findings and results suggest that the quality of explanations generated by popular attribution methods can be significantly influenced by the training objective of the classification layers.
>
> This naturally suggests that developing new loss functions specifically to enhance explainability is a promising direction for future research. For instance prior work has shown that
> * Incorporating sparsity constraints [1] or attribution priors [2] directly into the loss function or adding model guidance [3] during training could improve attribution quality, though such techniques might introduce additional computational costs.
> * Even a simple sparsity loss applied to the final classification layer has been shown to enhance attribution properties in some cases. This aligns well with our findings, which emphasize the importance of seemingly minor differences in training objectives (e.g., CE vs. BCE).
>
> That being said, one of the core takeaways from __our work is that it challenges a common belief within the explainable AI (xAI) community, that  post-hoc explanation methods often assume no influence from model training.__  Our systematic experiments demonstrate that even subtle changes in training objectives can have drastic implications for explainability, highlighting a critical area for future exploration.
>
> We once again thank the reviewer for their positive and encouraging remarks about our submission.
>
> We hope our response adequately addresses the reviewer’s concerns and provides clarifications to their questions.
>
> _References_
> 1. H. Cunningham, A. Ewart, L R. Smith, R. Huben and L. Sharkey. “Sparse Autoencoders Find Highly Interpretable Features in Language Models.” ICLR, 2024.
> 2. E. Weinberger,, J. D. Janizek and S. Lee. “Learning Deep Attribution Priors Based On Prior Knowledge.” NeurIPS, 2019.
> 3. F. Friedrich, W. Stammer, P. Schramowski, and K. Kersting. A Typology to Explore and Guide Explanatory Interactive Machine Learning. arXiv preprint arXiv:2203.03668, 2022.

---

> ### Comment · Reviewer_Awie · 2024-11-26
>
> I am pleased by the effort of the authors in responding to both my and the other reviewers' feedback. I am happy to leave my score as is. I would additionally agree with reviewer nDZf in suggesting the final version of this paper focuses on ViT over resnet experiments due to the increased popularity and usage of these models.

---

> > ### Author Response · Authors · 2024-11-26
> > **Author's Response to Reviewer Awie's Update**
> >
> > We thank the reviewer for their prompt reply to our response. We are pleased to find that the reviewer finds all their concerns well addressed, and are happy to retain their positive rating about our work.
> >
> > Thanks for the recommendation, yes we will thoroughly integrate the ViT results into the final revised version.

---

### Official Review · Reviewer_VgDa · 2024-10-29

**Soundness:** 3
**Presentation:** 4
**Contribution:** 3
**Rating:** 6
**Confidence:** 4

**Summary:**

The main motivation behind this work is that two models using the same training regime and ending at the same loss can produce two extremely different attributions for the same image. The authors demonstrate that the training paradigm for the final classification layer of a network is the most important decider in generating more precise attributions, regardless of the attribution method. They specifically show that a binary cross entropy trained output layer produces better attributions than a cross entropy trained output layer. The increase in attribution quality does typically come at the cost of <10% accuracy reduction when using a linear layer, but the accuracy can be improved by using a more complex output layer.

**Strengths:**

Overall, it is an interesting read and demonstrates some interesting results. The writing is generally clear, the issue is well defined, and the experiments are impactful. It is difficult for me to say exactly what the authors did well, other than that it is a good read.

1. In-depth motivation section, outlining the issues around generating consistently clear attributions
2. Plenty of qualitative results
3. Experiments over a variety of pre-trained models and datasets
4. The authors clearly show that this is an attribution-invariant issue.

**Weaknesses:**

There isn't any discussion of why there is an increase in accuracy and attribution quality with more complex output layers. Is it as simple as the layers being larger, or is there another reason? I assume proper train, test, and validation sets have been used?

**Questions:**

1. I am still confused as to why CE produces worse attribution than BCE. Could the authors explain this again?
    2. Also, why is it that the last output layer is so important? Why is the rest of the model have such little importance?

---

> ### Author Response · Authors · 2024-11-25
> **Author's Response to Reviewer VgDa (Part 1 of 2)**
>
> Thank you for your review and feedback. We appreciate your assessment of our work as __“an interesting read”__ with __“clear writing”__ and __“impactful experiments”__. We provide a pointwise response to individual concerns below.
>
> _Discussion of why there is an increase in accuracy and attribution quality with more complex output layers._: __We find Bcos-MLPs improve accuracy and attribution quality by better disentangling class-specific information from pre-trained features, which may not be linearly separable.__
>
> We thank the reviewer for their question! We aim to clarify it below:
>
> The features computed by self-supervised or vision-language backbones are not optimized to be linearly separable with respect to the classes for downstream tasks, limiting the utility of linear probes. The larger modeling capacity of B-cos MLPs allows them to extract information in a non-linear manner. This can lead to improvements in accuracy—interestingly, we find that this also significantly improves the ‘class-specificity’ and thus the localisation performance of explanation methods, more so than conventional MLPs (see Table 1 below).
>
> We attribute this to the alignment pressure introduced by B-cos layers [1]  that helps distill object-class relevant features and rely less on background context.
>
> __Table 1: Bcos MLP vs Std MLP: Change in Accuracy and GridPG Scores on ImageNet for Dino ResNet50 model.__
> |            | Bcos MLPs |           | Std Mlps  |           |
> |------------|-----------|-----------|-----------|-----------|
> | XAI Method | $\Delta^{acc}_{mlp-lp}$| $\Delta^{loc}_{mlp-lp}$ |$\Delta^{acc}_{mlp-lp}$ | $\Delta^{loc}_{mlp-lp}$ |
> | LIME       |       +2.9 |        **+38** |       +2.5 |        **+32** |
> | IntGrad    |       +2.9 |        **+22** |       +2.5 |        -6 |
> | GradCAM    |       +2.9 |         **+8** |       +2.5 |        -8 |
>
> We discuss this in the main paper, specifically in Sec 3.3 and find it validated empirically in our results in Sec 5.2. The findings do in fact suggest that the model with MLP probes relies on more specific ‘class-relevant’ features since it has the capacity to discover such features.
>
> We would be happy to further add an extended discussion about this in more detail to the paper, and how to interpret it better. This is very interesting and important for people to understand in the research community.
>
> _“I assume proper train, test, and validation sets have been used?”_: __Yes, we adhere to well-established standard dataset splits for all our experiments.__
>
> Specifically, train/minival/val splits for ImageNet [2], the standard train/val/test split for PascalVOC [3] and train/val for MS-COCO [4]. Details on dataset splits and implementation details are already included in the appendix (please see Appendix Section F]).
>
> _References:_
>
> 1. M Boehle, M. Fritz, and B. Schiele. B-cos networks: Alignment is all we need for interpretability. 2022 IEEE/CVF Conference on Computer Vision and Pattern Recognition (CVPR), pp. 10319–10328, 2022.
> 2. Tan, M., Le, Q.. EfficientNet: Rethinking Model Scaling for Convolutional Neural Networks. Proceedings of the 36th International Conference on Machine Learning, (2019).
> 3. http://www.pascal-network.org/challenges/VOC/voc2007/workshop/index.html.
> 4. https://cocodataset.org/

---

> ### Author Response · Authors · 2024-11-25
> **Author's Response to Reviewer VgDa (Part 2 of 2)**
>
> _Response to Questions_
>
> _1. “I am still confused as to why CE produces worse attribution than BCE. Could the authors explain this again?”:_ __We posit that BCE-trained probes yield more localized and class-specific attributions due to their non-shift-invariant optimization.__
>
> This is one of the core findings in our work which we explain in more detail below.
>
> While the BCE- and the CE-based probes rely on the same frozen backbones, BCE-trained models learn a combination of more-class discriminative features as compared to CE-trained models, even for simple linear classifiers (probes).
>
> We hypothesize (in Section 3.2) that the shift-invariance of the CE loss contributes to its poorer attribution quality. Specifically, the Softmax function within CE is invariant to adding a constant shift to all logits. This results in multiple (infinite) linear classifiers that are equivalent under the CE-based optimization. As most attribution methods, in some form or another, rely on the models’ backbone + classifier weights to compute the importance attributions, it cannot be expected that for CE-trained probes the attributions are calibrated such that ‘positive’ attributions will always be class specific.
>
> In contrast, the __BCE loss is not shift-invariant.__ It penalizes constant positive shifts to non-target classes, thereby biasing the model toward `class-specific’ features. As a result, __BCE-trained probes produce better-calibrated explanations, with higher class-specificity and improved localization of class objects__ (as is demonstrated by our results both qualitatively and quantitative in Sec 5.1).
>
> We will include a more detailed explanation in the final version of the paper for increased clarity.
>
>
> _2. “Also, why is it that the last output layer is so important? Why is the rest of the model have such little importance?”_: __That is one of our core findings!__
>
>
> Foundational vision models, pre-trained on large datasets with self-supervised or vision-language objectives, learn general features that transfer well to downstream tasks.
> Probing these pre-trained models is a commonly used approach for downstream tasks. Understanding how to obtain good explanations for the combined models is thus highly relevant in practice.
>
> In our work we make this very surprising yet important finding, that the training details of a pre-trained model’s classification layer (<10% of model parameters) play a crucial role, much more than the pre-training scheme itself. We study this empirically and find that even for simple linear probes, the localisation ability of the explanations highly depends on how the probes are trained—BCE-trained probes produce better localized explanations as compared to CE-trained probes (see Section 5.1 in main paper).
>
> This is interesting as different probes (CE / BCE) compute their outputs by nothing but a weighted mean of the frozen backbone features. As such, the explanations are largely dominated by the backbone features. Interestingly, BCE induces a combination of features that leads to more localizing explanations (which we now __also show for Vision Transformers!__).
>
> Further, we note that features from pre-trained backbones might not be linearly separable with respect to downstream classes, thus we also propose a simple tweak, by using B-cos MLP probes leads to improvements in both downstream performance and explanation quality (see Section 5.2 in main paper).
>
> These findings highlight a critical yet overlooked aspect of explainable AI (XI): __the interplay between model training and post-hoc explanations.__ We believe this has further potential for future research and is important for the community to build upon.
>
> We once again thank the reviewer for appreciating our efforts on clear writing and sharing our enthusiasm for the interesting and impactful findings.
>
> We hope our response adequately addresses the reviewer’s concerns and queries, and we would be grateful if the reviewer considered updating their score.

---

> > ### Comment · Reviewer_VgDa · 2024-11-26
> > **Clarifications**
> >
> > The authors have responded to my questions in a satisfactory manner. However, it is kind of puzzling that this approach improves the accuracy and the explainability. If method improves that accuracy, why even position the contributions as an XAI paper? Anyways, the paper is good enough given the performed revision.

---

> ### Author Response · Authors · 2024-11-27
> **Author's Response to Reviewer VgDa's Update**
>
> We are happy to have resolved the reviewer's concerns and appreciate the positive feedback.
>
> Regarding the remaining comment on MLP probes: in general, it should be expected that more complex probes yield higher accuracy due to their greater modeling capacity, see e.g. [1]. However, to the best of our knowledge, a similarly strong link to improvements in visual explanation quality as demonstrated in our results has not been reported before—i.e., the fact that it is possible to simultaneously improve __both the__ _explanations __and__ accuracy_.
>
> We believe this to be of high interest to the XAI community in particular as it highlights that accuracy and interpretability need not be at odds, which is a commonly held notion. In fact, our results suggest that particularly for inherently interpretable models, higher accuracy might indeed _improve explanation quality_, as the explanations are a reflection of the model's internal 'decision process' [2].
>
> We again thank the reviewer for their valuable feedback and insightful questions, and are pleased with their positive outlook on our work.
>
> _References_
> 1. J. Hewitt, & P. Liang. Designing and Interpreting Probes with Control Tasks. ACL, 2019.
> 2. M Boehle, M. Fritz, and B. Schiele. B-cos networks: Alignment is all we need for interpretability. 2022 IEEE/CVF Conference on Computer Vision and Pattern Recognition (CVPR), 2022.

---

### Official Review · Reviewer_7mwF · 2024-10-31

**Soundness:** 2
**Presentation:** 2
**Contribution:** 2
**Rating:** 5
**Confidence:** 5

**Summary:**

This paper discovers and demonstrates the strong dependence of post-hoc importance attribution methods on the training details of the classification layer of the pre-trained model. Based on this findings, the paper also proposes a simple but effective adjustment to the classification layer to significantly improve the quality of model explanations.

**Strengths:**

This paper reveals and demonstrates the strong dependence of post-hoc importance attribution methods on the training details of the classification layer in pre-trained models.

**Weaknesses:**

1. The experimental method is limited to ResNet50, and the results are not extensive enough. Thus, experimental results are not convincing enough to verify the effectiveness of their methods.

2. The contribution of this article is not enough. The author discovered the impact of training details on post-processing methods, but the evaluation metrics used and the subsequent B-cos model are not the author's innovation.

3. [minor] Figures in this paper have obvious flaws. It will be better that authors carefully revise their figures.

**Questions:**

1. In the case of backbone freezing, increasing classifier parameters can improve performance. Is the design of B-cos MLP necessary? Is MLP not possible?
2. Can you provide more loss function results to verify Softmax Shift-Invariance? How about the cross-entropy loss?

---

> ### Author Response · Authors · 2024-11-25
> **Author's Response to Reviewer 7mwF (Part 1 of 2)**
>
> Thank you for your efforts in reviewing our paper and for your feedback. We provide a pointwise response to individual concerns below.
>
> ***
>
> _1. “Limited experimental method to ResNets”_: __We have expanded our experiments to include additional architectures (ViTs), attribution methods, and many more qualitative results.__
>
>
> We summarise the key observations on ViTs below:
>
> We evaluated both conventional ViTs (using CGW1 [1], ViT-cx [2], and Rollout [3]) and inherently interpretable B-cos ViTs [4], and the __updated experiments fully corroborate our previous findings__. In particular we find:
>
> * BCE-trained classifiers to outperform CE-trained classifiers in interpretability scores across all pre-trainings and explanation methods.
> * BCE-trained models achieve similar accuracy as CE-trained models.
> * Probing ViT backbones with B-cos MLPs yields improvements in both interpretability and accuracy.
>
> For a more detailed clarification please refer to the common response here: https://openreview.net/forum?id=57NfyYxh5f&noteId=xEjpQlnGLg
>
> ***
>
> _2. “The contribution of this article is not enough … the evaluation metrics used and the subsequent B-cos model are not the author's innovation.”_: __We clarify our contributions as well as highlight our novel findings and their implications for XAI research.__
>
> We address the concerns of the reviewer regarding our contributions below-
>
> __Clarification regarding the contributions__
>
> First, we fully agree with the reviewer that the evaluation metrics used and the inherently interpretable B-cos models are not our innovations. They are simply a means to an end, which we use to evaluate our findings across a diverse set-of attribution methods and well-established metrics in XAI literature for a comprehensive experimental setting.
>
> Please note, we do not claim any of these as our innovations and have ensured that is not the case.
>
> We take this opportunity to highlight our __key contributions and findings__ more clearly:
>
> __Uncovering a Critical Problem in Attribution Methods__
> We show for the first time that model training objectives can significantly affect attribution methods, challenging the assumption that post-hoc attributions are agnostic to model training. In particular, BCE-trained models consistently yield better explanations than CE-trained models (see Section 5.1).
>
> __Simple Adjustments Improve Explanations__
> We find that replacing the final classification layer with non-linear B-cos MLP probes not only boosts downstream performance across pre-trained backbones but also the ‘class-specific’ localization ability of attribution methods (see Section 5.2).
>
> _To the best of our knowledge, __our work is the first to systematically study this interplay between attribution methods, the model’s training objective and the classifier’s complexity__._
>
> __Holistic Evaluation Setting__
> We convincingly demonstrate the generality of our findings by conducting a detailed study that includes five pre-training frameworks, a suite of ten attribution methods, both convolutional- and transformer-based architectures evaluated across multiple popular datasets.
>
> __Compatibility of Inherently Interpretable Models with Self-Supervised Frameworks__
> Our final contribution is showing for the first time that B-Cos networks are compatible with popular self-supervised pre-training paradigms (DINO, MoCo, BYOL) and retain their favorable ‘class-specific’ attributions and faithfulness properties.
>
> __Reproducibility__
> All code, including training recipes and evaluation scripts are open-sourced for the community (see https://anonymous.4open.science/r/how-to-probe-iclr/).
>
> Importantly, our findings uncover a new crucial aspect of explainable artificial intelligence (XAI) that should be considered when using existing attribution methods or developing new ones in the future.
>
> ***
>
> _3. “[Minor] obvious flaws in figures”_: __We have carefully reviewed all figures to ensure proper formatting and consistency within the manuscript.__
>
> We appreciate the reviewer’s feedback regarding the figures. However, as the comment is general, it would be very helpful if the reviewer could please specify the flaws they identified.
>
> We welcome any additional suggestions from the reviewer regarding specific figures that could further improve the paper’s clarity.

---

> ### Author Response · Authors · 2024-11-25
> **Author's Response to Reviewer 7mwF (Part 2 of 2)**
>
> _Response to Questions_
>
> ***
>
> _1. “In the case of backbone freezing, increasing classifier parameters can improve performance. Is the design of B-cos MLP necessary? Is MLP not possible?”_: __We find B-cos MLPs lead to more consistent improvements in accuracy and explanation quality, in contrast to conventional MLPs.__
>
> We thank the reviewer for this important question.
>
> __This is one of our core findings!__  The larger modeling capacity of B-cos MLPs allows them to extract information in a non-linear manner. This can lead to improvements in accuracy—interestingly, we find that this also significantly improves the localisation performance of explanation methods, more so than conventional MLPs.
>
> __B-cos MLPs__ lead to more consistent improvements in accuracy and explanation quality, in contrast to conventional MLPs that (in most cases) only lead to increase in accuracy (see Table 1 below). We attribute this to the alignment pressure introduced by B-cos layers [1]  that helps distill object-class relevant features and rely less on background context.
>
> __Table 1: Bcos MLP vs Std MLP: Change in Accuracy and GridPG Scores on ImageNet for Dino ResNet50 model.__
> |            | Bcos MLPs |           | Std Mlps  |           |
> |------------|-----------|-----------|-----------|-----------|
> | XAI Method | $\Delta^{acc}_{mlp-lp}$| $\Delta^{loc}_{mlp-lp}$ |$\Delta^{acc}_{mlp-lp}$ | $\Delta^{loc}_{mlp-lp}$ |
> | LIME       |       +2.9 |        **+38** |       +2.5 |        **+32** |
> | IxG    |       +2.9 |        **+48** |       +2.5 |        -0.5 |
> | IntGrad    |       +2.9 |        **+22** |       +2.5 |        -6 |
> | GradCAM    |       +2.9 |         **+8** |       +2.5 |        -8 |
>
> We present these quantitatively in figures __E6, E7__ (for the accuracy vs GridPG score computed on ImageNet) of the Appendix. This is also seen similarly for ViTs (Vision Transformers), please see Table 3. in the common response here: https://openreview.net/forum?id=57NfyYxh5f&noteId=xEjpQlnGLg
>
> We will revise the main paper to clarify these findings and their implications.
>
> ***
>
> _2. “Can you provide more loss function results to verify Softmax Shift-Invariance? How about the cross-entropy loss?”_: __We mathematically show that if Softmax is Shift-Invariant, then the Cross-Entropy loss is also Shift-Invariant.__
>
> We thank the reviewer for this question. If we understood the question correctly, the reviewer wants to understand the implications on the Cross-Entropy Loss.
>
> Below, we provide additional details to clarify softmax shift-invariance and its implications for the CE loss.
>
> __Softmax-Shift Invariance__
>
> Let $\hat y_{c, i}$ denote the probe's output logit for class $c$ and input $i$, and $t_{c, i}$ the respective one-hot encoded label, then
>
> Softmax: $\frac{\exp(\hat y_{c,i}+\delta_i)}{\sum_k \exp(\hat y_{k,i}+\delta_i)} = \frac{\exp(\hat y_{c,i})\exp(\delta_i)}{\sum_k \exp(\hat y_{k,i})\exp(\delta_i)}$ (Equation 2 in the main paper).
>
> __Cross-Entropy Loss Invariance__
>
> The cross-entropy (CE) loss applies the Softmax operation to the raw logits output from the classifier. The CE loss for an image $i$ with a ground-truth class $c$ is defined as:
>
> $\mathcal L_{\text{CE}, i} = -\sum_c\log \frac{\exp\left(\hat y_{c, i}\right)}{\sum_k \exp\left(\hat y_{k, i}\right)} \times {t_{c,i}}$
>
> When a shift $\delta_i$​ is applied, the logits become $\hat y_{c,i}'  = \hat y_{c,i}+\delta_i$. The CE loss then becomes
>
> $\mathcal L_{\text{CE}, i}' = -\sum_c\log \frac{\exp\left(\hat y_{c, i}'\right)}{\sum_k \exp\left(\hat y_{k, i}' \right)} \times {t_{c,i}} = -\sum_c\log \frac{\exp\left(\hat y_{c, i} +\delta_i \right)}{\sum_k \exp\left(\hat y_{k, i} + \delta_i \right)} \times {t_{c,i}}$
>
> Using the shift-invariance property of softmax (Equation (2)), we observe:
>
> $\mathcal L_{\text{CE}, i}' = -\sum_c\log \frac{\exp\left(\hat y_{c, i} \right)}{\sum_k \exp\left(\hat y_{k, i} \right)} \times {t_{c,i}} = \mathcal L_{\text{CE}, i}$
>
> Thus, the CE loss remains unchanged under any constant shift $\delta_i$.
>
> We will include this extended explanation in the final version of the paper for added clarity.
>
> ***
>
> We again thank the reviewer for the feedback and suggestions. We hope our clarifications and expanded analysis adequately addresses all concerns, and we would be grateful if the reviewer considered updating their score.
>
> _References_
> 1. M Boehle, M. Fritz, and B. Schiele. B-cos networks: Alignment is all we need for interpretability. 2022 IEEE/CVF Conference on Computer Vision and Pattern Recognition (CVPR), pp. 10319–10328, 2022.

---

> ### Author Response · Authors · 2024-11-29
> **Author's Follow-up To Reviewer 7mwF - we kindly ask if the rebuttal has addressed all concerns**
>
> Dear Reviewer 7mwF,
>
> We are grateful for the constructive feedback and thoughtful consideration. As the discussion period concludes soon, we kindly ask if our response and additional experiments have addressed all concerns.
>
> We’d be happy to provide more clarifications if needed.
>
> Many thanks!

---

> ### Author Response · Authors · 2024-12-02
> **Follow-up to Reviewer 7mwF Regarding Rebuttal Feedback**
>
> Dear Reviewer 7mwf,
>
> We deeply appreciate the constructive feedback, as well as the time and effort invested in evaluating our work. As we approach the conclusion of the discussion phase, we kindly ask if our response and additional experiments have addressed all concerns, and might prompt a re-assessment of the current score.
>
> We'd be happy to provide more clarifications if required.
>
> Thank you once again for your time and consideration.

---

### Official Review · Reviewer_nDZf · 2024-11-02

**Soundness:** 2
**Presentation:** 4
**Contribution:** 3
**Rating:** 6
**Confidence:** 5

**Summary:**

The paper challenges the tradition notation that model explanations are independent of training methods by demonstrating that the quality of attributions for pre-trained models depends significantly on how the classification head is trained. It shows that using binary cross-entropy (BCE) loss instead of conventional cross-entropy (CE) loss leads to marked improvements in interpretability metrics across several visual pre-training frameworks. Furthermore, it is found that the non-linear B-cos MLP probes boost the class-specific localization ability of attribution methods.

**Strengths:**

1.	Clarity and Organization: The paper is exceptionally well-written and structured, enhancing readability and accessibility of the key finding
2.	The study reveals that training probes using binary cross-entropy (BCE) loss instead of the traditional cross-entropy (CE) loss consistently enhances interpretability metrics. The analysis of the Softmax Shift-Invariance Issue in interesting and insightful. This could have substantial implications for various DNN-based applications.
3.	The improvements in interpretability metrics are shown to be consistent across various training methods for the visual encoder. The robustness of these findings was thoroughly validated using diverse learning paradigms, including supervised, self-supervised and CLIP.

**Weaknesses:**

1.	(Major) Limited Model Diversity: The research exclusively utilizes the ResNet50 model backbone, which  canot adequately represent the behavior across various architectures. Testing additional backbones, especially Vision Transformers (ViTs), and incorporating explanation methods tailored for these models (referenced as [1][2][3]), would provide a more robust validation of the findings.
2.	Inclusion of Additional Methods: The paper could be strengthened by including more population perturbation-based methods, such as RISE [4] and Score-CAM [5], to further substantiate the interpretability improvements.
3.	Selection of Examples: Concerns arise regarding whether the examples shown in Figures 1 and 6 are cherry-picked, especially since the GridPG Score in Figure 5 suggests that the BCE model does not always perform perfectly. Including a broader range of examples, particularly where the BCE model scores lower on the GridPG, would offer a more comprehensive understanding and enhance the paper's credibility.

I would be happy to improve my rating of the paper if these issues are addressed thoroughly.

[1] Transformer interpretability beyond attention visualization.

[2] Generic Attention-model Explainability for Interpreting Bi-Modal and Encoder-Decoder Transformers.

[3] Vit-cx: Causal explanation of vision transformers.

[4] RISE: Randomized Input Sampling for Explanation of Black-box Models.

[5] Score-CAM: Score-Weighted Visual Explanations for Convolutional Neural Networks.

**Questions:**

See weaknesses.

---

> ### Author Response · Authors · 2024-11-25
> **Author's response to Reviewer nDZf**
>
> Thank you for your review and feedback. We appreciate your assessment of our work as __“interesting and insightful”__ and __“exceptionally well-written and structured”__. We provide a pointwise response to individual concerns below.
>
> _1.“Limited Model Diversity”_: __We have expanded our experiments to include ViTs and incorporated these results into the manuscript.__
>
> We summarise the key observations on ViTs below:
>
> We evaluated both conventional ViTs (using CGW1 [1], ViT-cx [2], and Rollout [3]) and inherently interpretable B-cos ViTs [4], and the updated experiments fully corroborate our previous findings. In particular we find:
>
> * BCE-trained classifiers to outperform CE-trained classifiers in interpretability scores across all pre-trainings and explanation methods.
> * BCE-trained models achieve similar accuracy as CE-trained models.
> * Probing ViT backbones with B-cos MLPs yields improvements in both interpretability and accuracy.
>
> For a more detailed clarification please refer to the common response here: https://openreview.net/forum?id=57NfyYxh5f&noteId=xEjpQlnGLg
>
>
> _2. “Inclusion of Additional Perturbation-based explanation methods”_:  __We updated our submission additionally with results for ScoreCAM [6], which are very well consistent with our previous findings.__
>
> * Specifically, the interpretability scores with ScoreCAM [5] explanations for BCE-trained classifiers significantly outperform CE-trained classifiers across all pre-trainings (supervised, MoCo, BYOL, DINO) on both conventional and Bcos backbones (see following Table 1):
>
> __Table 1: BCE vs CE Probing—Localization Scores on ImageNet for conventional and Bcos backbones with different pre-trainings.__
> | Backbone | Pre-training |  Loc%$_{CE}$  |  Loc%$_{BCE}$ | $\Delta_{bce-ce}^{loc}$ |
> |:--------:|:-----------:|:----:|:----:|:-------------:|
> |   RN50   |    MoCov2   | 52.6 | 54.4 |      **+1.8**      |
> |   RN50   |     BYOL    | 38.1 | 40.5 |      **+2.4**      |
> |   RN50   |     DINO    | 44.2 | 55.9 |      **+11.7**     |
> |   BRN50  |    MoCov2   | 30.2 | 43.2 |       **+13.0**      |
> |   BRN50  |     BYOL    | 45.3 | 50.2 |      **+4.9**      |
> |   BRN50  |     DINO    |  51.0  | 67.7 |      **+16.7**     |
>
> We have updated the results in the manuscript under the Appendix in Section E1 and added Table E3, and also referred to this in Section 5 of the main paper.
>
> _3. “Concerns regarding the selection of examples”_: __We have updated our submission to address the reviewer's concern regarding the selection of visual examples by adding many more qualitative examples to better reflect the range of different localization scores.__
>
> To ensure a balanced and comprehensive view of our findings, we have expanded the qualitative analysis to include a diverse set of randomly sampled examples. These examples, now detailed in Section D.4 of the Appendix, are presented in Figures D10-13, and include a mix of high and low GridPG scores for BCE vs. CE probing and MLP probing. Each example is annotated with its localization score to provide clearer context and further transparency.
>
> We once again thank the reviewer, firstly for appreciating our efforts and sharing our enthusiasm for the interesting and insightful findings, and secondly for providing great feedback that makes our submission stronger!
>
> We hope our revisions and expanded analysis adequately addresses all concerns, and we would be grateful if the reviewer considered updating their score.
>
> _References:_
> 1. H. Chefer, S. Gur, and L. Wolf. Transformer interpretability beyond attention visualization. 2021 IEEE/CVF Conference on Computer Vision and Pattern Recognition (CVPR), pp. 782–791, 2020.
> 2. W. Xie, X. H. Li, G. C. Cao, and N. L. Zhang. Vit-cx: Causal explanation of vision transformers. In International Joint Conference on Artificial Intelligence, 2022.
> 3. S. Abnar and W. Zuidema. Quantifying attention flow in transformers. In Annual Meeting of the Association for Computational Linguistics, 2020.
> 4. M. Boehle, N. Singh, M. Fritz, and B. Schiele. B-cos Alignment for Inherently Interpretable CNNs and Vision Transformers, 2023.
> 5. H. Wang, Z. Wang, M. Du, F. Yang, Z. Zhang, S. Ding, P. Mardziel, and X. Hu. Score-cam: Score-weighted visual explanations for convolutional neural networks. In 2020 IEEE/CVF Conference on Computer Vision and Pattern Recognition Workshops (CVPRW), pp. 111–119, 2020.

---

> > ### Comment · Reviewer_nDZf · 2024-11-26
> >
> > Thanks for the authors' responses, which have well addressed my concerns. I therefore raised my score to 6.
> >
> > For the revised version of the paper, I would recommend that the authors more prominently include the results related to  ViTs in the main text. This is crucial to demonstrate that the findings described in the paper are not confined to a single model, such as ResNet50.

---

> ### Author Response · Authors · 2024-11-26
> **Author's Response to Reviewer nDZf's Update**
>
> We thank the reviewer for their prompt reply to our response. We are pleased to find that the reviewer finds all their concerns well addressed.
>
> Thanks for the recommendation, yes we will thoroughly integrate the ViT results into the final revised version.

---

### Author Response · Authors · 2024-11-25
**Author’s Response to Common Concern Regarding Limited Model Diversity**

In the following, we address the common concern that was shared by Reviewers (nDZf, 7mwF) regarding limited model diversity.

__We updated our submission with results for ViTs, which fully corroborate our previous findings; we thank the reviewers for this excellent suggestion which we believe to further strengthen our submission.__

In particular, following the reviewer’s suggestion, we evaluated on both the conventional ViTs interpretability using CGW1 [1], ViT-cx (CasualX) [2], Rollout [3] and the inherently-interpretable B-cos ViTs [4]. To summarize we observe the following:

* The interpretability scores for BCE-trained classifiers consistently outperform CE-trained classifiers across all pre-trainings (supervised, MoCo, DINO) and explanation methods (see Table 1,2 below).
* The classification accuracy on ImageNet for models trained with BCE loss matches the accuracy of models trained with CE loss (See Table 1,2 below).
* When probing the ViT backbones with Bcos MLPs, we find improvements in both interpretability scores and accuracy (see Table 3).

__Table 1:Accuracy and GridPG Localization Scores for Conventional ViTs on [1,2,3]__
|               |              | ACC % |      |  |           Localization%    |               |
|---------------|--------------|:-----:|:----:|:-------------:|:-------------:|:-------------:|
| Backbone | Pre-training |   CE  |  BCE |   $\Delta_{bce-ce}^{CGW1}$ [1]  | $\Delta_{bce-ce}^{Rollout}$ [2] | $\Delta_{bce-ce}^{CausalX}$ [3] |
|   ViT-B/16  |      Sup     |  73.2 | 72.8 |     **+15.7**     |      **+0.4**     |      **+0.7**     |
|   ViT-B/16  |     DINO     |  77.2 | 77.6 |     **+11.7**     |      **+4.8**     |      **+5.3**     |
|   ViT-B/16  |    MoCov3    |  76.3 | 75.1 |      **+4.4**     |      **+3.3**     |      **+3.8**     |

__Table2: Accuracy and GridPG Localization Scores for Bcos ViTs [4]__
|  Backbone | Pre-training | Acc$_{CE}$% |  Acc$_{BCE}$ | $\Delta_{bce-ce}^{Bcos}$ [4] |
|:--------------:|:------------:|:-----:|:----:|:----------:|
|  B-ViT$_c$-B/16  |      Sup     |  77.3 | 78.1 |    **+23.5**   |
| B-ViT$_c$-S/16 |     DINO     |  73.2 | 73.4 |     **+32.7**  |
| B-ViT$_c$-B/16 |     DINO     |  77.1 | 77.3 |     **+20.4**  |


__Table 3: Bcos MLP vs Linear Probing—Accuracy and GridPG Localization Scores on B-ViTs [4]__
| Backbone     | Classifier   | Acc% | $\Delta_{mlp-lp}^{acc}$ | Loc.% | $\Delta_{mlp-lp}^{loc}$ |
|--------------|--------------|------|-------------------------|-------|-------------------------|
| B-ViT$_c$-S/16 | Linear Probe | 73.4 | --                      |  79.9 | --                      |
| B-ViT$_c$-S/16 | Bcos-MLP-2   | 74.2 |                     **+0.8** |  80.4 |                     **+0.5**|
| B-ViT$_c$-S/16 | Bcos-MLP-3   | 74.7 |                     **+1.3** |  82.4 |                     **+2.5** |
| B-ViT$_c$-B/16 | Linear Probe | 77.3 | --                      |  80.3 | --      |
| B-ViT$_c$-B/16 | Bcos-MLP-2   | 78.2 |                     **+0.9** |  82.1 |                     **+1.8** |
| B-ViT$_c$-B/16 | Bcos-MLP-3   | 79.7 |                     **+2.4** |  83.4 |                     **+3.1** |

We have updated the results in the revised submission under the Appendix in Section C (BCE vs. CE Probing — Table C1, C2: accuracy and localization scores; MLP Probing—Table C3) and referred to this in Section 5 of the main paper.

__In short, the new results complement our previous findings and further highlight the general applicability of the findings to diverse architectures and backbones.__

For individual queries please refer to the respective rebuttal sections.

References:
1. H. Chefer, S. Gur, and L. Wolf. Transformer interpretability beyond attention visualization. 2021 IEEE/CVF Conference on Computer Vision and Pattern Recognition (CVPR), pp. 782–791, 2020.
2. S. Abnar and W. Zuidema. Quantifying attention flow in transformers. In Annual Meeting of the Association for Computational Linguistics, 2020.
3. W. Xie, X. hui Li, C. C. Cao, and N. L. Zhang. Vit-cx: Causal explanation of vision transformers. In International Joint Conference on Artificial Intelligence, 2022.
4. M. Boehle, N. Singh, M. Fritz, and B. Schiele. B-cos Alignment for Inherently Interpretable CNNs and Vision Transformers, 2023.

---

### Author Response · Authors · 2024-11-25
**Author's Summary of Rebuttal Discussion**

We would like to thank all the reviewers for their constructive feedback and are pleased that they find our paper to be __“exceptionally well-written and structured"__ (nDZf, VgDa, Awie), our experiments to be __“well defined, planned, [as well as] thoroughly validated”__, showing __“impactful, interesting and insightful results"__ (nDZf, VgDa, Awie) that can have __“substantial implications for various DNN-based application(s)”__ (nDZf).

This is a summary of main concerns and how we addressed them:

1._"Limited Model Diversity"_ (nDZf, 7mwF): __We updated our submission with results for ViTs, which fully corroborate our previous findings; we thank the reviewers for this excellent suggestion which we believe to further strengthen our submission.__
(refer to: https://openreview.net/forum?id=57NfyYxh5f&noteId=xEjpQlnGLg)

2._“Inclusion of Additional Perturbation-based explanation methods” (nDZf)_: __We updated our submission additionally with results for ScoreCAM which are very well consistent with our previous findings.__  (refer to: https://openreview.net/forum?id=57NfyYxh5f&noteId=D3n6dTt028)

3._“Concerns regarding the selection of examples” (nDZf)_: __We have updated our submission to address the reviewer's concern regarding the selection of visual examples by adding many more qualitative examples to better reflect the range of different localization scores.__ (refer to: https://openreview.net/forum?id=57NfyYxh5f&noteId=D3n6dTt028)

Importantly, as the reviewers agree (nDZf, VgDa, Awie), __our findings uncover a new crucial aspect of explainable artificial intelligence (XAI)__ that should be considered when using existing attribution methods or developing new ones in the future.

We hope our revisions and expanded analysis adequately addresses all concerns from the reviewers, and we would be happy to provide any further clarifications.

_Note_
* _All additions to the content in the revised submission have been made in green text, to clearly highlight the changes._
* _Please refresh your browser tab in case the tables / math does not render properly on OpenReview._

__Updates__

* Reviewer (nDZf) have already taken the time to assess our changes and response. They have increased their rating to a positive score, and find our responses to have "well addressed their concerns".
* Reviewer (Awie) have also acknowledged their satisfaction with all our responses, and have retained their positive score.
* Reviewer (VgDa) are satisfied with our response to their questions, and have retained their positive score.

---

### Meta-Review · Area_Chair_JoQP · 2024-12-19

**Metareview:**

This paper reveals that post-hoc importance attribution methods are strongly influenced by the training details of the classification layer in a pre-trained model. Building on these findings, the authors propose a simple yet effective adjustment to the classification layer, resulting in significantly improved model explanations. Considering the paper's novelty and the generally positive feedback from reviewers, I am inclined to accept this submission.

**Additional Comments On Reviewer Discussion:**

1. **Limited Model Diversity (nDZf, 7mwF):**
   The authors have included results for Vision Transformers (ViTs), which fully corroborate their previous findings. This addition, suggested by the reviewers, further strengthens the submission.

2. **Inclusion of Additional Perturbation-based Explanation Methods (nDZf):**
   The authors have added results for ScoreCAM, which remain consistent with their earlier observations. This inclusion offers a more comprehensive assessment of explanation methods.

3. **Concerns Regarding the Selection of Examples (nDZf):**
   The authors have expanded the range of qualitative examples to better represent various localization scores, thus addressing the reviewer’s concerns about the selection of visual examples.

Importantly, as noted by the reviewers (nDZf, VgDa, Awie), these findings highlight a crucial new aspect of explainable artificial intelligence (XAI) that should be considered when employing existing attribution methods or developing new ones in the future.

The authors hope that these revisions and the additional analysis adequately address all reviewer concerns. They remain available to provide further clarifications if needed.

---

### Decision · Program_Chairs · 2025-01-22

Accept (Poster)